# Redox-responsive polymer micelles co-encapsulating immune checkpoint inhibitors and chemotherapeutic agents for glio-blastoma therapy

Zhiqi Zhang [1], Xiaoxuan Xu[1], Jiawei Du[1], Xin Chen[2], Yonger Xue [3], Jianqiong Zhang[1,2], Xue Yang[1], Xiaoyuan Chen [4,5,6,7] ✉, Jinbing Xie[1] ✉ & Shenghong Ju [1] ✉

Immunotherapy with immune checkpoint blockade (ICB) for glioblastoma (GBM) is promising but its clinical efficacy is seriously challenged by the blood-tumor barrier (BTB) and immunosuppressive tumor microenvironment. Here, anti-programmed death-ligand 1 antibodies (aPD-L1) are loaded into a redox-responsive micelle and the ICB efficacy is further amplified by paclitaxel (PTX)-induced immunogenic cell death (ICD) via a co-encapsulation approach for the reinvigoration of local anti-GBM immune responses. Consequently, the micelles cross the BTB and are retained in the reductive tumor micro-environment without altering the bioactivity of aPD-L1. The ICB efficacy is enhanced by the aPD-L1 and PTX combination with suppression of primary and recurrent GBM, accumulation of cytotoxic T lymphocytes, and induction of long-lasting immunological memory in the orthotopic GBM-bearing mice. The co-encapsulation approach facilitating efficient antibody delivery and com-bining with chemotherapeutic agent-induced ICD demonstrate that the chemo-immunotherapy might reprogram local immunity to empower immu-notherapy against GBM.

Glioblastoma (GBM) is a clinically formidable and challenging brain tumor with a poor prognosis, even with the standard of care, including surgery followed by chemotherapy and radiotherapy[1,2]. Immune check-point blockade (ICB) therapy targeting programmed death-1 (PD-1)/programmed death-ligand 1 (PD-L1) axis has become increasingly prominent in potent treatment of various malignancies, such as mela-noma and non-small-cell lung cancer over the past decades[3-6]. Present evidences indicate that the overexpressed PD-L1 is a major prognostic biomarker and predominant therapeutic target for GBM[7,8], showing the great potential of ICB therapy for GBM. Unfortunately, recent clinical

[1]Nurturing Center of Jiangsu Province for State Laboratory of AI Imaging & Interventional Radiology, Department of Radiology, Zhongda Hospital, Medical School, Southeast University, Nanjing 210009, China. [2]Department of Microbiology and Immunology, Medical School, Southeast University, Nanjing 210009, China. [3]Center for BioDelivery Sciences, School of Pharmacy, Shanghai Jiao Tong University, Shanghai 200240, China. [4]Departments of Diagnostic Radiology, Surgery, Chemical and Biomolecular Engineering, and Biomedical Engineering, Yong Loo Lin School of Medicine and College of Design and Engineering, National University of Singapore, Singapore 119074, Singapore. [5]Nanomedicine Translational Research Program, NUS Center for Nanomedicine, Yong Loo Lin School of Medicine, National University of Singapore, Singapore 117597, Singapore. [6]Clinical Imaging Research Centre, Centre for Translational Medicine, Yong Loo Lin School of Medicine, National University of Singapore, Singapore 117599, Singapore. [7]Institute of Molecular and Cell Biology, Agency for Science, Technology, and Research (A*STAR), 61 Biopolis Drive, Proteos, Singapore 138673, Singapore. ✉e-mail: chen.shawn@nus.edu.sg; xiejb@seu.edu.cn; jsh@seu.edu.cn

trials demonstrated that ICB therapy has poor efficacy for most GBM patients[9–11], which is mainly due to the restricted penetration of the ICB antibodies and the immunosuppressive tumor microenvironment (TME). In particular, the blood-brain barrier (BBB) innately hinders the transportation of biologics, including therapeutic antibodies to the brain, even though it is partially destroyed into blood-tumor barrier (BTB) as brain tumor lesions expand[12]. Furthermore, the functional ICB therapy requires a viable T cell compartment[13,14], while rampant T cell dysfunction has long been established as a hallmark of the GBM tumor immune microenvironment[15]. Therefore, innovative strategies to increase response level of the ICB therapy for GBM by effectively delivering antibodies across the BTB and simultaneously modulating the immunosuppressive milieu is urgently required.

Current clinical trials using chemotherapy to enhance immunotherapy response have shown favorable outcomes in many types of solid tumors[16–19]. Recent studies suggested that chemotherapy-induced immune cell death (ICD) is accompanied by the production of damage-associated molecular patterns (DAMPs) from dying cells and facilitating dendritic cell (DC) maturation and antecedent T lymphocytes activation[20–23]. Thus, chemotherapy that can activate naïve T cells might considerably augment the efficacy of ICB antibodies against GBM. Although several antibody-based brain delivery techniques have been developed and evaluated in preclinical and clinical studies, including antibody engineering via chemical modifications[24], none of them have actually been allowed for the widespread use of the delivery of monoclonal antibodies targeting GBM[25], due to their low loading capacity which hampered binding affinity and potential immunogenicity[26]. In addition, systemic administration of chemotherapeutic agents could lead to lymphodepletion[27], which may weaken the antitumor effects of ICB therapy. The uniquely appealing features of realizing the targeted delivery and controlled release of drugs[28,29] make nano-based drug delivery system (NanoDDS) as promising tools to co-deliver antibodies and chemotherapeutic agents for targeted GBM chemo-immunotherapy while reducing systemic side effects. Several flexible approaches have been developed to co-deliver different drugs, of which co-encapsulation of all drugs into one system allows uniform drug distribution in vivo and accumulate different drugs in the tumor tissue with the appropriate ratios[30]. However, the co-encapsulation approach for antibody-based GBM chemo-immunotherapy has substantial challenges with complex preparation process, as it is necessary to consider the loading manner, bioactivity of the antibody and the efficiency of the drug release in NanoDDS[31,32].

In this work, we integrate antibodies and chemotherapeutic agents into a single micelle (angiopep2-aPD-L1@PTX nano-micelle, A2-APM) by crosslinking anti-PD-L1 antibodies (aPD-L1) and electrostatically binding paclitaxel (PTX) onto the amino groups of polyethylene glycol-poly-L-lysine (PEG-PLL) for effective GBM chemo-immunotherapy (Fig. 1a). The targeted angiopep-2 (A2) peptide is decorated on the surface of micelles to allow the BTB penetration[33]. Moreover, the reductive microenvironment of GBM[34,35] triggers the release of aPD-L1 from the micelles via the cross-linker chain breakage without altering their structure and function, which further increases the GBM selectivity of ICB. At the same time, the dissociation of the micelles accelerates the release of PTX in the tumor microenvironment. The rapidly released PTX not only exhibits excellent inhibitory properties to GBM cells, but also sensitizes tumors to the PD-1/PD-L1 blockade by inducing ICD to facilitate ICB therapy[36,37]. This study aims to provide an applicable chemo-immunotherapeutic approach to address the limitation of ICB therapy in brain malignancies in a bio-safe and effective manner.

## Results

### High expression of PD-L1 from histological sections in glioma patients and glioma-bearing mice

Given reports that the overexpressed PD-L1 is a major prognostic biomarker and predominant therapeutic target for GBM[7,8], we investigated PD-L1 expression from The Cancer Genome Atlas (TCGA). PD-L1 expression is high in high-grade glioma (HGG, GBM, Supplementary Fig. 1a). A Kaplan-Meier survival curve investigation of PD-L1 expression with mortality indicated that glioma patients with higher PD-L1 expression had poorer survival outcomes (Supplementary Fig. 1b). In addition, we collected tumor samples from human glioma patients, which was classified by WHO classification, including low-grade glioma (LGG) and HGG. We evaluated the integral optical density (IOD) of PD-L1 immunofluorescence (IF) in the samples to quantify the PD-L1 expression (Supplementary Fig. 1c, d and Supplementary Table 1). The PD-L1 was considerably higher in the HGG group than in the LGG group ($P < 0.05$), which follows the same trend of TCGA (Supplementary Fig. 1e, including 144 cases of HGG and 500 cases of LGG, and the PD-L1 expression was significantly higher ($P < 0.001$) in the HGG group than in the LGG group). These results confirmed that a high PD-L1 expression is apparently an adverse prognostic factor for glioma.

Then, the PD-L1 expression of GBM patient-derived xenograft (PDX) cell lines (G7 and WL1), human glioma cell lines (U87 and U251), and mouse glioma cell lines (GL261 and G422) were evaluated. These cell lines were inoculated into the brains of mice (BALB/c nu for G7, WL1, U87, and U251 cells, C57BL/6 for GL261 cells, and Kunming for G422 cells, G7 and WL1 tumor bearing-mice were PDX mice). To evaluate the IOD of PD-L1 IF in histological sections from G7, WL1, U87, U251, GL261 and G422 tumor mice, we found that the results of GL261 and G422 tumor mice were similar to the PDX mice ($P > 0.05$) (Supplementary Fig. 1f). Subsequently, the expression levels of PD-L1 on the surface of these cell lines were assessed by flow cytometry and showed that all these cell lines had significant PD-L1 expression (Supplementary Fig. 1g–i). These findings demonstrate that PDX cell lines, human cell lines, and murine cell lines all bind to aPD-L1, providing a justification for using an anti-PD-L1 therapy in GBM. Based on this, we designed angiopep2-aPD-L1@PTX nano-micelles (A2-APM), which integrated aPD-L1 and PTXs into individual micelles.

### Characterization of redox-responsive A2-APM

To construct A2-APM, firstly we designed redox-sensitive chains by covalent conjugation among antibodies, maleimide-polyethylene glycol-poly-L-lysine (Mal-PEG-PLL) and 2-[(2-[(4-succinimidoxy) carbonyl] oxyethyl) disulfanyl] ethyl 4-succinimidyl carbonate [noted as SC-$(CH_2)_2$-S-S-$(CH_2)_2$-SC] (Supplementary Fig. 2). Then, Mal-PEG-PLL encapsulated negatively charged PTX to generate PTX co-loaded micelles via electrostatic interaction, followed by installation of the maleimide moiety to the angiopep-2 peptide via click reaction. The micelles were formed at a molar ratio of 5:1:12 of Mal-PEG-PLL: antibody: cross-linker. To ensure the efficient induction of ICD, PTX with 2-fold of the antibody mass was put in. Dynamic light scatterings (DLS) analysis revealed that the synthesized APM was homogeneous with diameters of about 58.8 nm (Fig. 1b and c, Supplementary Fig. 3), and a positive zeta potential of $21.64 \pm 0.92$ mV (Supplementary Fig. 4). The encapsulation efficiency and loading content of IgG antibody were $82.31 \pm 4.30\%$ and $22.43 \pm 1.17\%$, respectively, while those of PTX were $87.63 \pm 2.18\%$ and $47.76 \pm 1.19\%$ (Supplementary Table 2). The stability test indicated that the micelles could maintain their original size in different environments (pH = 7.4 PBS, DMEM medium, 10% FBS in PBS, pH = 6.8 PBS, and pH = 5.5 PBS) over a week (Supplementary Fig. 5). In addition, the antibody or PTX-loaded micelles (denoted as AM or PM) with a similar formula were prepared and used as the control in this study (Supplementary Figs. 6, 7).

Since the recovery of IgG antibody to its original structure upon glutathione (GSH, 2 mM, pH = 7.4) is of great importance for antibody delivery, the redox-responsive in vitro drug release behavior of APM was further evaluated. The IgG release triggered by disulfide bond cleavage was first verified by gel electrophoresis assay [SDS-polyacrylamide gel electrophoresis (PAGE)]. As shown in Supplementary Fig. 8, a clear electrophoretic band of free IgG (~150 kDa) appeared for

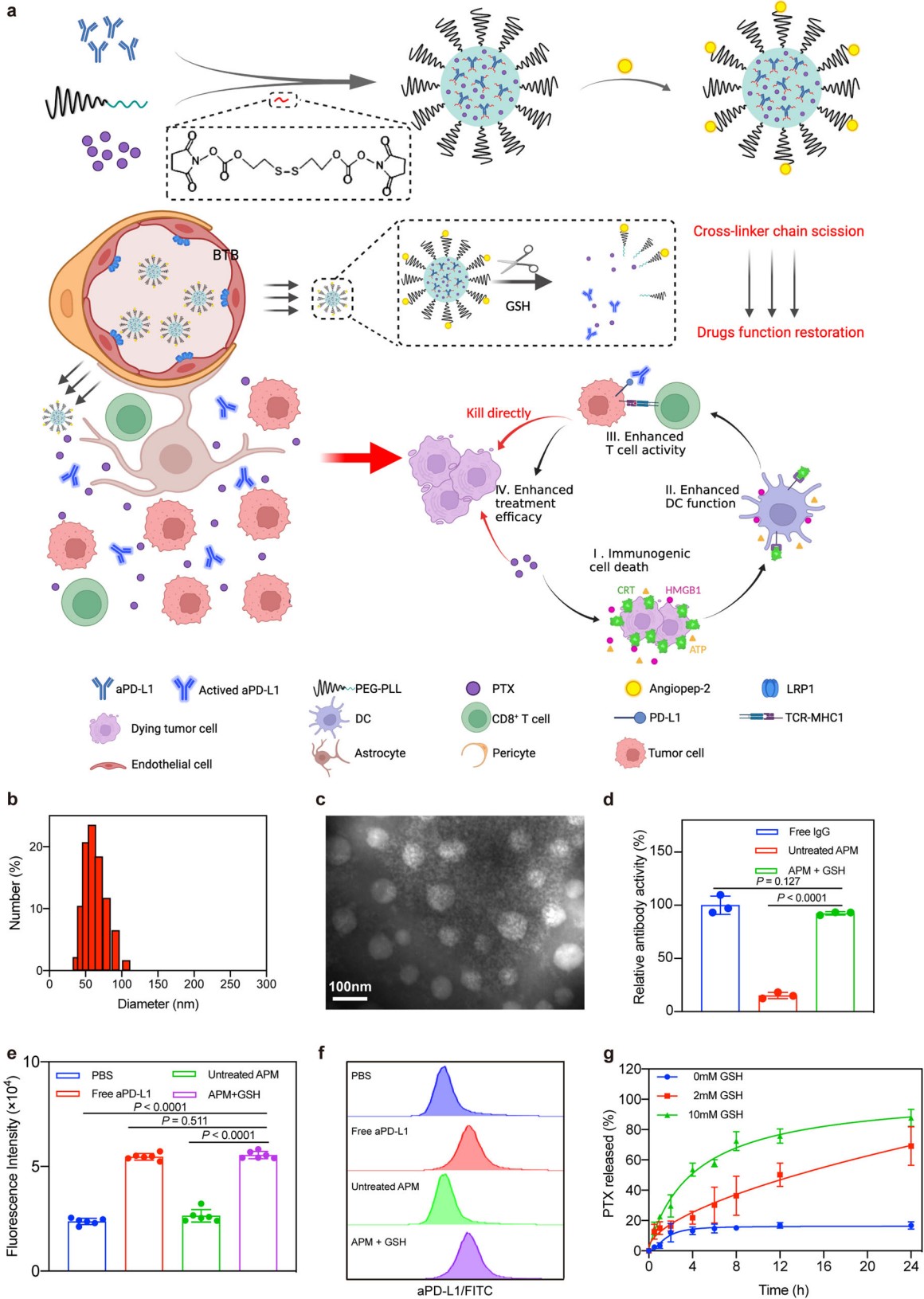

APM after treatment with 2 mM GSH, while no electrophoretic band of free IgG was found for APM in the absence of GSH (untreated APM), indicating that IgG was released from the micelles and migrated under the electric field after the disulfide bond was cleaved with 2 mM GSH. Moreover, to measure the binding capacity of the released IgG to its ligand in APM, an enzyme-linked immunosorbent assay (ELISA) was used. The results (Fig. 1d and Supplementary Fig. 9) showed that the untreated APM had negligible binding potential because it barely released IgG in a physiological environment (pH = 7.4). However, a considerably increased binding amount of IgG in APM was observed with GSH concentration-mediated and time-dependent features, indicating that the cross-linker was cleaved and the natural IgG was

**Fig. 1 | Characterizations of redox-responsive A2-APM. a** Schematic illustration of A2-APM structure and strategy for enhanced ICB therapy against GBM. A redox-responsive micelle was developed which covalently linked massive aPD-L1 via recoverable cross-linkers. PTX was co-encapsulated into the micelle followed by conjugating the angiopep-2 peptide that actively transported the antibody and PTX across the BTB to elicit activatable immune response against GBM through redox-responsive cross-linker chain-breaking, and amplified ICB efficacy by PTX-induced ICD effect to enhance anti-GBM therapy. **b** Dynamic light scatterings (DLS) analysis of the mean particle size distribution of APM ($n = 3$ biologically independent samples). **c** Transmission electron microscopy (TEM) image of APM. Scale bar = 100 nm. ($n = 3$ biologically independent samples). **d** Relative antibody activity of free IgG and APM with or without GSH treatment (nsource = 3 biologically independent samples) **e, f** Quantification and representative flow cytometry histogram of aPD-L1 binding on GL261 cells incubated with free aPD-L1 and APM with or without GSH treatment ($n = 6$ biologically independent samples). **g** Profiles of PTX released by APM in the presence of different concentrations of GSH ($n = 3$ biologically independent samples). All statistics are expressed as mean$n \pm$ standard deviation (SD). BioRender.com was used to create (a). Statistical significance was calculated by one-way ANOVA with Fisher's LSD test. Source data are provided as a Source Data file.

released to activate its binding affinity. Further, we assessed the binding capacity of antibodies released by APM to PD-L1 on tumor cell membrane. Free aPD-L1 was incubated with GL261 cells for 1 h as a positive control. The fluorescence shift and the binding affinity of aPD-L1 in APM for GL261 cells were evaluated by flow cytometry (Fig. 1e and f). The sharp increase in fluorescence shift of aPD-L1 on GL261 cell surface after incubation with GSH pre-activated APM was noted, which was similar to the positive control, and the bound aPD-L1 was insignificant when co-incubated with non-GSH-activated APM. The results clearly suggested that the binding affinity of aPD-L1 loaded in APM to PD-L1 was efficiently restored in the reductive environment.

Next, the release profiles of PTX from APM were investigated to ensure that both drugs were released at similar rates in the reductive environment. Under physiological conditions, as seen in Fig. 1g, drug release rate was somewhat modest, with only 20% of PTX released from APM within 24 h. The release rate of PTX under 2 mM GSH condition, however, was substantially expedited with 69.17% of PTX released within 24 h. The rapid drug release behavior of APM at reductive environment was the result of micelle disintegration. Based on these findings, the micelles can effectively respond to GSH to release aPD-L1 and PTX, and restore their activities while minimizing drug leakage under physiological conditions.

### Cytotoxicity and ICD induction ability of APM in GL261 cells
The anti-proliferative effect of APM on GL261 cells was determined by cell counting kit-8 assay (CCK-8). As shown in Fig. 2a, blank micelles assembled from Mal-PEG-PLL and SC-$(CH_2)_2$-S-S-$(CH_2)_2$-SC showed negligible cytotoxicity at any concentration. Similar result could be found for AM. However, the PM or APM treated groups exhibited high cytotoxicity which was positively correlated with the concentration of loaded PTX, demonstrating the potent anti-GBM effect of PTX in vitro. In addition, a live/dead assay was used to further verify the results. After treatment with free PTX, PM, and APM, most of the floating dead cells were lost when washed with PBS (pH = 7.4), showing a low cell density of GL261 cells, a rounded cell morphology, and a large population of red-stained dead cells (Supplementary Fig. 10). Later, apoptosis of GL261 cells induced by different treatments was detected by flow cytometry. As displayed in Fig. 2b and Supplementary Fig. 11, GL261 cells supplemented with APM, PM, and free PTX showed more severe early apoptosis and late apoptosis than those treated with blank micelles and AM, which was consistent with the results of the CCK-8 assay. All these results indicated that the micelles containing PTX exhibited efficacious anti-GBM efficacy in vitro, and blank micelles were almost non-toxic.

As PTX is known to induce ICD of cancer cells[36,38], the in vitro ICD effect induced by PTX in APM was further analyzed. The levels of cell surface calreticulin (CRT) expression, high mobility group box 1 (HMGB1) and adenosine triphosphate (ATP) secretion, which are signal molecules known as DAMPs associated with ICD effects, were tested after treating GL261 cells with different micelles. As shown in Fig. 2c, d, blank micelles or AM induced less CRT exposure, while free PTX markedly increased the exposure of CRT on tumor cells. The exposure level of CRT increased when GL261 cells were incubated with PTX-

containing micelles, which was similar to the free PTX. The levels of HMGB1 and ATP secretion in the cell supernatant (Fig. 2e, f) also showed similar tendency, and the intracellular HMGB1 in PTX-containing micelles groups was lower than that in the other groups (Supplementary Fig. 12).

The released DAMPs serve as adjuvant stimuli for recognition and processing by antigen-presenting cells (APCs) such as DCs[39]. Thus, we used a binary co-incubation system to validate the effect of ICD on DC maturation by exposing GL261 cells to blank micelles, free PTX, AM, PM, or APM for 24 h, respectively, and then incubating them with immature DC2.4 cells for another 24 h. Analysis of biomarkers of mature DCs (CD11c, MHC II, CD80, and CD86) by flow cytometry showed that when GL261 cells were pretreated with PM or APM, the maturation of DCs was greatly promoted (Fig. 2g, h). The expression levels of CD11c, MHC II, CD80, and CD86 in APM treated group were similar to those of the free PTX treatment group, but significantly higher than those of the blank micelles and AM treatment groups ($P < 0.05$).

To investigate the effect of APM-treated tumor cells on macrophages or DCs and their effect on T cells, bone marrow-derived macrophages (BMDMs), bone marrow-derived dendritic cells (BMDCs) and T cells were extracted from C57 mice. Then BMDMs and BMDCs were treated with tumor cells or A2-APM treated tumor cells (Supplementary Fig. 13a and Supplementary Fig. 14a) before adding T cells into the culture dish. In the A2-APM treated tumor cell group, CD206, the marker of inhibition, significantly decreased, and CD86, the marker of anti-tumor, increased (Supplementary Fig. 13b). Meanwhile, the BMDCs were matured with the rise of CD80 and CD86 expression (Supplementary Fig. 14b). Additionally, T cells were observed with the IFN-γ expression level elevated in CD3+CD8+ T cells. These findings implied that APM can elicit sufficient ICD for the immune system to recognize tumor cells, allowing DCs and macrophages to activate a potent antigen-presenting function and establishing the groundwork for T cell stimulation.

### ICD induction ability of APM in glioma stem-like cells and human cell lines
Glioma stem-like cells (GSCs) are the most resistant cells in glioma, therefore, GSCs were obtained and tested to further confirm the above results. Analysis of biomarkers of GSCs (CD133, CD44, SOX2, and NESTIN) by flow cytometry and colony formation assay were performed to identify the GSCs (Supplementary Fig. 15a–d). Then, the cytotoxicity of APM were determined by CCK-8 assay, and the PM or APM treated groups showed a positive correlation of cytotoxicity with the concentration of loaded PTX (Fig. 2i). However, due to the drug resistance nature of GSCs, in vitro cytotoxicity of PTX was reduced in GSC cells compared to GL261 cells. To further explore the ICD induction ability of APM for GSCs, the level of CRT in CD133+ GSCs was measured by flow cytometry, which suggested that the PTX-contained micelles were potent in inducing the ICD effect of GSCs (Fig. 2j and Supplementary Fig. 15e, f).

In addition, to demonstrate that PTX can induce ICD effect in murine GBM cells in addition to human GBM cells, we also examined

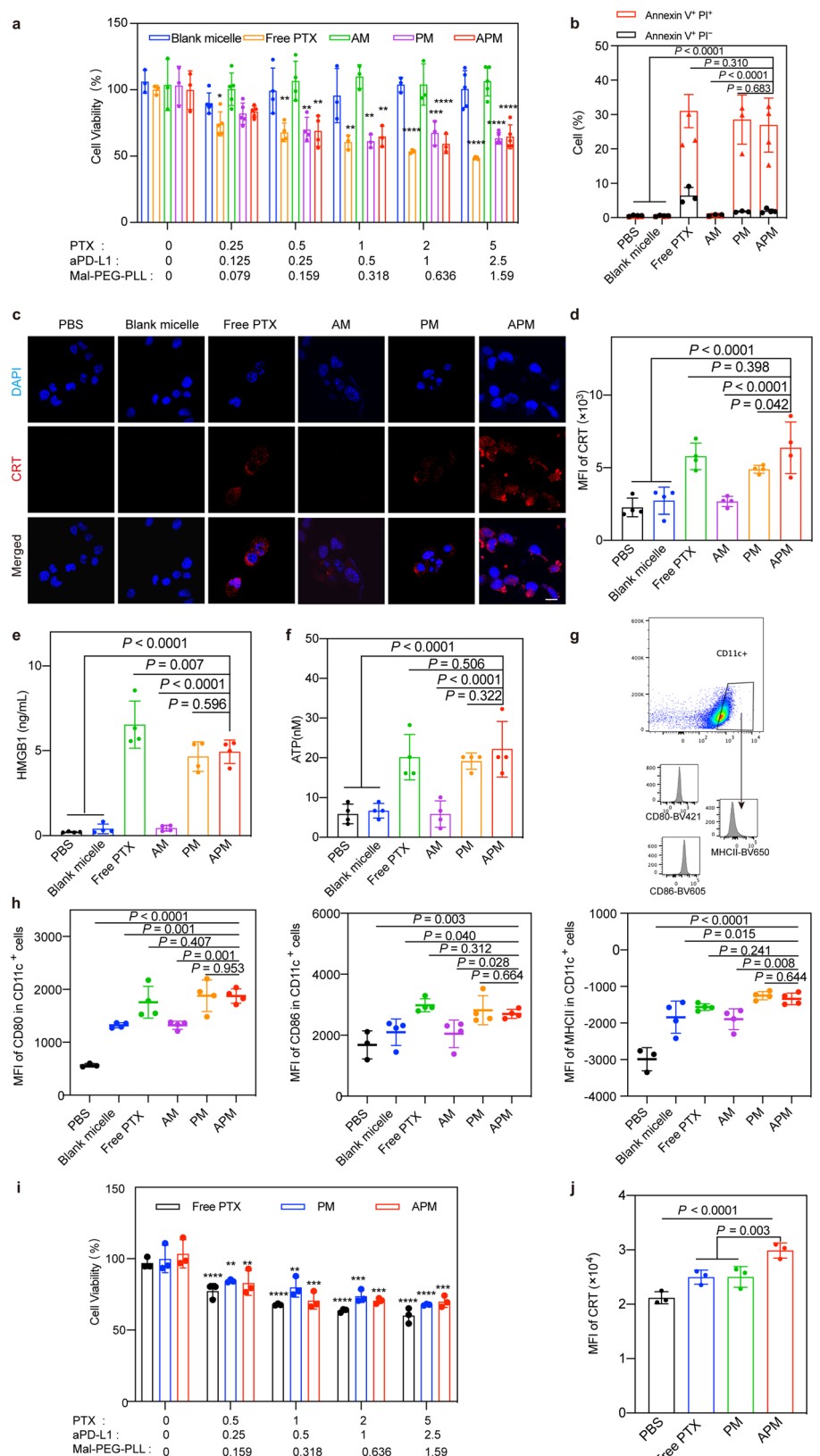

the ability to induce CRT expression in G7, WL1, U87, U251, GL261, and G422 cells under different concentrations of PTX conditions. The results showed that 2 μg/mL PTX induced significant CRT expression in mouse cell lines, and in all types of cells, 6 μg/mL PTX induced noteworthy CRT expression ($P < 0.05$) (Supplementary Fig. 16).

## Accumulation and biodistribution of A2-APM in tumor models

It is crucial for drugs to cross the BTB and selectively accumulate at the tumor site in GBM. To assess the BTB penetration ability of A2-APM, we first cultured bEnd.3 monolayer cells into the upper chamber and GL261 cells into the lower chamber as an in vitro BTB model (Fig. 3a)[40]. aPD-L1 was labeled with FITC dye to prepare fluorescent aPD-L1, APM

**Fig. 2 | Cytotoxicity and ICD induction in GL261 cells and GSCs. a** Cytotoxicity results of GL261 cells after receiving different treatments (concentration unit: µg/mL, incubation time: 24 h. PTX concentration = 0, 1, and 2 µg/mL, $n = 3$ biologically independent samples. PTX concentration = 0.5 µg/mL, $n = 4$ biologically independent samples. PTX concentration = 0.25 and 5 µg/mL, $n = 5$ biologically independent samples. **b** Apoptosis results of GL261 cells after receiving different treatments (PTX concentration = 2 µg/mL, incubation time: 24 h. PBS, blank micelle and APM groups, $n = 4$ biologically independent samples. Free PTX, AM and PM groups, $n = 3$ biologically independent samples) **c, d** Exposure and quantification of CRT exposure on the surface of GL261 cells after receiving different treatments (scale bar = 20 µm) ($n = 4$ biologically independent samples). **e, f** The levels of released HMGB1 and ATP after receiving different treatments ($n = 4$ biologically

independent samples). **g** Gating strategy to assess the levels of MHCII, CD80, and CD86 in DC2.4 cells gated on Live$^+$ CD11c$^+$ cells. **h** Promotion of DC2.4 maturation after co-incubation with pretreated GL261 cells (PBS group, $n = 3$ biologically independent samples, other groups, $n = 4$ biologically independent samples). **i** Cytotoxicity results of GSCs after receiving different treatments for 24 h (concentration unit: µg/mL, $n = 3$ biologically independent samples). **j** Qualification of the exposure of CRT on GSCs after receiving different treatments for 24 h, $n = 3$ biologically independent samples. All statistics are expressed as mean$n \pm$ SD, *$P < 0.05$, **$P < 0.01$, ***$P < 0.001$ and ****$P < 0.0001$. Statistical significance was calculated by one-way ANOVA with Fisher's LSD test. Source data are provided as a Source Data file.

and A2-APM, the formulations of which were then added to the upper chamber. After 4 h of co-culture, the FITC-aPD-L1 quantity of A2-APM at the filtrate side dramatically increased to 49.43% (Fig. 3b). By contrast, all the formulations without angiopep-2 peptide showed low permeability through the monolayer, the quantity of FITC-aPD-L1 delivered by free aPD-L1 and APM in the lower chamber were 16.20% and 25.64%, respectively. These results indicated that the angiopep-2 peptide-modified APM could efficiently migrate across the bEnd.3 cell monolayers, increasing the penetration efficiency of nano-micelles in the in vitro BTB model.

Then, the in vivo pharmacokinetics of free aPD-L1 and A2-APM were examined to investigate their transportability in the vasculature (Supplementary Fig. 17). A2-APM ($t_{1/2} = 33.05$ h) had a longer half-life in the blood than free aPD-L1 ($t_{1/2} = 23.86$ h), which is beneficial in prolonging circulation. The result represented that aPD-L1 encapsulation in micelles effectively decreased the plasma clearance of aPD-L1, mostly due to the presence of PEG-shell. In vivo imaging system (IVIS) was further used to determine the accumulation potential of A2-APM in GBM model. aPD-L1 was labeled with Cy7.5 dye to prepare fluorescent micelles. Then, free aPD-L1, APM, and A2-APM were intravenously (i.v.) injected into mice at equivalent aPD-L1 doses, and the drug accumulation in the brain was observed in real-time. As displayed in Fig. 3c-f and Supplementary Fig. 18a–d, A2-APM revealed obvious tumor accumulation after injection and significantly increased fluorescence intensity at 72 h in both GL261 and G422 models. As for the free drug or APM-treated mice, however, poor accumulation with low fluorescent signals in the tumors were observed within 72 h. Moreover, the major organs were collected for ex vivo imaging at 90 min post-injection (Fig. 3g, h). Those injected with A2-APM displayed greatly enhanced Cy7.5 signals in the tumor, showing 13.04-fold increase over free aPD-L1 and 2.93-fold increase over APM in GL261 tumor mice (Fig. 3e, f), 6.12-fold increase over free aPD-L1 and 1.79-fold increase over APM in G422 model (Supplementary Fig. 18c, d). Meanwhile, we detected that the concentration of PTX accumulated in the brain was $6.05 \pm 2.19$ µg/mL, at 90 min after A2-APM injection. These findings suggested that angiopep-2 peptide facilitates APM to actively transport drugs across the BTB and greatly increases their accumulation and retention in the brain tissues for an extended period.

In addition, the cellular biodistribution of A2-APM within tumors and peripheral organs, including blood cells, would be very informative[41,42]. Histological analysis showed that the tumor site specifically took up Alexa647 dye labeled A2-APM (A2-APM-Alexa647), and nano-micelles incorporation was detected in Gl261-GFP$^+$ tumor cells (Fig. 3i). Flow cytometry results revealed a high proportion of monocyte-derived myeloid cells that conjunct with A2-APM in tumor hemisphere, whereas the tumor microglia, and the monocyte-derived myeloid cells and macrophage in liver, blood, lung, kidneys, and heart revealed less nano-micelle uptake (Fig. 3j, k and Supplementary Fig. 19).

### In vivo antitumor efficacy of A2-APM
To assess the efficacy of A2-APM against orthotopic brain tumors, the GL261 and G422 cell lines were injected into the right brain of 6–8-

week-old C57 and 4-5-week-old Kunming mice. The antitumor efficacy of A2-APM was examined using the treatment regimen displayed in Fig. 4a and Supplementary Fig. 20a. On day 7 and day 14 after implantation, GBM tumor-bearing mice received intravenous administration of saline, free aPD-L1, free PTX, A2-AM, A2-PM, APM, and A2-APM (1.5 mg/kg based on aPD-L1 concentration, or 3 mg/kg based on PTX concentration), respectively. The development of brain tumors was traced by magnetic resonance (MR) every 5 days, and the changes in tumor volume were shown in Fig. 4b, c and Supplementary Fig. 20b–d. In contrast to the free aPD-L1, free PTX, A2-AM, A2-PM, and APM treatment groups, mice treated with A2-APM showed significant tumor regression after day 7. On day 22, it was found that 60% A2-APM treated GL261 tumor-bearing mice exhibited a strong antitumor response based on size reduction using MR imaging (Supplementary Fig. 21), confirming the potential tumor suppressive effect by A2-APM. Importantly, free aPD-L1, free PTX, and non-targeted APM failed to elicit an effective antitumor response in mice bearing GL261 tumors due to inadequate brain accumulation. Although the A2-AM and A2-PM increased the tumor accumulation of drugs, tumor suppression was still negligible, probably due to the low effectiveness of a single therapeutic agent. It is worth noting that as compared to the other groups, the survival rate of mice in the A2-APM group was considerably higher (Fig. 4d, e), verifying that the enhanced antitumor efficacy was achieved via the chemo-immunotherapy. A pathological study was also performed using tumor samples collected from various groups. Histological examination of hematoxylin and eosin (H&E) staining of tumor tissues (Supplementary Fig. 22) definitely proved that A2-APM caused a high amount of cell apoptosis and necrosis compared to the other groups. To assess the additive effect of co-encapsulated PTX and aPD-L1 delivered into the brain, the antitumor efficacy of free aPD-L1 and free PTX and A2-APM were examined in GL261 and G422 tumor mice (Supplementary Fig. 23). The survival in the A2-APM treatment group was significantly improved compared to the free aPD-L1 and free PTX groups ($P < 0.05$).

As surgical resection is the first clinical intervention for GBM patients, the effect of A2-APM on resected GL261 tumor model was explored. On day 14 after the implantation of GL261 cells, the visible tumor was surgically removed under microscope to establish resected GBM model (Supplementary Fig. 24a, b). The post-operative mice underwent surgery were randomly grouped into each treatment. Saline, free aPD-L1, free PTX, APM, and A2-APM were then injected intravenously into the resection models. As shown in Supplementary Fig. 24c–e, A2-APM therapy had a greater ability to extend the median survival of post-operative mice ($P < 0.05$) and decelerate the recurrence of GBM. On days 16, 21, and 28 in the saline group, tumor resection cavities and residual tumor lesions were observed using H&E staining (Supplementary Fig. 24f). On day 21 and day 28 post-implantation, the tumors presented a prominent infiltration pattern of invasive tumor cells, along with many tumor cell islands, both around the resection cavity and inside the surrounding normal brain parenchyma. On day 28, the histological analysis of the brain of mice treated with A2-APM showed no signs of

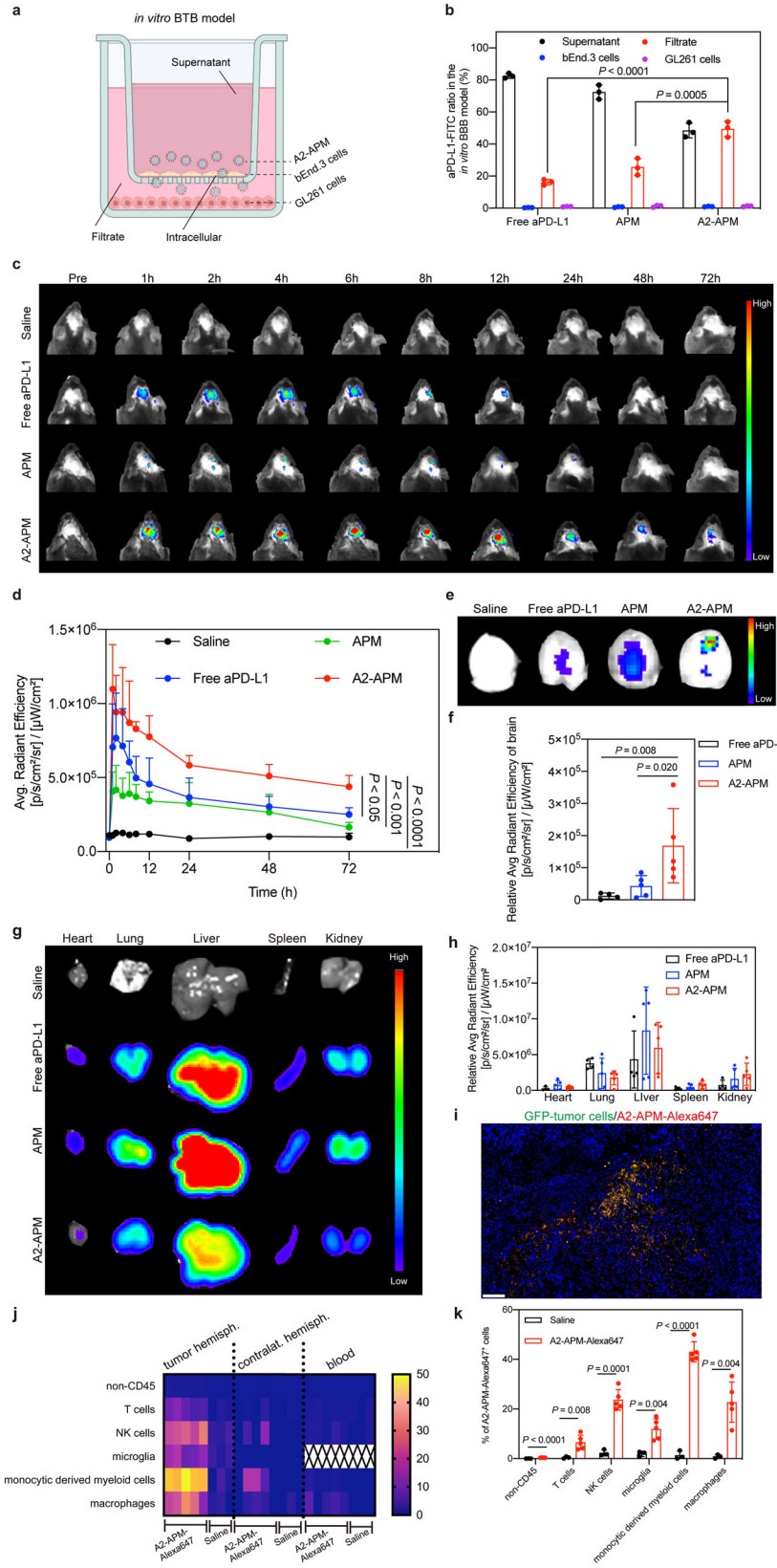

tumor cells infiltrating the normal brain parenchyma around the tumor resection cavity (Supplementary Fig. 24g). In contrast, all mice subjected to the other treatments had many invasive tumor cells and islands of tumor cells in their brains. These results implied that A2-APM not only inhibits primary tumors, but also shows suppression of recurrence of residual tumor lesions.

## Biosafety of A2-APM

No obvious body weight loss was detected in the mice of each group during 22 days of observation period, indicating low systemic toxicity (Supplementary Fig. 25). To determine whether the prolonged blood circulation time might increase their distribution in other organs, saline and A2-APM (Cy7.5-labeled aPD-L1, aPD-L1 equivalent dose 1.5 mg/

**Fig. 3 | A2-APM effectively crosses BTB. a** Schematic illustration of the in vitro BTB model using a transwell system to evaluate the penetration capability of A2-APM across the endothelial monolayer. Quantities of aPD-L1 were determined in the supernatant, bEnd.3 cells, filtrate, and GL261 cells. **b** Quantification of the aPD-L1 distribution in the chamber after incubation with free aPD-L1, APM and A2-APM for 4 h (n = 3 biologically independent samples). **c, d** Fluorescence imaging and signal intensities at tumor sites at 1, 2, 4, 6, 8, 12, 24, 48 and 72 h after i.v. injection of GL261 tumor bearing-mice with saline, aPD-L1, APM and A2-APM (n = 5). **e, f** Representative ex vivo fluorescence images and semiquantitative biodistribution of Cy7.5-aPD-L1 in brains collected from mice injected with saline, free aPD-L1, APM, and A2-APM at 90 min post i.v. injection (saline and free aPD-L1 treated mice, n = 4; APM and A2-APM treated mice, n = 5). **g, h** Representative ex vivo fluorescence images and semiquantitat ion of biodistribution of Cy7.5-aPD-L1 in main

organs collected from mice injected with saline, free aPD-L1, APM, and A2-APM at 90 min post i.v. injection (saline and free aPD-L1, n = 4; APM and A2-APM, n = 5). **i** Representative histological images showing A2-APM accumulation in the TME of GL261 glioma. n = 3. Scale bar = 100 μm. **j, k** Biodistribution analysis of A2-APM-Alexa647 at 90 min after a single intravenous dose as assessed by flow cytometry of T cells (CD45$^+$CD3$^+$), NK cells (CD45$^+$NK1.1$^+$), microglia (CD45$^+$CD11b$^{intermediate}$), monocyte-derived myeloid cells (CD45$^+$CD11b$^{high}$Ly6C$^+$) and macrophages (CD45$^+$CD11b$^{high}$ F4/80$^+$) in the tumor-bearing hemisphere, contralateral hemisphere and blood. A2-APM-Alexa647 treated mice, n = 5 and saline-treated mice, n = 3. All statistics are expressed as mean ± SD. Statistical significance was calculated by one-way ANOVA with Fisher's LSD test in (**b**), (**d**) and (**f**), and two-sided t-test in (**k**). BioRender.com was used to create (**a**). Source data are provided as a Source Data file.

kg) were intravenously given to C57 mice bearing GL261 and G422 tumors. The mice were sacrificed at 90 min, 24 h, and 96 h post-injection to extract the heart, liver, spleen, lung, kidneys, and brain (Supplementary Fig. 26). The semiquantitative results revealed that A2-APM was excreted from the body over time and did not accumulate in healthy organs. 30 days after i.v. injection of different samples into healthy C57 mice, histological studies of tissue sections from major organs were investigated, and hematological indices were tested to value the side effects of treatments (Supplementary Fig. 27). Plasma concentrations of aspartate aminotransferase (AST), which is associated with liver, heart, or kidney damage, alanine aminotransferase (ALT), which is a marker of liver damage, and blood urea nitrogen (BUN) and creatinine (CR), which are markers of nephrotoxicity, showed no significant differences in any groups (Supplementary Fig. 28). All these results indicated the acceptable biocompatibility of micelles.

To investigate whether the addition of A2 peptide could cause chemotherapeutic agents-induced cerebrovascular abnormalities, full-field laser perfusion imager was used to determine the change of cerebral blood flow. Saline, free PTX, APM, and A2-APM were i.v. injected into mice at equivalent PTX doses, and the change in the cerebral blood flow was observed in real-time. As shown in Supplementary Fig. 29a, b at 4 h after injection, each group had a transient increase in blood flow rate, but there were no appreciable variations in blood flow changes across groups at any of the time points. Additionally, tumor samples from mice in various treatment groups were examined using blood vessel immunofluorescence, which revealed intact vessel walls in the other groups compared to the saline group at all time points (Supplementary Fig. 29c, d). Meanwhile, when the mouse brains from various treatment groups were taken at various time points, wet and dry weights were measured, and the brain water content was compared across the groups, there was no evidence that the micelles might cause brain edema (Supplementary Fig. 30). These results demonstrated that the addition of A2 peptide does not lead to vascular abnormalities, probably due to the high specificity of A2 peptide, which enables micelles to penetrate the BBB and locate the tumor without interacting with blood vessels.

To evaluate whether A2-APM can effectively mitigate lymphodepletion, the drugs were injected intravenously to GL261 tumor-bearing mice. Saline, free PTX, and A2-APM (5 mg/kg PTX) were administered on days 14, 16, and 18. Blood, lymph nodes and spleen were collected on day 21 for flow cytometry analysis (Supplementary Fig. 31). There was no significant difference in the proportion of CD45$^+$ cells between the saline and A2-APM groups in blood, lymph nodes, and spleen. In contrast, the free PTX group showed a significant decrease in the proportion of CD45$^+$ cells in the blood and in the proportion of CD3$^+$CD8$^+$ T cells in the spleen. Furthermore, we found that the proportion of inhibitory T cells (PD-1$^+$ or TIM-3$^+$ CD3$^+$CD8$^+$ T cells) in the A2-APM group was significantly lower than that in the free PTX group. These results suggested that A2-APM did not lead to lymphatic depletion and did not increase the proportion of exhausted T cells.

## A2-APM treatment activates effector T cells

To further examine in vivo immunological response triggered by A2-APM, the infiltration of CD8$^+$ T lymphocytes into the tumor region was analyzed after three days with two doses of A2-APM. Immunofluorescence slices of GL261 tumors without therapy exhibited minimal CD8$^+$ T cell infiltration (Supplementary Fig. 32 and 33a, b). Nonetheless, the A2-APM treated group had a notably increased number of total intra-tumoral CD3$^+$ T cells, and most importantly, the proliferation of cytotoxic CD8$^+$ T cells was 4.63- and 1.98-fold higher than the untreated and free antibody groups, respectively (Supplementary Fig. 33c). The APM-treated group had more Foxp3$^+$ T cells, which are known as regulatory T cells (Tregs) to hinder the antitumor effectiveness of cytotoxic T cells (CTLs), while Treg levels in the other groups were roughly equivalent to those of mice in the saline group. The CD8$^+$/Foxp3$^+$ T cell ratio, which is positively related to the survival outcomes in GBM mice, was significantly high with the A2-APM treatment group being 5.85- and 1.77-fold greater than the saline and free aPD-L1 groups, respectively. Moreover, CD8$^+$ T cells in saline, free aPD-L1, free PTX, A2-AM, A2-PM, and APM treated groups were found mostly near the perimeter of the tumor tissue, but the A2-APM treated group showed more CTLs penetrating the interior of the tumor tissue, making it easier to destroy deep tumor cells. Flow cytometry analysis (Fig. 5a, b and Supplementary Fig. 33d) further confirmed these results, with the A2-APM group having remarkably 3.17-fold more CD8$^+$ T cells and a 6.60-fold higher CD8/Treg cell ratio than those of the free aPD-L1-treated group. Further, we examined the key markers on CD8$^+$ T cells and found that in the A2-APM- treated group, the antitumor markers (IFNγ; Granzyme B, GrzB) were upregulated, whereas the inhibitory markers (PD-1, TIM-3) were decreased (Fig. 5c, Supplementary Fig. 34).

## A2-APM treatment efficacy is dependent on CD8$^+$ T cells

Since our results above demonstrated that T cells play an important role in A2-APM anti-GBM efficacy, next we investigated which subtype of T cells (CD4 or CD8) plays a dominant role in A2-APM antitumor therapy. Notably, based on the increase in tumor volume over four weeks, depletion of CD8$^+$ T cells significantly abrogated the therapeutic effect of A2-APM (P = 0.042), whereas depletion of CD4$^+$ T cells did not (P = 0.140) (Fig. 5d, Supplementary Fig. 35). Therefore, we identified a major role for CD8$^+$ T cells in mediating tumor regression of A2-APM therapy.

## A2-APM treatment promotes DC maturation

To clarify the promoting factors of CTL activation, the presence of DCs in GBM tumors was also examined 3 days after treatment with A2-APM. According to immunofluorescence results (Supplementary Fig. 36), the level of CD11c$^+$ DCs was enhanced by A2-APM, which was equivalent to the A2-PM treatment group, indicating that PTX-containing targeted micelles could increase the number of active CD11c$^+$ DCs. Mature DCs (mDCs) are a more notable sign for improving antitumor activity through effective tumor antigen presentation.

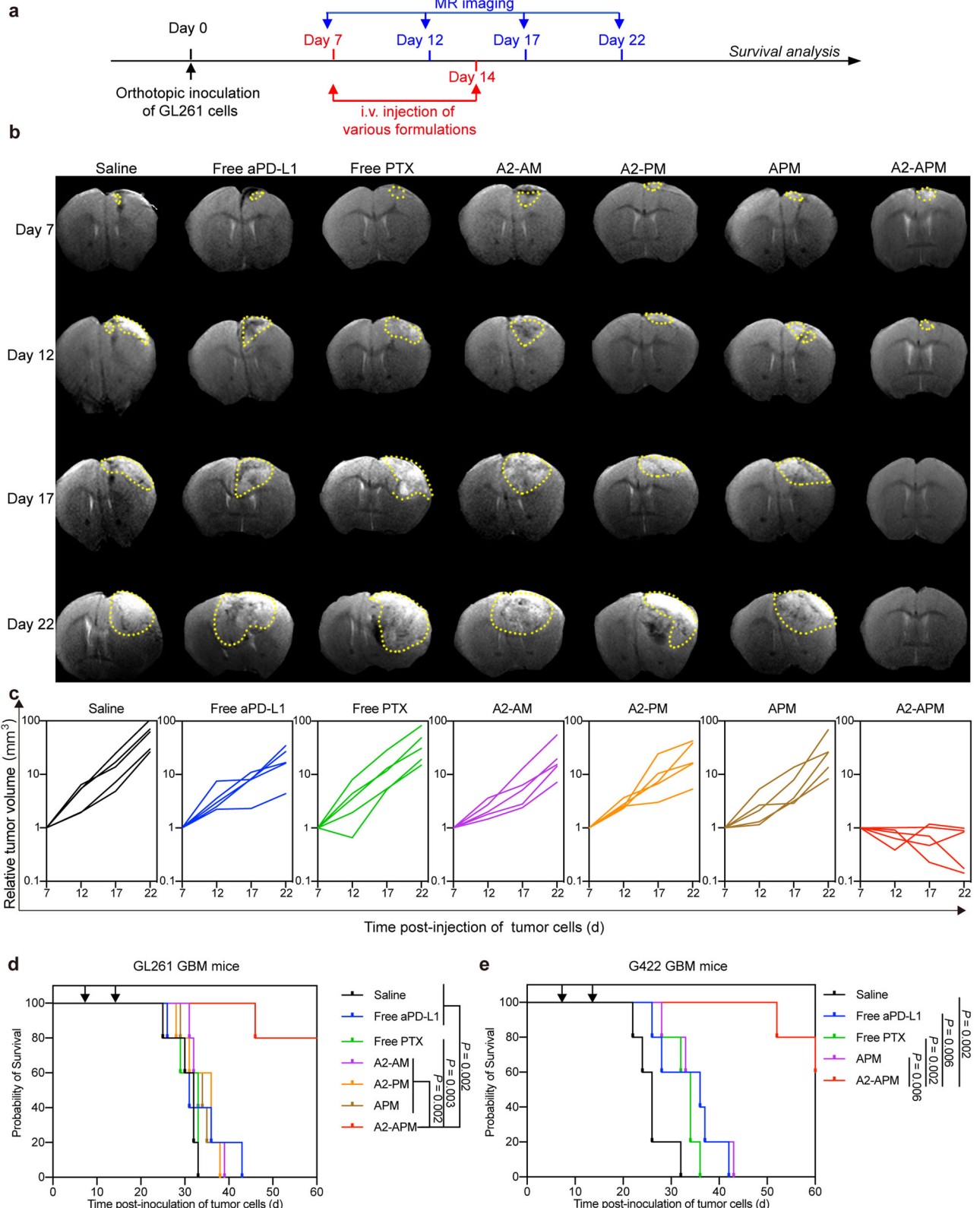

**Fig. 4 | A2-APM therapy reduces glioma growth and prolongs survival.**
**a** Schematic of the treatment regimen. **b-c** Representative MR images and quantified signal intensity of GL261 tumor-bearing mice treated with saline, free aPD-L1, free PTX, A2-AM, A2-PM, APM, and A2-APM. The yellow dashed lines indicate the tumor area. **d** Survival curves for the treated and control GL261 tumor mice. **e** Survival curves for the treated and control G422 tumor mice. $n = 5$. Statistical significance was calculated by log-rank test. Source data are provided as a Source Data file.

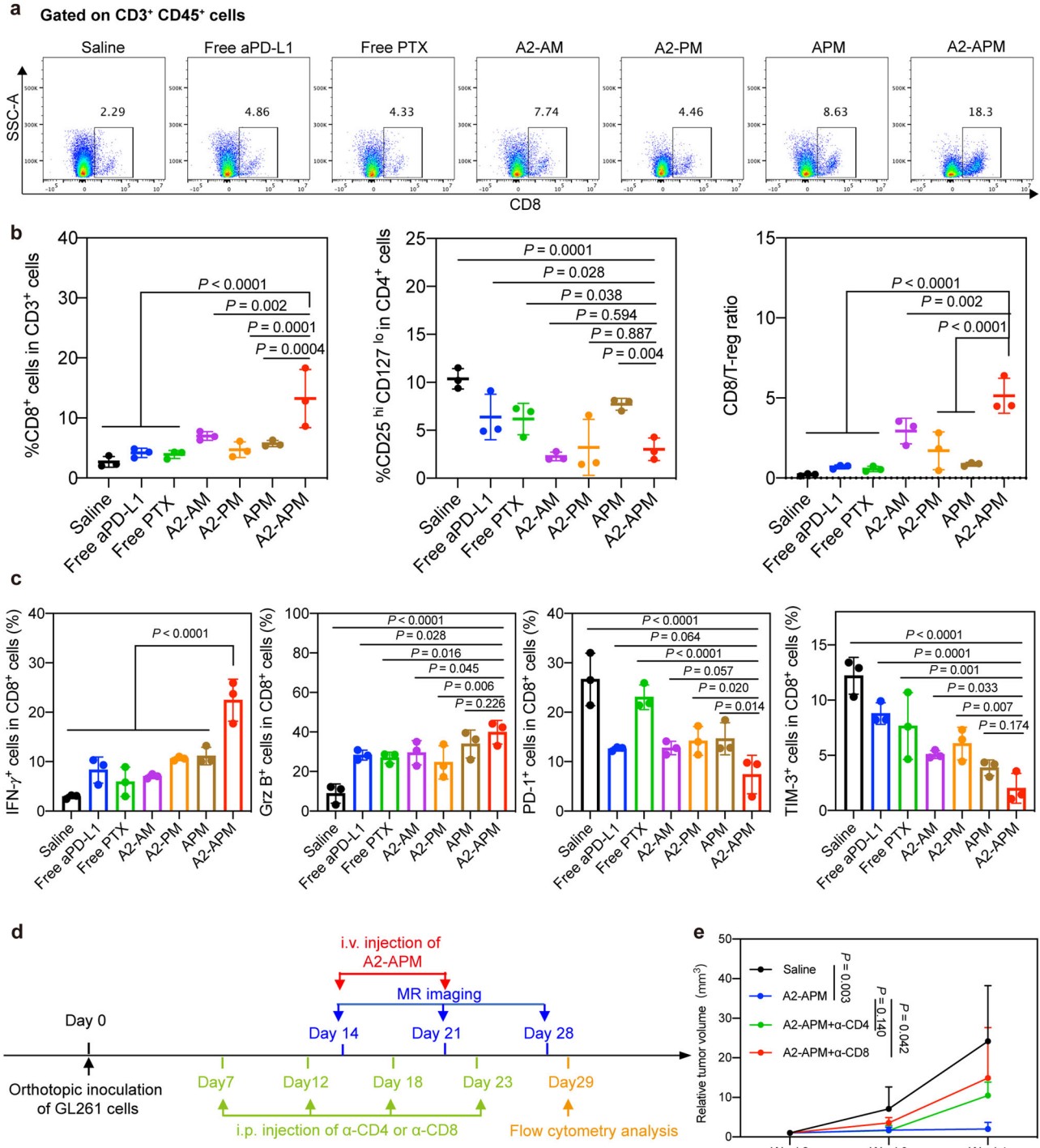

**Fig. 5 | A2-APM enhances specific immunity and its treatment effect is dependent on CD8[+] T cells. a** Representative flow cytometric contour plots of tumor-infiltrating CD8[+] T cell three days after two treatments with saline, free aPD-L1, free PTX, A2-AM, A2-PM, APM and A2-APM. Insets: the subsets of CD8[+] T cells in CD45[+]CD3[+] cells; the numbers indicate the percentage of CD8[+] T cells in CD45[+]CD3[+] cells after various treatments. **b** Quantification of tumor-infiltrating T cell three days after two treatments with saline, free aPD-L1, free PTX, A2-AM, A2-PM, APM and A2-APM, $n = 3$ mice. **c** Quantification of tumor-infiltrating IFNγ[+], Granzyme B[+] (GrzB[+]), PD-1[+] and TIM-3[+] T cells three days after two treatments with saline, free aPD-L1, free PTX, A2-AM, A2-PM, APM and A2-APM, $n = 3$ per group. **d** Schematic of the treatment regimen for depletion of T cells. **e** GL261 tumor volumes measured by MR after treatment with saline ($n = 3$), A2-APM ($n = 5$), A2-APM + α-CD4 ($n = 5$), A2-APM + α-CD8 ($n = 4$). All statistics are expressed as mean ± SD. Statistical significance was calculated by one-way ANOVA with Fisher's LSD test. Source data are provided as a Source Data file.

Therefore, mDCs in tumors was analyzed by flow cytometry. As shown in Fig. 6a, b and Supplementary Fig. 37a, biomarkers of mDCs (MHCII, CD80 and CD86) in the A2-APM group were significantly higher than those in the other groups. To figure out why there were more mDCs in A2-APM treated mice, the level of CRT, one of DAMPs that stimulate DC maturation, in GFP[+] GL261 tumor cells after various therapies were estimated (Supplementary Fig. 37b, c). A2-APM treatment significantly induced CRT exposure in GFP[+] GL261 tumor cells, supporting the ICD induction ability of this strategy to accelerate DC development.

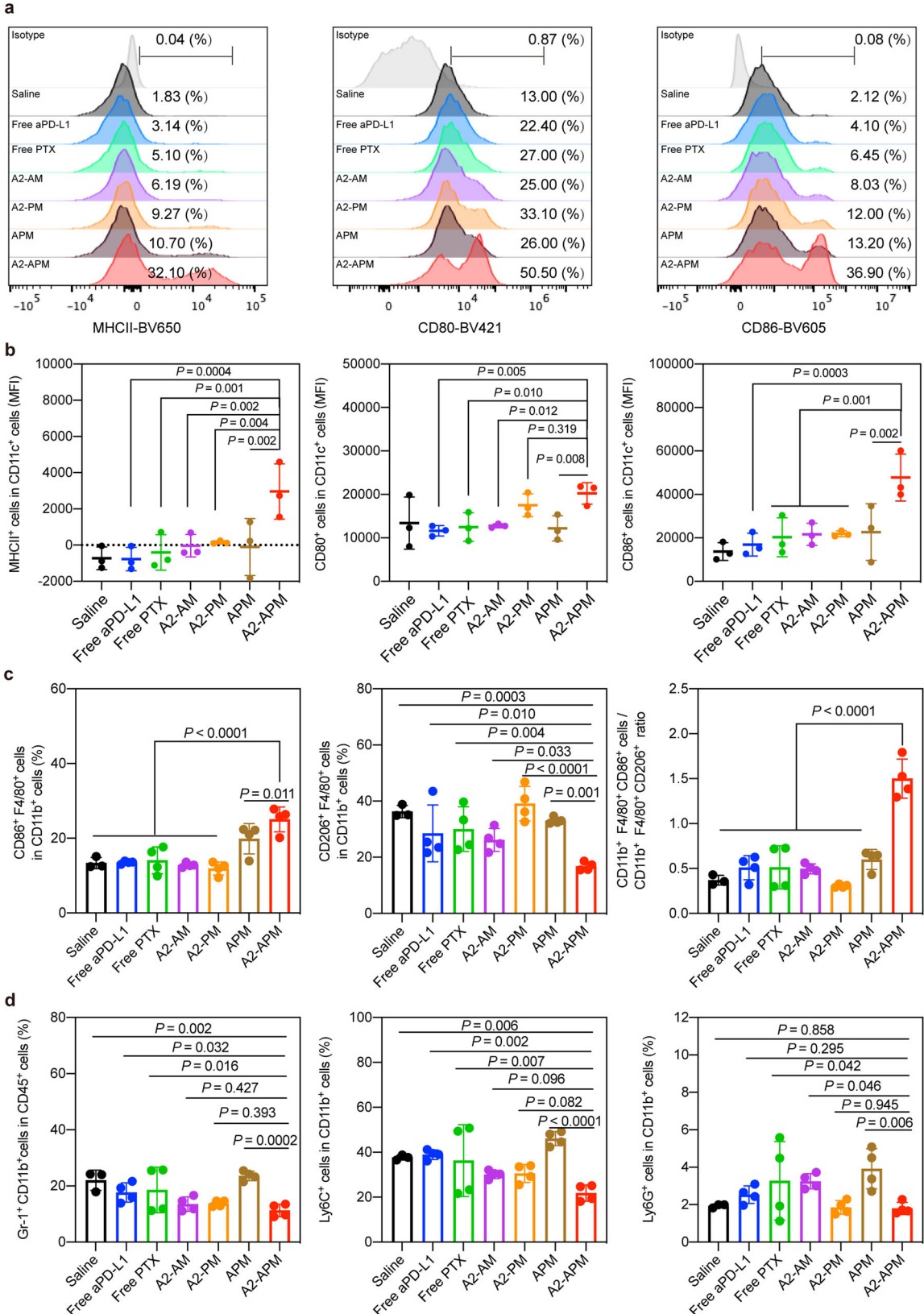

**Fig. 6 | A2-APM treatment promotes DC maturation and results in pro-inflammatory transformation of macrophages and decreased myeloid-derived suppressor cells. a** Representative histograms display each marker's (MHCII, CD80 and CD86) expression levels compared to isotype controls. **b** MFI of each marker's expression were calculated based on (**a**), n = 3 mice. **c** Quantification of tumor-infiltrating CD86+ F4/80+ and CD206+ F4/80+ cells three days after two treatments with saline, free aPD-L1, free PTX, A2-AM, A2-PM, APM and A2-APM. Saline treated mice (n = 3), free aPD-L1, free PTX, A2-AM, A2-PM, APM and A2-APM treated mice (n = 4). **d** Quantification of tumor-infiltrating Gr-1+ CD11b+, Ly6C+ and Ly6G+ cells three days after two treatments with saline, free aPD-L1, free PTX, A2-AM, A2-PM, APM and A2-APM. Saline treated mice (n = 3), free aPD-L1, free PTX, A2-AM, A2-PM, APM and A2-APM treated mice (n = 4). All statistics are expressed as mean ± SD. Statistical significance was calculated by one-way ANOVA with Fisher's LSD test. Source data are provided as a Source Data file.

This flow cytometry analysis showed a similar tendency of change in the brain of G422 GBM model, which confirmed the ability of A2-APM to elicit a strong immune response in GBM tumors (Supplementary Figs. 38–41). These findings manifest that A2-APM can effectively trigger a substantial local immunological response in tumors, resulting in CTL proliferation and infiltration, which suggests that A2-APM can enhance ICB effectiveness and be useful in the treatment of fatal GBM.

### A2-APM treatment results in a pro-inflammatory shift of macrophages and decreased myeloid derived suppressor cells

Myeloid cells are one of determinants of glioma TME immunotherapy[42], and PTX-induced ICD increases the adjuvanticity, which can activate both adaptive and innate antitumor immune responses[43]. We further identified the effect of the co-encapsulated aPD-L1 and PTX micelles on the dominant innate cells inside GBM tumor, such as tumor-associated macrophages (TAMs)[15]. The number of iNOS[+] pro-inflammatory macrophages was noticeably increased in the A2-APM treated group although the total number of F4/80[+] macrophages was basically unchanged, as shown in Supplementary Fig. 42. Since M0 macrophages were reported to express CD206 in prior research on GL261 innate immune cells[22], we further stained Arginase1 (Arg1) to observe anti-inflammatory macrophages. According to the data, only a small fraction of macrophages was Arg1-positive after the A2-APM treatment. Further, we confirmed the results by flow cytometry, and the increased percentage of CD86[+]F4/80[+] cells in CD11b[+]CD45[+] cells and decreased percentage of CD206[+]F4/80[+] cells in CD11b[+]CD45[+] cells were observed in A2-APM group in GL261 and G422 tumor mice (Fig. 6c, Supplementary Fig. 43a and 44).

Next, the change of myeloid derived suppressor cells (MDSCs) was detected. The number of either Gr-1[+]CD11b[+] cells or Ly6C[+]Ly6G[-] (monocytic-MDSCs, M-MDSCs) and Ly6C[-]Ly6G[+] (polymorphonuclear-MDSCs, PMN-MDSCs) cells was minimized in the A2-APM treated group as shown in Fig. 6d, Supplementary Figs. 43b and 45. These results imply that A2-APM is a nano-micelle that not only acts on T cells but also shifts macrophages to pro-inflammatory phenotype, and decreases MDSCs.

### A2-APM treatment elicits a prolonged immune memory

Immunological memory is essential to prevent GBM relapse, which defends against previously encountered infections and cancer cells[44]. The memory T cells in the brain, draining lymph nodes (dLNs), and spleen of the mice in native and A2-APM treatment groups were further examined on day 45 to confirm the induction of memory T cells and their potential to prevent tumor recurrence (Fig. 7a). The percentages of CD44[hi]CD8[+] T cells obtained from the brains and dLNs of the A2-APM treated mice were higher than those from native mice (Fig. 7b, e). Then the phenotype of memory T cells was evaluated by measuring the expression of CD62L, which is a lymphoid homing receptor. Compared with CD44[hi]CD8[+] T cells in the brain of native mice, the expression of CD62L was lower in A2-APM treated mice (Fig. 7c), which suggested that most of CD44[hi]CD8[+] memory T cells in the brain are resident, and are unable to circulate due to the lack CD62L to provide sufficient immunity to local infections[45].

The effector memory T cells ($T_{EM}$) and central memory T cells ($T_{CM}$) in the spleen of the mice in different treatment groups were further examined, the A2-APM group had a 2.71-fold increase in CD44[hi]CD62L[lo] $T_{EM}$ subsets (Fig. 7d, e), and a 7.59-fold increase in the ratio of $T_{EM}/T_{CM}$, demonstrating that A2-APM treated GBM animals had an activated immune memory.

In addition, in a tumor rechallenge assay, it was found that tumor development was decreased after re-inoculation of GL261 cells in the left brain of mice treated with A2-APM. Native mice, on the other hand, displayed highly aggressive tumor development with a much shorter survival time ($P = 0.007$, Supplementary Fig. 7f and g). These findings indicated that A2-APM treatment can activate a robust immuno-surveillance capacity, which is important for long-term survival of GBM.

## Discussion

We have developed a redox-sensitive micelle that co-encapsulates an immune checkpoint inhibitor (aPD-L1) and a chemotherapeutic agent (PTX) for effective therapy of GBM. GBM patient seldomly benefit from ICB therapy in the clinic due to limited antibody delivery and a restrictive immune milieu. Previous studies have confirmed the disruption of BTB, particularly with high-grade brain tumors, such as GBM[46]. Despite the increased permeability, the BTB retains the key characteristic of preventing the entry of macromolecular anticancer agents[25]. On the other hand, T cell dysfunction is a marker of GBM immune microenvironment, which imposes potent suppression of ICB therapy of GBM[47,48]. Emerging preclinical and clinical studies have shown that the combination of ICB with immunogenic drugs, such as chemotherapeutic agents that induce ICD, could enhance ICB efficacy by evoking the activation of antecedent T lymphocytes[13,49]. Thus, the combination of ICB-based immunotherapy with chemotherapy is an appealing solution aimed at achieving optimal benefit. NanoDDS provides promising strategies for cancer chemo-immunotherapy.

For the combination of multiple drugs, one agent can be administered as free form and others by NanoDDS (Free drug + Nano), or both were delivered by similar or different NanoDDS, respectively (Nano + Nano), or both agents were co-encapsulated in one NanoDDS (co-encapsulation)[26]. However, the "Free drug + Nano" and "Nano + Nano" approaches still suffer from potential mismatched pharmacokinetics of different drugs. "Free drug + Nano"[22] is mostly used to evaluate the potentiation of free drugs by nanomedicines, but regardless of whether the free drug is a chemotherapeutic agent or an antibody, it may be difficult to reach the target, resulting in insufficient therapeutic efficacy or side effects caused by overdose of the drug given. "Nano + Nano"[50] can be used for therapies that target different tissues, such as lymph nodes and tumors, but it may also allow the drug to accumulate in excess in healthy organs due to multi-nano formulation delivery. Moreover, the two types of nanoparticles may have mismatched half-lives and pharmacokinetics in vivo, and the onset of action in tumor tissue may be uncontrollable. Whether it is "Free + Nano" or "Nano + Nano" delivery, clinicians have to consider the sequence of drug delivery, balance the pharmacokinetics of the two drugs to reach the tumor site, and the time window between the two drugs, which increases the clinical workload. By contrast, the co-encapsulation approach allows for uniform distribution of drugs in vivo, and controls the accumulation in tumor tissue at a proper ratio[30,51]. Nonetheless, its application in GBM chemo-immunotherapy still faces the obstacle of a complex preparation process to protect the biofunction of antibodies.

Our NanoDDS is an effective remedy for the issues listed above. In vivo and ex vivo imaging studies indicated that A2-APM quickly detected the low-density lipoprotein receptor-related protein 1 (LRP1) on endothelial cells and GBM cells after intravenous treatment, to boost aPD-L1 and PTX entry into the tumor environment of GBM, resulting in substantially increased accumulation of antibodies in the brain in several tumor xenograft models, compared to free aPD-L1 group. Furthermore, the co-encapsulation design of the micelle avoids complex drug time windows or other adverse effects caused by systemic injection of antibodies and chemotherapeutic agents. In response to the reductive tumor microenvironment of GBM, the micelles were activated for the release of aPD-L1 and PTX in a safe manner. aPD-L1 recovered from cross-linking without losing its features to interrupt the PD-1/PD-L1 immune checkpoint, and PTX successfully regressed tumor cells while also amplifying ICB efficacy by inducing ICD effect. The sufficient T-cell receptor stimulation and co-stimulatory engagement with DCs are provided for normalizing T cell function to against tumors. The results of immunofluorescence and

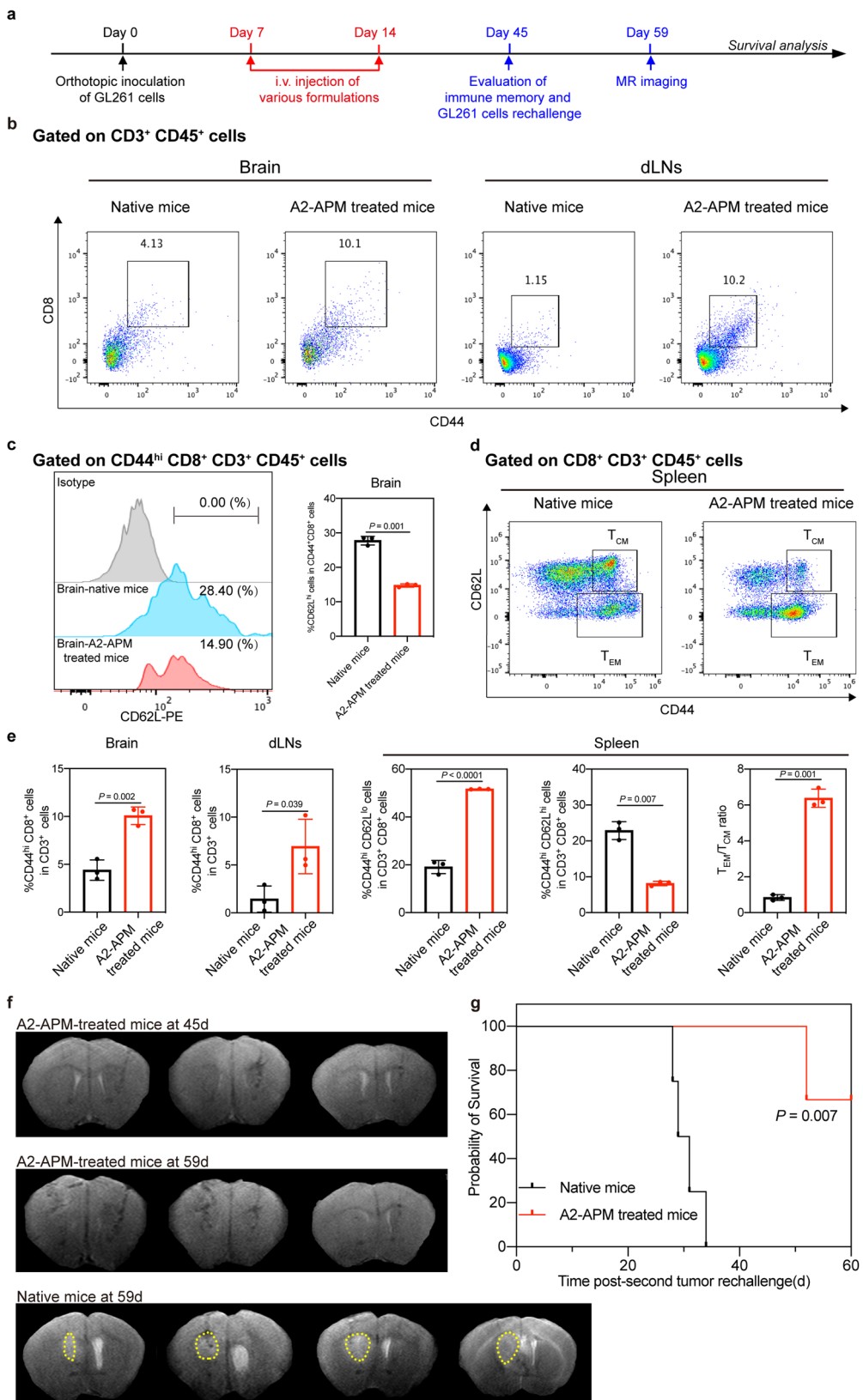

flow cytometry showed that A2-APM treatment greatly increased CTLs infiltration in the tumor microenvironment. Therefore, this strategy not only considerably increases ICB antibody delivery, but also improves tumor responsiveness to aPD-L1 to enhance the ICB efficacy. As a result, engineered A2-APM effectively targets GBM and amplifies the ICB potency, halting the growth of primary tumor and the recurrence of postoperative GBM, and producing a long-term immune memory that protects against tumor rechallenge. Our NanoDDS can increase the low response rate of ICB for GBM through increased co-encapsulation and specific activation of ICB antibodies and chemotherapeutic agents, as evidenced by the therapy of A2-APM to potently eradicate GBM in preclinical models.

**Fig. 7 | A2-APM treatment induces prolonged immune memory and protects the GBM mice against rechallenge with tumor cells. a** Description of the in vivo study plan. On days 7 and 14 after GL261 cell transplantation, mice received different treatments via i.v. injection. Then 3 out of the survived mice on day 45 post-inoculation were selected for the analysis of effector memory T cells in splenocytes. **b** Representative flow cytometry dot plot of CD8⁺CD44ʰⁱ memory T cells in brain and draining lymph nodes (dLNs) of naïve and A2-APM treated mice. **c** Representative flow cytometry histogram and quantification of CD62Lʰⁱ memory T cells in the brains of naïve and A2-APM treated mice ($n = 3$). **d** Representative flow cytometry dot plot of CD44ʰⁱ CD62Lʰⁱ central memory T cells ($T_{CM}$) and effector memory T cells ($T_{EM}$) in spleen of naïve and A2-APM treated mice. **e** Quantification of CD44ʰⁱCD8⁺ memory T cells in the brain and dLNs, CD44ʰⁱ CD62Lʰⁱ central memory T cells ($T_{CM}$) and effector memory T cells ($T_{EM}$) in the spleen of naïve and A2-APM treated mice ($n = 3$). **f** 3 out of the survived mice on day 45 post-inoculation were rechalled with GL261 tumor cells to check the performance of long-term memory effect. MR images of naïve and A2-APM treated mice rechalled with GL261 cells on days 45 and 59. The yellow dashed lines indicate the tumor area. **g** Survival curves for naïve ($n = 4$) and A2-APM ($n = 3$) treated mice after rechallenging with GL261 cells. All statistics are expressed as mean ± SD. Statistical significance was calculated by two-sided $t$-test in (**c**) and (**e**). Statistical significance was calculated by log-rank test in (**g**). Source data are provided as a Source Data file.

While combining chemotherapy and immunotherapy has been shown to improve the outcomes of many cancers[52,53], it is challenging to determine the precise effects of low antibody and chemotherapeutic agent delivery in GBM patients due to the ineffectiveness of ICB therapy and the systemic immunosuppressive effects of chemotherapy. Our NanoDDS may shed light on these issues with a simple manufacturing process to address the limitation of co-delivery of antibodies and chemotherapeutic agents into GBM. Therefore, it would be feasible to precisely regulate the therapy by measuring the responders throughout treatment, for instance, by adjusting drug doses or time window to overcome the chemo-immunotherapy resistance. The micelle provides good opportunities for drug co-delivery and controlled drug release, while reducing drug leakage during circulation and avoiding toxic effects, without requiring complicated custom modifications or alterations, due to its maneuverability of drug loading and redox-sensitive characteristics. In addition to aPD-L1, this smart nano-system may administer antagonists of multiple additional immunotherapeutic drugs, such as anti-CTLA4 antibody, anti-CD47 antibody, and other antibody molecules. Micelles enable the co-encapsulation of various medications to improve effectiveness, allowing immunotherapy and other therapeutic agents to be combined.

Moreover, the original materials utilized in A2-APM are simple and safe; this technology might allow for greater clinical practice coordination. A2-APM is a relatively simple formula that is readily scalable. The active pharmaceutical ingredients used in our study, aPD-L1 and PTX, have demonstrated manageable toxicity in early clinical trials[54,55]. PEG-PLL is already commercially available, and some nano-micelles represented by Genexol® PM (PTX-encapsulated nano-micelles) are already clinically available[56]. In our study, A2-APM demonstrated good stability (Supplementary Fig. 5), and good safety profile. It did not cause vascular abnormalities in the brain or induce cerebral edema (Supplementary Figs. 25–31). Therefore, A2-APM is promising for clinical translation.

Although our study provides a clear example of the importance of co-encapsulation drug delivery formulations for reversing the immunosuppressive tumor microenvironment, the use of GBM mouse tumor model is still limited due to the complexity of GBM immune cell heterogeneity[57]. Future studies could explore larger animal models of GBM (such as canine or pig) and evaluate more clinically relevant therapeutic benefits, such as functional outcomes. Most importantly, acute toxicity tests, long-term dosing tests, genotoxicity tests, and reproductive toxicity tests should be performed and the pharmacokinetics of A2-APM should be thoroughly evaluated before clinical translation.

In summary, our studies provide an approach for antibody penetration into the brain and to use chemotherapy-induced ICD for boosting antitumor immunity with brain tumors, and the micelles might have broader implications for immunotherapy in other immunosuppressive malignancies.

## Methods
### Ethics statement
All animal experiments were conducted according to the ethical guidelines of the Animal Care & Welfare Committee of Southeast University, Jiangsu, China. All procedures performed in studies involving human participants were in accordance with the ethical standards of the Ethics Review Committee of Huashan Hospital Affiliated to Fudan University. According to the Ethics Committee, the mice should be euthanized when the size of brain tumors exceeds 100 mm³ and neurological impairment of the mice was observed, such as full paralysis of the hindlimbs and tail and partial paralysis of the forelimbs. In survival analysis, in addition to the death outcome, the moribund status of mice, including lack of response to stimulation, immobility, or inability to eat or drink, were also used as an endpoint of euthanasia.

### Materials
Maleimide-polyethylene glycol-poly-L-lysine (Mal-PEG-PLL) (molecular weight of PEG is 12,000 and poly-L-lysine degree of polymerization 72) was purchased from Xi'an Ruixi Biological Technology Co., Ltd. (Shaanxi, China). Paclitaxel (PTX) was purchased from MedChemExpress LLC. (Shanghai, China). Angiopep-2 peptide (TFFYGGSRGKRNNFKTEEYC) was purchased from ChinaPeptides Co., Ltd. (Shanghai, China). FITC-NHS, Alexa Fluor 647-NHS, and Cy7.5-NHS were purchased from Nanjing Goyoo Biotech. Co., Ltd. (Jiangsu, China). Reductive glutathione (GSH) was obtained from Shanghai Yuanye Biotech. Co., Ltd. (Shanghai, China). Cell counting kit-8 assay (CCK-8) was purchased from Vazyme Biotech. Co., Ltd. (Jiangsu, China). Calcein/PI Live/Dead Viability/Cytotoxicity assay kit and goat serum were obtained from Beyotime Biotech. Co., Ltd. (Shanghai, China). SDS-PAGE gel preparation kit and Annexin V-FITC/PI apoptosis detection kit were obtained from KeyGEN Biotech. Co., Ltd. (Jiangsu, China). Commassie blue staining and destaining solution, Trion X-100 and antifade mounting medium for fluorescence (with DAPI) were purchased from Biosharp (Anhui, China). Cell passive lysis buffer used in this research was purchased from Promega (Wisconsin, USA). ATP assay kit and mice lymphocyte separation medium were provided by Beijing Solarbio Science & Technology Co., Ltd (Beijing, China). CD45 magnetic beads were designed by Nanjing Nanoeast Biotech. Co., Ltd. (Jiangsu, China). FITC-TSA, Cy3-TSA and 647-TSA were purchased from Wuhan Servicebio Technology Co., Ltd (Hubei, China).

### Cell culture and animals
The murine GL261 cells and bEnd.3 cells were obtained from KeyGEN Biotech. Co., Ltd. (Jiangsu, China). The murine G422 cells were obtained from FuHeng Biology (Shanghai, China). The G7 and WL1 patient-derived xenograft (PDX) cell lines were obtained from Professor Xiuxing Wang's laboratory cell bank. The human U87 cells and U251 cells and the murine GL261-GFP and G422-GFP cells were obtained from our laboratory cell bank. G7 and WL1 cells ere cultured in Neurobasal® medium (Life Technologies, Cat. No.21103049), supplemented with 2% B27-supplement without Vitamin A (Life Technologies, Cat. No. 12587010), 1% Sodium Pyruvate (Life Technologies, Cat. No.11360-070), 1% Gluta-MAX™ Supplement (Life Technologies, Cat. No. 35050061), 100 U/mL penicillin/streptomycin (Invitrogen), 20 ng/mL rhEGF (R&D, Cat. No. 4114-TC-01M) and 20 ng/mL FGF (Proteintech, Cat. No. 100-8C-10). U87, U251, GL261, and G422 cells were cultured in DMEM medium (HyClone) supplemented with 10% (v/v) fetal bovine serum (FBS) (Gibco) and 100 U/mL penicillin/streptomycin (Invitrogen). All cells were cultured at

37 °C in a 5% $CO_2$ water-jacketed incubator (Thermo Fisher Scientific, Massachusetts, USA).

C57BL/6 mice (male; 6-8 weeks old), Kunming mice (female; 4-5 weeks old), BALB/c nu mice (female; 6-8 weeks old) and SD rats (male; 6-8 weeks old) were purchased from Cavens Biogle(suzhou) Model Animal Research Co., Ltd. (Jiangsu, China). All animal experiments were conducted according to the ethical guidelines of the Animal Care & Welfare Committee of Southeast University, Jiangsu, China.

## Patient samples
Brain tumor specimens were obtained from patients with informed consent and were reviewed by the pathologist and surgeon. Pathologist classified the type and grade of the tumors in accordance with the WHO histological grading of central nervous system tumors. 14 cases of gliomas were selected from Huashan Hospital, Fudan University, and 6 cases of gliomas were selected from Zhongda Hospital, Southeast University. Characteristics of glioma patients were shown in Supplementary Table 1.

## Antibody
Anti-mouse PD-L1 (B7-H1) (Cat. No. BE0101, 1.5 mg/kg for mice, 1:1000 for cells), anti-CD4 (Cat. No. BE0119, 1000 μg/mouse), anti-CD8 (Cat. No. BE0061, 500 μg /mouse) were purchased from Bio X Cell (New Hampshire, USA). Rabbit IgG control (whole molecule), purified (Cat. No. A01008, 1:1000) was purchased from GenScript (Jiangsu, China). CD16/CD32 monoclonal antibody (Cat. No. 14-0161-81, 1 μg/$10^6$ cells) and the antibodies used for flow cytometry were specific for CD3 (Cat. No. 69-0032-82, 0.5 μg/$10^6$ cells), CD4 (Cat. No. 11-0041-82, 0.25 μg/$10^6$ cells), CD8a (Cat. No. 78-0081-82, 0.5 μg/$10^6$ cells), CD25 (Cat. No. 17-0251-81, 0.125 μg/$10^6$ cells), CD127 (Cat. No. 12-1271-82, 0.25 μg/$10^6$ cells), CD11c (Cat. No. 45-0114-82, 0.25 μg/$10^6$ cells), MHC II (Cat. No. 64-5321-82, 0.125 μg/$10^6$ cells), CD80 (Cat. No. 62-0801-82, 0.06 μg/$10^6$ cells), CD86 (Cat. No. 63-0862-80, 0.125 μg/$10^6$ cells), CD133 (Cat. No.11-1331-80, 0.5 μg/$10^6$ cells), CD44 (Cat. No.45-0441-80, 0.25 μg/$10^6$ cells), NESTIN (Cat. No. MA523574, 10 μL/$10^6$ cells), SOX2 (Cat. No. 50-9811-80, 0.125 μg/$10^6$ cells), F4/80 (Cat. No. 12-4801-82, 0.25 μg/$10^6$ cells), and the antibodies used for immunofluorescence were specific for PD-L1 (Cat. No. 14-5983-82, 5 μg/mL), PD-1(Cat. No. 14-2798-82, 5 μg/mL), CD11b (Cat. No. MA1-80091, 5 μg/mL) were purchased from eBioscience (California, USA). The antibodies used for flow cytometry were specific for CD44 (Cat. No. 103049, 0.5 μg/$10^6$ cells) and CD62L (Cat. No. 104407, 0.25 μg/$10^6$ cells), CD45 (Cat. No. 147705, 0.5 μg/$10^6$ cells), PD-1 (Cat. No. 114117, 1 μg/$10^6$ cells), TIM-3 (Cat. No. 119723, 0.25 μg/$10^6$ cells), NK1.1 (Cat. No. 108745, 0.25 μg/$10^6$ cells), Gr-1(Cat. No. 108443, 0.25 μg/$10^6$ cells), Ly6C (Cat. No. 128032, 5 μL/$10^6$ cells), Ly6G (Cat. No. 127614, 0.06 μg/$10^6$ cells), Granzyme B (Cat. No. 396424, 5 μL/$10^6$ cells), IFN-γ (Cat. No. 505810, 1 μg/$10^6$ cells), CD11b (Cat. No. 101206, 0.25 μg/$10^6$ cells), CD206 (Cat. No. 321124, 5 μL/$10^6$ cells) were purchased from Biolegend (California, USA). Anti-mouse Calreticulin (CRT) monoclonal antibody was purchased from Cell Signaling Technology (Cat. No. 12238 s, 1:200), anti-human CRT (Cat. No. 10292-1-AP, 1: 50) and anti-human TIM-3 (Cat. No. 60355-1-Ig, 1: 50) monoclonal antibody was purchased from Proteintech. HMGB1 monoclonal antibody was purchased from Abcam (Cat. No. ab18256, 1 μg/mL). The antibodies used for immunofluorescence were specific for CD3 (Cat. No. GB13014, 1:200), CD4 (Cat. No. GB13064-2, 1:3000), CD8 (Cat. No. GB13429, 1:1000), Foxp3 (Cat. No. GB112325, 1:3000), CD11c (Cat. No. GB11059, 1:3000), F4/80 (Cat. No. GB13373, 1:3000), CD206 (Cat. No. GB13438, 1:3000), iNOS (Cat. No. GB11119, 1:400), Arg1 (Cat. No. GB11285, 1:200) were purchased from Wuhan Servicebio Technology Co., Ltd (Hubei, China). The secondary antibodies used for immunostaining were FITC conjugated goat anti-rat IgG (H&L) (Cat. No. SA00003-11, 1:1000, Proteintech Group Inc., Illinois, USA), Cy3 conjugated goat anti-rat IgG (H&L) (Cat. No. SA00009-1, 1:1000, Proteintech Group Inc., Illinois, USA), goat anti-rabbit IgG H&L (Alexa Fluor 647) (Cat. No. ab150079, 1:1000, Abcam, Cambridge, UK), HRP conjugated goat anti-rabbit IgG (H&L) (Cat. No. GB23303, 1:500, Servicebio, Hubei, China),

Cy3 conjugated goat anti-rabbit IgG (H&L) (Cat. No. GB21303, 1:500, Servicebio, Hubei, China) and Alexa Fluor 549 conjugated goat anti-rabbit IgG (H&L) (Cat. No.111-585-003, 1:1000, Jackson ImmunoResearch Inc., Pennsylvania, USA). Enzyme-linked immunosorbent assay (ELISA) kits were used to measure the culture medium concentration of IgG (Cat. No. CSB-E06949Rb, Cusa Bio., Hubei, China) and HMGB1 (Cat. No. ARG81310, Arigo Biolaboratories Co., Taiwan, China).

## Detection of PD-L1 expression in GBM patients, mice and cells
For immunofluorescence, tissue sections were used, and PD-L1, CD11b, DAPI stainings were performed according to the manufacturer's instructions. CaseViewer 2.4 was used to observe the images. Five different parts of each slice were randomly selected, and Image Pro Plus 6.0 was used to count positive cells.

For cell flow cytometry, G7, WL1, U251, U87, GL261, and G422 tumor cells were incubated with aPD-L1 for 30 min and washed with PBS 3 times. Next, the Alexa647-labeled or Cy3-labeled secondary antibody was incubated with different cells for another 30 min, before flow cytometry (NovoCyte Flow Cytometer, ACEA Biosciences, California, USA), resuspend the cells in 500 μL PBS, and then analyzed using the FlowJo 10.4 software package to measure the binding of PD-L1 on the cell membrane.

## Synthesis of SC-$(CH_2)_2$-S-S-$(CH_2)_2$-SC
Dithiodiethanol (1.54 g, 10 mmol) and N,N'-disuccinimidyl carbonate (DSC) (5.89 g, 23 mmol) were dissolved in anhydrous MeCN (10 mL). The solution was cooled down to 0 °C and triethylamine (2.32 g, 23 mmol) was added. The reaction was allowed to warm to room temperature and left to react for 4 h. The solvent was removed in vacuo and the residue dissolved in DCM. The organic phase was washed 3 times with water, dried with magnesium sulfate and removed in vacuo. The remaining oil was purified by flash column chromatography (silica gel 60 Å, 35-70 μm, eluent 95/5 diethyl ether−methanol).

## Synthesis and characterization of micelles
APM is synthesized using Mal-PEG-PLL, SC-$(CH_2)_2$-S-S-$(CH_2)_2$-SC, aPD-L1 and PTX. Mal-PEG-PLL (19 kDa, 0.318 mg, 1 mg/mL) and aPD-L1 (0.5 mg, 1 mg/mL) were stirred at 4 °C for 10 min. Next, vortex the mixed solution with the SC-$(CH_2)_2$-S-S-$(CH_2)_2$-SC (16.8 μg, 1 mg/mL) for 1 min, and then add PTX (1 mg, 20 mg/mL) to the reaction mixture. The mixture was left at 4 °C for 4 h, then angiopep-2 (0.04 mg, 1 mg/mL) was added, and the reaction was complete within 1 h at room temperature. The residual free drug was removed by ultrafiltration (MWCO: 100 kDa) with PBS (pH = 7.4). The preparation methods of AM and PM were the same as above, except for the absence of the other drug.

The size distribution and zeta potential of APM were evaluated by using dynamic light scattering measurements in PBS (pH 7.4) at 25 °C using a ZetaPlus (Brookhaven Instruments, New York, USA). The morphology of micelles was observed by transmission electron microscopy (TEM) (Hitachi H7650, Tokyo, Japan).

The ELISA kit quantifies the loading content (LC%) and encapsulation efficiency (EE%) of IgG following the manufacturers' instructions. The PTX concentration quantified by HPLC analysis (Waters, Massachusetts, USA) was performed on methanol/acetonitrile/water (40/35/25 volume ratio) as the eluent. EE% and LC% were calculated by the following formulas (1) and (2):

$$EE\% = \frac{amount\ of\ loaded\ drug}{amount\ of\ drug\ added} \times 100\% \tag{1}$$

$$LC\% = \frac{amount\ of\ loaded\ drug}{amount\ of\ drug\ loaded\ micelles} \times 100\% \tag{2}$$

## Reduction-induced activity recovery of APM

SDS-PAGE was used to test antibody conjugation and release studies. The sample solution (11.25 μL; 7 μg aPD-L1) was mixed with 3.75 μL non-reducing 5× loading buffer. Then, 15 μL of the sample solution was loaded on the acrylamide gel. Among these samples, APM samples were pretreated with or without GSH (2 mM), incubated at 37 °C for 1 h, then GSH was removed by ultrafiltration (MWCO: 3 kDa) with PBS (pH 7.4) before being subjected to SDS-PAGE. The image was taken by Molecular Imager Gel Doc XR⁺ System (BioRad, California, USA)

To quantitatively measure the binding of aPD-L1 and APM with PD-L1, GL261 tumor cells were incubated with aPD-L1 and APM pretreated with or without GSH (2 mM) for 1 h and washed with PBS 3 times. Next, the FITC-labeled secondary antibody was incubated with GL261 cells for another hour. Before flow cytometry (NovoCyte Flow Cytometer, ACEA Biosciences, California, USA), the cells were resuspended in 500 μL PBS, and then analyzed using the FlowJo 10.4 software package to measure the binding of PD-L1 on the cell membrane.

The dialysis diffusion method was used to estimate the in vitro release behavior of PTX and IgG from APM under different GSH concentrations. In short, 1 mL of APM solution (containing 300 μg PTX and 150 μg IgG) was added to each dialysis bag (MWCO: 300 kDa). Then, the dialysis bag was kept in 20 mL of release medium at a temperature of $37 \pm 0.5\,°C$ and shaken horizontally (100 rpm). At predetermined intervals, 400 μL of the release medium was removed from the dialysis bag and was replaced by the same amount of fresh medium. An ELISA kit was used to detect the content of IgG in the harvested samples. Then, an equal volume of acetonitrile was added to the solution, and HPLC analysis was performed to detect the concentration of released PTX.

## Acquisition and identification of glioma stem-like cells

To obtain glioma-stem like cells (GSCs), GL261 cells were firstly seeded in 6-well ultra-low-attachment plates (Corning) at 5000 cells per well to obtain tumorspheres. The medium was changed after 3 days. Tumorspheres were cultured for 7 days for the subsequent experiments. All cells were cultured at 37 °C in a 5% $CO_2$ water-jacketed incubator (Thermo Fisher Scientific, Massachusetts, USA).

To obtain CD133⁺ GSCs, tumorspheres were harvested and washed with PBS twice, then resuspended with 200 μg/m of Liberase-TL (Roche) at 37 °C. Thereafter, the cells were stained with anti-CD133 antibodies for 30 min. The stained cells were further flow sorted by flow cytometry (FACSCelesta, Becton, Dickinson and Company, New Jersey, USA) for CD133 cell populations. The CD133⁺ GSCs were cultured with neural stem cell medium for the subsequent experiments.

To identify GSCs, the GSCs were harvested and washed with PBS twice, then stained with anti-CD133, anti-CD44, anti-SOX2, and anti-NESTIN antibodies for 30 min. Then the cells were resuspended in 500 μL PBS and examined using a NovoCyte Flow Cytometer (ACEA Biosciences, California, USA). The data were analyzed with Flowjo.

For colony formation assay, GL261 cells or GSCs ($2 \times 10^3$) suspensions were incubated in an upper layer of 0.7% agarose in DMEM with 10% FBS. The suspension was overlaid on 1.2% basal agar with 10% FBS in a 60 mm dish and placed at room temperature until the agarose solidified. The plates were transferred to a 5% CO2 incubator and were incubated at 37 °C for 2 weeks, and then colonies larger than 0.5 mm in diameter were counted.

## Cytotoxicity and cell apoptosis

GL261 cells were seeded into a 96-well plate at a density of $1 \times 10^4$ cells per well, and incubated with blank micelle, free PTX, AM, PM, or APM at different concentrations for 24 h. 20 μL of CCK-8 assay kit (CCK-8 reagent: medium = 1:10, v/ v) was added to each well and then incubated for 1-4 h in the dark. A microplate reader (Multiskan Go, Thermo Fisher Scientific, Massachusetts, USA) was used to measure the

absorbance at 450 nm. Untreated control cells were considered 100% viable.

GSCs were seeded into a 96-well plate at a density of $5 \times 10^3$ cells per well, and incubated with free PTX, PM, or APM at different concentrations for 24 h. 20 μL of CCK-8 assay kit (CCK-8 reagent: medium = 1:10, v/ v) was added to each well and incubated for 1–4 h in the dark. A microplate reader (Multiskan Go, Thermo Fisher Scientific, Massachusetts, USA) was used to measure the absorbance at 450 nm. Untreated control cells were considered 100% viable.

Then, GL261 cells were seeded in a 12-well plate at a density of $1 \times 10^5$ cells per well, different treatments were added to incubate for 24 h. For live/dead detection, the cells were washed once with PBS, and then 500 μL Calcein/PI Live/Dead Viability/Cytotoxicity assay kit was added to each well and incubated for 30 min at 37 °C in the dark. Finally, each well was washed twice with PBS, and then transferred to a fluorescence microscope (ECLIPSE Ti2-U microscope, Nikon Corporation, Tokyo, Japan) for observation. For cell apoptosis analysis, the cells were collected and analyzed using the Annexin V-FITC/PI staining kit. Before examination using a NovoCyte Flow Cytometer (ACEA Biosciences, California, USA), the cells were resuspended in 500 μL PBS, and then the data were analyzed using Flowjo.

## ICD effect of APM in vitro

To evaluate the ICD effect of GL261 cells or GSCs before and after different treatments, GL261 cells or GSCs ($1 \times 10^5$ cells) were seeded in a 12-well plate. After incubation for 24 h, the cell culture medium was removed, and the cells were not treated or treated with blank micelle, free PTX, AM, PM, or APM (1 μg/mL aPD-L1 or 2 μg/mL PTX equivalent; free PTX, PM or APM for GSCs) for 24 h to induce ICD. To measure the CRT expressions, after treatment, the GL261 cells or GSCs were washed with cold PBS, and further incubated with anti-CRT polyclonal antibody (diluted 1:100) at 4 °C overnight and stained with Alexa Fluor 647 labeled goat antirabbit IgG (H&L) highly cross-adsorbed secondary antibody (1:200 dilution) for 30 min at room temperature. Then the cells were resuspended in 500 μL PBS and examined using a NovoCyte Flow Cytometer (ACEA Biosciences, California, USA), analyzed with Flowjo. To observe the CRT expressions on the GL261 cells, the nuclei were stained with 1 μg/mL DAPI for 5 min, and then evaluated using a Laser Scanning Confocal Microscopy (Olympus FV3000, Tokyo, Japan).

To evaluate the ICD effect of G7, WL1, U87, U251, GL261, and G422 cells before and after PTX treatment at different concentrations, G7, WL1, U87, U251, GL261, and G422 cells ($1 \times 10^5$ cells) were seeded in a 12-well plate. After incubation for 24 h, the cell culture medium was removed, and the cells were not treated or treated with for different concentrations of PTX for 24 h to induce ICD. To measure the CRT expressions, after treatment, the cells were washed with cold PBS, and further incubated with anti-CRT polyclonal antibody (diluted 1:100, anti-human antibody for G7, WL1, U87, and U251 cells and anti-mouse antibody for GL261 and G422 cells) at 4 °C for 30 min, and stained with Alexa Fluor 647 labeled goat antirabbit IgG (H&L), a highly cross-adsorbed secondary antibody (1:200 dilution), for 30 min at room temperature. Then the cells were resuspended in 500 μL PBS and examined using a NovoCyte Flow Cytometer (ACEA Biosciences, California, USA), and the data were analyzed with Flowjo.

To evaluate the secretion of ATP and HMGB1, after treatment, the cell supernatant was detected with the ATP assay kit and mouse HMGB1 ELISA kit, respectively, following the manufacturer's instructions. To observe the secretion of intracellular HMGB1, the cells were treated with Triton X-100, then incubated with anti-HMGB1 polyclonal antibody (use a concentration of 1 μg/mL) at 4 °C overnight, and stained with Alexa Fluor 647 labeled goat antirabbit IgG (H&L) highly cross-adsorbed secondary antibody (1:200 dilution) for 30 min at room temperature, the nuclei were stained with 1 μg/mL DAPI for 5 min, while images were obtained by ECLIPSE Ti2-U microscope (Nikon Corporation, Tokyo, Japan). To evaluate the secretion of

intracellular HMGB1, the cell nuclei were separated by Nuclear and Cytoplasmic Extraction Reagents (Hangzhou FUDE Biological. Technology CO., LTD. Cat. N0. FD0199) first and then detected with the mouse HMGB1 ELISA kit

To evaluate the ICD effect on DC cells, DC2.4 ($5 \times 10^5$ cells) were seeded in a 12-well plate overnight. Then GL261 cells (1:1) pretreated with different treatments (1 μg/mL aPD-L1 or 2 μg/mL PTX equivalent) were added. After 24 h of co-cultivation, these treated DC2.4 cells were then stained with anti-CD11c, anti-MHC-II, anti-CD80, and anti-CD86 antibodies before measurement by flow cytometry (FACSCelesta, Becton, Dickinson and Company, New Jersey, USA). Results were analyzed by Flowjo.

To evaluate the ICD effect on macrophages or DCs and their effect on T cell activation, we extracted primary BMDMs and BMDCs. In brief, the femurs, tibias, and spleen were dissected from C57BL/6 mice. Then, the bone marrow was rinsed and cultured in a medium (DMEM containing 20 ng mL$^{-1}$ M-CSF for macrophages, RPMI1640 containing 20 ng mL$^{-1}$M-CSF for DCs) for 6 days, the spleen was sheared and digested, then sorted by magnetic beads and cultured in a RPMI1640 medium. On day 7, naïve BMDMs were collected and cultured with a medium containing untreated tumor cells and APM-treated tumor cells for 24 h. After that, T cells were collected and cultured in the culture dish for another 24 h. Then the cells were washed with cold PBS, and further incubated with anti-F4/80, anti-CD86, and anti-CD206 antibodies for macrophages, anti-CD11c, anti-CD80, and anti-CD86 antibodies for DCs, and anti-CD3, anti-CD8, and anti-IFN-γ for T cells for 30 min. Then the cells were resuspended in 500 μL PBS, examined using a flow cytometry (FACSCelesta, Becton, Dickinson and Company, New Jersey, USA), and analyzed with Flowjo.

### Penetration studies of brain targeting ability on BTB model
The mouse brain endothelial bEnd.3 and GL261 cells were seeded into the transwell upper and lower chambers (Corning, New York, USA), respectively, at a density of $2 \times 10^5$ cells/mL per well in a 6-well plate for monolayer cell culture. FITC labeled aPD-L1 was used to prepare fluorescent-free aPD-L1, APM, and A2-APM. Then, different treatments were added to the upper chamber and incubated for 4 h. The solution in the lower chamber was collected and measured by a microplate reader (Varioska LUX, Thermo Fisher Scientific, Massachusetts, USA).

### In vivo pharmacokinetics and biodistribution
To investigate the pharmacokinetic behavior, Alexa Fluor 647 labeled aPD-L1 was used to prepare fluorescent aPD-L1 and A2-APM, then different treatments (aPD-L1 equivalent dose of 1.05 mg/kg) were intravenously given to healthy SD rats. Blood samples were collected at 0.5, 1, 2, 4, 6, 8, 12, 18, 24, 36, and 48 h and treated with GSH (2 mM). Later, a fluorescence microplate reader (Varioska LUX, Thermo Fisher Scientific, Massachusetts, USA) was used to measure the concentration of aPD-L1.

To explore the biodistribution, Cy7.5-labeled aPD-L1 was used to prepare fluorescent aPD-L1, APM and A2-APM, then different formulas (aPD-L1 equivalent dose of 1.5 mg/kg) were intravenously given to C57BL/6 mice bearing GL261 tumors, near-infrared imaging (CRi, Woburn, Massachusetts, USA) was performed at 1, 2, 4, 6, 8, 12, 24, 48, and 72 h post-injection. Next, the mice were sacrificed at 90 min post-injection to extract the heart, liver, spleen, lung, kidneys, and brain for ex vivo fluorescence imaging. Then the main organs were collected at 90 min after injection, mouse lymphocyte separation medium was used for dissociation, the cells were stained with anti-CD45, anti-CD3, anti-NK1.1, anti-CD11b, anti-CDLy6C, and anti-CDF4/80 antibodies for 30 min. Thereafter, the stained cells were analyzed by flow cytometry (FACSCelesta, Becton,Dickinson and Company, New Jersey, USA).

To determine the amount of PTX reaching the brain in A2APM, at 90 min post-injection of A2-APM, the GL261 tumor-bearing mice were sacrificed, and the brains were harvested. The homogenate (0.2 mL) was mixed with acetonitrile (0.2 mL), vortexed for 5 min and

centrifuged at 10,000 g for 10 min. The amount of PTX extracted in the supernatant was quantified by HPLC. The brain-targeting concentration was then calculated.

### Antitumor activity in orthotopic GBM models
GL261 cells or G422 cells ($1.0 \times 10^5$/2 μL, respectively) were intracranially inoculated in the right brain of C57BL/6 or Kunming mice. The inoculation day was defined as day 0. Seven days later, mice were randomized, and the drugs were injected intravenously via the tail vein. Free aPD-L1, Free PTX, Free aPD-L1 and Free PTX, A2-AM, A2-PM, APM, A2-APM (1.5 or 3 mg/kg on an aPD-L1 or PTX basis), were administered twice, on days 7 and 14. MR were used to monitor tumor growth. Mouse survival and body weight were recorded and the statistical significance of survival extension was analyzed by Log-rank test. All tumor size did not reach a volume of 100 mm$^3$(maximum tumor size).

Tumor rechallenge study was further investigated. C57BL/6 mice were injected with $1 \times 10^5$/2 μL GL261 cells in the left brain 45 days after the first injection of GL261 cells and subsequently treated with A2-APM. As control, $1.0 \times 10^5$/2 μL GL261 cells were intracranially inoculated in the left brain of naïve C57BL/6 mice. MR was then applied to monitor tumor growth until day 59 and the survival time was recorded. The statistical significance of survival extension was analyzed by Log-rank test.

For the resected GBM mouse model, the tumor-bearing mice underwent surgical resection on day 14 under a microscope at 20× magnification. Two days later, mice were randomized, and the drugs were injected intravenously via the tail vein. Free aPD-L1, Free PTX, APM, and A2-APM (1.5 or 3 mg/kg on an aPD-L1 or PTX basis) were administered twice, on days 16 and 23. MR was used to monitor tumor growth before and after surgery. Mouse survival was recorded and the statistical significance of survival extension was analyzed by Log-rank test.

For hematoxylin-eosin (H&E) staining of tumor, mice from each group were sacrificed and perfused with cold PBS followed by 4% paraformaldehyde (PFA) at scheduled time. Brains were carefully isolated, fixed for an additional 24 h with 4% PFA, and processed for preparing paraffin sections. Brain sections were stained with H&E to visualize the tumor. For detection, slices were placed under the scanner (Pannoramic MIDI, 3Dhistech, Budapest, Hungary) for image acquisition.

### MR of orthotopic tumors
In vivo MR was performed using a 7.0-Tesla small animal MR scanner (Bruker Pharmascan, Ettlingen, Germany). Mice were anesthetized with 1% isoflurane. Respiratory rate and body temperature were monitored using a physiological monitor (1025; SA Instruments, Stone Creek, New York). The parameters of T2-weighted images were as follows: TR = 3000 ms, TE = 36 ms; matrix size = 256 × 256; field of view = 2.0 × 2.0 cm; and slice thickness = 1 mm. MR images analysis were performed using Horos.

### Immune status investigation
For immunofluorescence, C57BL/6 mice bearing tumors (2 weeks after inoculation) were intravenously injected with free aPD-L1, free PTX, A2-AM, A2-PM, APM, or A2-APM (1.5 or 3 mg/kg on an aPD-L1 or PTX basis). The immune cell infiltration in the tumors was analyzed 3 days after the last treatment. GBM-bearing mouse brains were fixed by reflux of an intravenous injection of a 4% PFA. Then, whole brains were collected and fixed with a 4% PFA for 24 h, and processed for preparing paraffin sections. The slides were then incubated with the corresponding primary antibody followed by HRP-labeled secondary antibody and stained with fluorescently labeled tyramide. Repeat steps to incubate remaining primary antibody, secondary antibody, and tyramide signal amplification (TSA) marker, and microwave to remove the primary and secondary antibodies. The nuclei were stained with DAPI for 10 min. For detection, slices were placed under the scanner (Pannoramic MIDI, 3Dhistech, Budapest, Hungary) for image acquisition. CaseViewer 2.4

was used to observe the images. Five different parts of each slice were randomly selected, and Image Pro Plus 6.0 was used to count positive cells.

For the flow cytometry analysis of the tumor immuno-microenvironment, tumor tissues were removed 3 days after the last injection of different treatments (1.5 or 3 mg/kg on an aPD-L1 or PTX basis). For T cell analysis, mouse lymphocyte separation medium was used for dissociation. After sorting with anti-CD45 magnetic beads, the tumor cells were stained with anti-CD3, anti-CD8, anti-CD4, anti-CD25, anti-CD127, anti-PD-1, anti-TIM-3, and anti-Granzyme B antibodies for 30 min, fixed and permeabilized using a Foxp3/Transcription Factor Staining Buffer Set (eBioscience), then stained with anti-IFN-γ antibody for 30 min. Thereafter, the stained cells were analyzed by flow cytometry (FACSCelesta, Becton, Dickinson and Company, New Jersey, USA) for CD8 and Treg cell populations (collecting $1 \times 10^4$ events for analysis).

For DC analysis, mouse lymphocyte separation medium was used for dissociation. After sorting with anti-CD45 magnetic beads, the tumor cells were stained with anti-CD11c, anti-MHC II, anti-CD80, and anti-CD86 antibodies for 30 min. After that, flow cytometry (FACSCelesta, Becton, Dickinson and Company, New Jersey, USA) was performed on the stained cells to analyze the mature DC population (collecting $1 \times 10^4$ events for analysis).

For the CRT expression, cell suspensions were incubated with anti-CRT polyclonal antibody (1:100 dilution) for 1 h at 37 °C. Then, cells were washed with 1% BSA buffer twice and stained with Alexa Fluor 647 labeled goat antirabbit IgG (H&L) highly cross-adsorbed secondary antibody (1:200 dilution) for 30 min at 4 °C. Thereafter, the stained cells were analyzed by flow cytometry (NovoCyte Flow Cytometer, ACEA Biosciences, California, USA) for CRT-positive cell populations.

For the macrophage analysis, mouse lymphocyte separation medium was used for dissociation. The tumor cells were stained with anti-CD11b, anti-F4/80 and anti-CD86 antibodies for 30 min and stained with CD206 after fixation. After that, flow cytometry (FACS-Celesta, Becton, Dickinson and Company, New Jersey, USA) was performed on the stained cells to analyze the macrophage population (collecting $1 \times 10^4$ events for analysis).

For the MDSC analysis, mouse lymphocyte separation medium was used for dissociation. The tumor cells were stained with anti-CD11b, anti-Gr-1, anti-Ly6C, and anti-Ly6G antibodies for 30 min. After that, flow cytometry (FACSCelesta, Becton, Dickinson and Company, New Jersey, USA) was performed on the stained cells to analyze the MDSC population (collecting $1 \times 10^4$ events for analysis).

To analyze effector memory T cells, mice treated with A2-APM were selected on day 45 post-inoculation. Brain, dLNs, and spleen were removed and stained with anti-CD3, anti-CD8, anti-CD44, and anti-CD62L antibodies. Afterwards, the stained cells were subjected to flow cytometry analysis (FACSCelesta, Becton, Dickinson and Company, New Jersey, USA) (collecting $1 \times 10^4$ events for analysis).

All antibodies were used following the manufacturer's instructions. The fluorochromes conjugated on the antibody were exactly matched to the same fluorochrome channel.

### In vivo depletion experiments
Depletion of CD8+ and CD4+ T cells was performed by i.p. injection of respective in vivo antibodies in 200 µL PBS per mouse (all from BioXCell): anti-CD4 (Clone, YTS191; 1000 µg/mouse); anti-CD8 (Clone, 2.4; 500 µg /mouse) injected on days 7, 12, 18, and 23. At the experiment endpoint, depletion was verified at the brain tumor site using flow cytometry.

### Toxicity assessment
Healthy C57BL/6 mice were randomly divided into 8 groups ($n = 4$ in each group, day 0). The different treatment formulas were administered intravenously on day 0 and day 7 (1.5 or 3 mg/kg on an aPD-L1 or

PTX basis). On day 30, the mice were sacrificed, and the organs (heart, liver, spleen, lung, and kidneys) were harvested, followed by processing for preparing paraffin sections. The manufacturer's instructions were performed for H&E staining, and images were obtained by ECLIPSE Ti2-U microscope (Nikon Corporation, Tokyo, Japan). Blood samples for each drug were collected from 4 anesthetized mice through a vein behind the eye. Whole blood specimens were left at room temperature for 2 h and then centrifuged at 2–8 °C for 15 min at 3000 rpm, and the supernatant was removed for immediate detection. The concentration of AST, ALT, BUN, and creatinine in plasma is quantified by Chemray 800 (Rayto, Shenzhen, China).

To detect whether the prolonged blood circulation time might increase their distribution in other organs, Cy7.5-labeled aPD-L1 was used to prepare fluorescent A2-APM, then saline and A2-APM (aPD-L1 equivalent dose 1.5 mg/kg) were intravenously given to GBM mice. The mice were sacrificed at 90 min, 24 h, and 96 h post-injection to extract the heart, liver, spleen, lung, kidneys, and brain for ex vivo fluorescence imaging.

For the detection of cerebrovascular conditions, tumor-bearing mice were randomized, and the drugs were injected intravenously via the tail vein. Saline, free PTX, APM, A2-APM (1.5 or 3 mg/kg on an aPD-L1 or PTX basis) were administered, and the blood vessels were monitored by full-field laser perfusion imager (Moor Instruments Ltd, Millwey, UK) at 0, 4, 24, 48, and 96 h post-injection. Meanwhile, the mice were sacrificed at 0, 4, 24, 48, and 96 h post-injection to extract the brain for blood vessel immunofluorescence.

To investigate whether drugs cause edema, tumor-bearing mice were randomized, and the drugs were injected intravenously via the tail vein. Saline, free PTX, APM, A2-APM (1.5 or 3 mg/kg on an aPD-L1 or PTX basis) were administered. Then, the mice were sacrificed at 0, 4, 24, 48, and 96 h post-injection, and the brains were extracted, and the wet brains were weighed immediately and then dried in an oven at 60 °C for 24 h to measure the weight of dry brains. The water content of the brain % was calculated by the following formula (3):

$$\frac{Wet\,weight - dry\,weight}{Wet\,weight} \times 100\% \quad (3)$$

To evaluate whether A2-APM can effectively mitigate lymphodepletion, GL261 tumor-bearing mice were randomized, and the drugs were injected intravenously via the tail vein. Saline and free PTX, A2-APM (5 mg/kg PTX) were administered on days 14, 16, and 18. For the flow cytometry analysis, blood, lymph nodes, and spleen were collected on day 21. A mouse lymphocyte separation medium was used for dissociation. The cells were stained with anti-CD45, anti-CD3, anti-CD8, anti-PD-1, and anti-TIM-3, antibodies for 30 min. Thereafter, the stained cells were analyzed by flow cytometry (FACSCelesta, Becton, Dickinson and Company, New Jersey, USA) for CD8 and Treg cell populations (collecting $1 \times 10^4$ events for analysis).

### Statistical analysis
All statistical analyses were performed on Graphpad Prism 8.0, SPSS 26.0 software was used for statistical analysis of all data. Data were expressed as the mean and SD. $P < 0.05$ was considered to be statistically significant. the $P$ values were determined with one-way analysis of variance (ANOVA) as indicated in the figure legends. For comparisons of two groups, the $P$ values were determined with $t$-test, as indicated in the figure legends. Survival was analyzed using Kaplan−Meier curves and log-rank (Mantel−Cox) tests.

### Reporting summary
Further information on research design is available in the Nature Portfolio Reporting Summary linked to this article.

## Data availability

The publicly available data used in Supplementary Fig. 1a, b and e are available in the TCGA database (survminer package version 0.4.9, R version 4.3.1, https://xenabrowser.net/datapages/). The authors declare that all data supporting the findings of this study are available within the paper and its Supplementary Information files. Source data are provided with this paper.

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

## Acknowledgements

This study was supported by the National Natural Science Foundation of China (NSFC, No. 82330060, 92059202, 92359304 and 61821002 for S.J., 82172010 for J.X. and 82302272 for X.Y.), the National Key R&D Program of China (2021YFF0501504 and 2021YFA1101304 for S.J.), the Natural Science Foundation of Jiangsu Province (BK20220831 for X.Y.), the National University of Singapore (NUHSRO/2020/133/Startup/08, NUHSRO/2023/008/NUSMed/TCE/LOA, NUHSRO/2021/034/TRP/09/ Nanomedicine for X.C.), National Medical Research Council (MOH-001388-00, CG21APR1005 for X.C.), Singapore Ministry of Education (MOE-000387-00 for X.C.), and National Research Foundation (NRF-000352-00 for X.C.). We thank Prof. Cong Li, Dr. Cong Wang, Zhi Li, Fangzheng Tian, Zhenyu Hou, Jue Wang, Yinlu Wang and Huiyan Ding for their assistance in the experiments. We thank Prof. Xiuxing Wang and Daqi Li for providing the G7 and WL1 cell lines.

## Author contributions

Z.Z. conducted the synthesis, characterization, cell studies, animal experiments, and data analysis, and wrote the manuscript. X.X. helped in the construction of an orthotopic brain tumor model and animal experiments. J.D., X.C. (Xin Chen), Y.X., X.Y., and J.Z. contributed to the data analysis and discussed the data. X.C. (Xiaoyuan Chen), J.X., and S.J. commented on and revised the manuscript, and supervised the project.

## Competing interests

The authors declare no competing interests.
