## [Peer Review File · Nature Communications]

REVIEWER COMMENTS

Reviewer #1 (Remarks to the Author): with expertise in nanotechnology for cancer therapy

In this article, Zhang and colleagues present a method for treating glioblastoma (GBM) by co-encapsulating anti-PDL1 antibodies and paclitaxel in redox-sensitive micelles. They demonstrated the effectiveness of the micelles in two different glioma models to show efficacy against primary and postoperative recurrent tumors. They also examined the regulation of the immune microenvironment of GBM by micelles, which results in accumulation and proliferation of CD8+ T cells within the tumor. Sufficient memory T cells are elicited within the tumor, draining lymph nodes, and spleen to suppress postoperative recurrent and rechallenged tumors. This is a novel and meaningful study as GBM is universally fatal, and the work is well executed with convincing results. However, some minor points in the manuscript should be considered before publication.

1. Why was PTX selected for this study among various chemotherapy drugs?
2. Please add statistical data points to Figure 2a/b and Figure S11e.
3. As GBM typically has a mean survival of 25 days, maximum 30 days, please provide a comment on this discrepancy.
4. Clinically, glioblastoma is an extremely malignance, please discuss more in the discussion section about the prospects and obstacles for clinical translation of the micelle.
5. Please carefully check the article for typos or formatting errors.
6. In the Figure legends in Fig.1 b-c, it is indicated that the number of experimental samples (n) is 3, while no error bar was shown.

Reviewer #2 (Remarks to the Author): with expertise in glioblastoma, neuro-immunology

This is a comprehensive manuscript that assess a novel NP delivery platform for PD-L1 and PTX agents. It extensively describes the NP mode of action and its in vivo characteristics in a glioma model. The manuscript is overall well written and the in vivo characteristics including the memory effect are impressive.

Major points

- bio distribution analysis are crucial for these kind of NP platforms. It would be very informative to use flow cytometry in addition to better understand the cellular biodistribution of the NP within the tumor and peripheral organs including blood cells.

Also there have been previous studies that performed biodistribution analysis of NP formulation that quantify NP biodistribution on the cellular level (10.1073/pnas.1609397113; doi: 10.1038/s41467-023-36321-6)

- checkpoint inhibition also likely acts systemically on peripheral immune cell priming and exhaustion and not necessarily in the TME. What is the argument that checkpoint inhibitor delivery is actually necessary in the TME (and not just systemically). Please discuss.

- Myeloid cells are major determinants of glioma TME immunotherapy (e.g. doi: 10.1038/s41467-023-36321-6). The effect of the described NP platform could be discussed more in the context of the myeloid compartment.

- PTX is a rather old cytotoxic drug. Is the platform also versatile to include other drug formulations? Please discuss.

Minor suggestions:

- Biodistribution analysis should be shifted to the main figures as this is an important aspect of the platform

Reviewer #3 (Remarks to the Author): with expertise in glioblastoma, immunotherapy

The manuscript by Zhang et al reports on the development of a redox-sensitive micelle that co-encapsulates an immune checkpoint inhibitor (aPD-L1) and paclitaxel. This nanoparticle drug delivery system targets the LRP1 on endothelial cells, enhancing the delivery of both aPD-L1 and paclitaxel into the tumor microenvironment. In GBM tumor-bearing mouse models, this strategy resulted in a significantly higher accumulation of these agents compared to a free aPD-L1 group. The engineered micelles effectively targeted GBM, halted tumor growth, prevented postoperative GBM recurrence, and induced long-term immune memory, protecting against tumor rechallenge. The authors suggest that their system could improve the low response rate of IDCB therapy for GBM, as demonstrated by its potent effectiveness in preclinical models. Overall, this study investigates the potential of the approach, which is of high interest and timely. However, there are concerns that should be addressed before this work can be considered for publication. Please find specific comments below:

Major comments:

Although the co-encapsulation approach is performed, it would be beneficial to discuss why this is a novel or superior method for drug delivery compared to other methods. The authors could further elaborate on how their findings contribute to the existing body of knowledge and how they compare to previous studies. This could provide a clearer picture of the novelty and significance of their work.

Several studies have demonstrated low PD-L1 expression in the tumor microenvironment of human GBM. Though the authors show that aPD-L1 binds to GL261, it would be important to demonstrate in PDX cell lines that the human antibody against PD-L1 is binding to GBM cells or tumor-associated macrophages from human tissues to provide a justification for using an anti-PD-L1 therapy.

The authors performed cytotoxicity assays with free paclitaxel and PTX-contained micelles in GL261 and GSCs. The results derived from these assays are shown in the main figures for GL261 but not for the GSCs which are more clinically relevant for GBM patients considering the big genomic and phenotypic differences between GL261 and human gliomas. Thus, it is recommended to show the cytotoxicity assay and ICD results derived from GL261 and the GSCs in the main figures.

The rationale for the use of paclitaxel as opposed to other chemotherapeutic drugs to induce immunogenic cell death is missing. Where there any comparative experiments in this context? There is some clinical trial literature on the use of chemotherapy to enhance response to immunotherapy. This should be considered and properly cited.

It is important to provide a paclitaxel concentration context. Are the concentrations achieved leading to immunogenic cell death in human glioma cell lines and PDX?

For Figures 2g and 2h, it is unclear, and it is important to show how the gating was done to arrive to the conclusion that there are more CD11c⁺ MHC II⁺ CD80⁺ CD86⁺ cells. For instance, the authors mention that “the expression of CD11c, MHC II, CD80, and CD86 was similar to that of the free PTX treated group, but more than a 10-fold increase as compared to that of the blank micelles and AM treated groups.” but there is no comparison of CD11c and MHC II expression between groups. It is recommended to perform the comparison of the expression of each marker to determine that there are mature DCs.

The BBB in vitro system in Figures 3a and 3b has several limitations. The BBB is normally constituted by not just endothelial cells, but astrocytic end-feet, microglia, pericytes, and basement membrane. The findings cannot be extrapolated to an in vivo model or a phenomenon occurring in GBM patients.

Figure 4e needs a group of mice treated with a combination of free aPD-L1 and paclitaxel to compare with the group that receive A2-APM. Without this group, it is not possible to assess the additive effect of encapsulating both drugs for delivery into the brain. Same comment applies to Supplementary Figure 17.

Would recommend moving Supplementary Figure 17 to the main Figure to show efficacy in two separate intracranial model.

Did the A2-APM lead to survival benefit in the post- resection model ? This is not clearly shown.

It is recommended to refrain from using the terms M1/M2 to refer to tumor-associated macrophages as this can mislead the real phenotype of both macrophages and microglia described by single-cell RNA-sequencing analyses where the M1/M2 has not been demonstrated. Instead, the function of the markers identified by flow cytometry can be described for the experimental group. More compelling demonstration of the macrophage phenotype and how this is modulated by their treatment, would require co-culture with T-cells to determine the effect of these macrophages on T-cell activity.

Appropriate statistical analyses need to be performed for several figures. For instance, it is recommended to use one-way ANOVA with multiple comparison post hoc tests in Fig. 1d, 1e, 2b, 2d, 2e, 2f, 2g, 3f, 5c, and 5e. A different statistical test is recommended to be used in outcomes involving 2 variables such as two-way ANOVA. The inclusion of these tests would strengthen the argument that observed differences are significant and not due to random chance.

Minor comments

- It is not clear the description of what is A2-APM in the introduction and the first section of the results.
- There are no P values in Fig 4e.

Reviewer #4 (Remarks to the Author): with expertise in nanotechnology for cancer therapy, glioblastoma

In this manuscript, Zhang et al have developed a redox-sensitive polymer micelle called A2-APM, which demonstrates the co-encapsulation of anti-PD-L1 antibody and PTX. By employing angiopep-2 peptide decoration, A2-APM exhibits the ability to traverse the blood-tumor barrier (BTB). Within the reductive microenvironment of glioblastoma (GBM), the anti-PD-L1 antibody and PTX are selectively released. Notably, the released PTX elicits immunogenic cell death in glioma cells, thereby enhancing immunogenicity and facilitating a localized immune response. Furthermore, the released anti-PD-L1 antibody serves to revitalize infiltrated T cells. Despite the authors' comprehensive investigations supporting the strong anti-GBM efficacy, the manuscript suffers from substandard writing quality and lacks the requisite novelty to warrant publication in this journal. Moreover, several concerns remain unaddressed and require further attention.

1. The paper exhibits limited novelty in regards to the designed micelles, as the authors have not adequately demonstrated the essentiality of combining the anti-PD-L1 antibody and PTX. Moreover, the manuscript lacks a coherent structure that effectively highlights the pivotal scientific problem and the unique contributions of the current study.
2. To elucidate the underlying mechanisms driving the antitumor activity, a depletion study is imperative to investigate the specific contribution of immune cells in this context.
3. In the discussion section, the authors highlighted the potential of chemotherapeutic agents inducing immunogenic cell death (ICD) to augment the efficacy of immune checkpoint blockade (ICB) through the activation of T cells. To validate this hypothesis, it is crucial to confirm alterations in immune markers (such as PD-1, TIM-3, IFN- γ , GzmB, et al) within T cells via flow cytometry analysis.
4. The authors employed t-tests instead of employing one-way analysis of variance (ANOVA) comparisons to assess the significance among multiple groups. As the data in this paper are analyzed, it is essential to meticulously examine the statistical significance and draw appropriate conclusions by utilizing the appropriate comparison method.
5. In the introduction section, the authors made a claim regarding the potential lymphodepletion resulting from the systemic administration of chemotherapeutic agents. However, the current studies lack data supporting this assertion. It is imperative for the authors to provide evidence demonstrating whether the constructed micelles, through the systemic administration of PTX, can effectively mitigate lymphodepletion.
6. HMGB1, a cytokine closely associated with inflammatory responses, predominantly localizes within the nucleus. Upon chemotherapeutic treatment, HMGB1 can undergo translocation from the nucleus to the extracellular matrix. Surprisingly, the expression of HMGB1 was observed to be low in the PBS group but high in the APM group within the nucleus, contrary to the expectations based on Supplementary Figure 10. This discrepancy contradicts the data presented in Figure 2e.
7. The release of PTX within the reductive microenvironment of GBM facilitates the induction of ICD in glioma cells. However, it is crucial to consider the potential impact of PTX release on the activity of

infiltrated immune cells, including T cells and dendritic cells (DCs). It is necessary to provide an explanation for how PTX-induced ICD might affect the functionality of these immune cells.

8. In line 237-238 on page 9, the author asserted that the PEG-shell effectively hindered the off-target distribution of free drugs, preventing their accumulation in healthy organs. However, it is essential to verify if any cited references support this claim. Additionally, the description of this mechanism appears inaccurate. In reality, the PEG-shell serves to prolong the circulation time of coated nanoparticles and impedes their recognition and phagocytosis by the mononuclear phagocyte system (MPS). As a result, the distribution of nanoparticles in the liver and spleen is reduced. However, it is important to note that the prolonged blood circulation time might increase their distribution in other organs such as the kidney and heart.

9. The G422 cell lines were derived by intracranial injection of methylcholanthrene in Kunming mice, which are frequently utilized as hosts. However, it is necessary to clarify the rationale behind employing C57 mice for constructing the G422 model. Furthermore, it is crucial to address whether the authors took into account the potential immunogenicity associated with the use of C57 mice in this context.

10. In Supplementary Figure 19, a black signal was observed in day21 in A2-APM group. Please explain this phenomenon.

11. In Supplementary Figure 25, the signals of CD3 seem to be less than those of CD4 and CD8. Please explain this. In addition, CD4 signals seem to be not overlapped by CD3 signals.

12. In Supplementary Figure 29, the authors made reference to the exposure of calreticulin (CRT) on tumor cells. However, a noteworthy concern arises as the authors did not utilize antibodies or fluorescence labeling to specifically identify the tumor cells. It is crucial to thoroughly evaluate this description and ensure the appropriateness of the experimental approach. To address this concern, the authors are encouraged to employ tumor cells expressing green fluorescent protein (GFP) or red fluorescent protein (RFP) to conduct the corresponding experiments, thereby enabling precise visualization and characterization of the tumor cell population of interest.

Comments:

Reviewer #1 (Remarks to the Author): with expertise in nanotechnology for cancer therapy

In this article, Zhang and colleagues present a method for treating glioblastoma (GBM) by co-encapsulating anti-PDL1 antibodies and paclitaxel in redox-sensitive micelles. They demonstrated the effectiveness of the micelles in two different glioma models to show efficacy against primary and postoperative recurrent tumors. They also examined the regulation of the immune microenvironment of GBM by micelles, which results in accumulation and proliferation of CD8+ T cells within the tumor. Sufficient memory T cells are elicited within the tumor, draining lymph nodes, and spleen to suppress postoperative recurrent and rechallenged tumors. This is a novel and meaningful study as GBM is universally fatal, and the work is well executed with convincing results. However, some minor points in the manuscript should be considered before publication.

1. Why was PTX selected for this study among various chemotherapy drugs?

Re: Thanks for your professional comment. The reasons for choosing paclitaxel (PTX) in this study were sophisticated, the main reason is that PTX has shown an extremely strong anti-tumor effect in a wide range of cancer subtypes. Moreover, PTX can induce a significant immune cell death (ICD) effect, which enhanced the immunotherapy efficacy of the nano-micelle (A2-APM).

PTX is one of the most widely used anti-tumor drugs due to its significant efficacy in a wide range of cancers [1-4]. However, its application in the treatment of GBM was limited after the failure of early clinical trials because of its poor brain penetration [5,6]. The non-invasive and low-toxicity delivery of PTX to the brain remains a challenging area of research [7]. In this study, we employed nano-micelle encapsulated PTX and modified it with the A2 peptide on the surface. This modification enabled the nano-micelle to traverse the blood-brain barrier (BBB), providing a promising method for delivering PTX to GBM.

PTX is highly cytotoxic and induces a significant ICD effect [8]. In our study, PTX-induced ICD effect activated an immune response while aPD-L1 blocked PD-L1 in the TME to enhance the antitumor effect, which is conducive to the exploration of PTX as an immunoadjuvant in GBM, a class of immunosuppressed tumors. On the other hand, PTX exhibits a negative charge and is connected to the PEG-PLL polymer backbone applied in this study by electrostatic interaction, which avoids the complicated chemical synthesis process and can better preserve the function and structure of PTX itself.

Reference:

- [1] Scribano, Christina M et al. Chromosomal instability sensitizes patient breast tumors to multipolar divisions induced by paclitaxel. *Science translational medicine* vol. 13,610 (2021): eabd4811.
- [2] Abu Samaan, T. M., Samec, M., Liskova, A., Kubatka, P. & Büsselberg, D. Paclitaxel's mechanistic and clinical effects on breast cancer. *Biomolecules* 9, 789 (2019).
- [3] Kampan, N. C., Madondo, M. T., McNally, O. M., Quinn, M., & Plebanski, M. Paclitaxel and its evolving role in the management of ovarian cancer. *Biomed. Res. Int.* 2015, 413076 (2015).
- [4] Blair, H. A. & Deeks, E. D. Albumin-bound paclitaxel: a review in non- small cell lung cancer. *Drugs* 75,2017–2024 (2015).
- [5] Fountzilias, G et al. "Radiation and concomitant weekly administration of paclitaxel in patients with glioblastoma

multiforme. A phase II study.” Journal of neuro-oncology vol. 45,2 (1999): 159-65.

[6] Elinzano, Heinrich et al. “PPX and Concurrent Radiation for Newly Diagnosed Glioblastoma Without MGMT Methylation: A Randomized Phase II Study: BrUOG 244.” American journal of clinical oncology vol. 41,2 (2018): 159-162

[7] Sonabend, Adam M et al. “Repeated blood-brain barrier opening with an implantable ultrasound device for delivery of albumin-bound paclitaxel in patients with recurrent glioblastoma: a phase 1 trial.” The Lancet. Oncology vol. 24,5 (2023): 509-522.

[8] Lau, Tat San et al. “Paclitaxel Induces Immunogenic Cell Death in Ovarian Cancer via TLR4/IKK2/SNARE-Dependent Exocytosis.” Cancer immunology research vol. 8,8 (2020): 1099-1111. doi:10.1158/2326-6066.CIR-19-0616

2. Please add statistical data points to Figure 2a/b and Figure S11e.

Re: Thanks for this suggestion. We have revised Figure2a, b and Figure S11e (new figure2i) as the following:

Fig. 2 Cytotoxicity and ICD induction in GL261 cells and GSCs of APM. a, b Cytotoxicity and apoptosis results of GL261 cells after receiving different treatments (n = 3 biologically independent samples, concentration unit: μg/mL, incubation time: 24h). **i** Cytotoxicity results of GL261 cells after receiving different treatments 24 hours (concentration unit: μg/mL). All statistics are expressed as mean ± SD, * $P < 0.05$, ** $P < 0.01$, *** $P < 0.001$ and **** $P < 0.0001$. Statistical significance was calculated by one-way ANOVA with Fisher’s LSD test.

3.As GBM typically has a mean survival of 25 days, maximum 30 days, please provide a comment on this discrepancy.

Re: Thanks for your professional comment. Our GL261 model has a median survival of 31 days, which is not very different from other studies (Kinoh, Hiroaki et al. ACS Nano vol. 14,8

(2020): 10127-10140.figure 3B) (Li, Yujie et al. *Advanced Science* vol. 8,17 (2021): e2004381.figure 5); our data are within a reasonable range of median survival. In addition, differences in the state of the mice at the time of cell implantation may also contribute to the small differences in final median survival.

4. Clinically, glioblastoma is an extremely malignancy, please discuss more in the discussion section about the prospects and obstacles for clinical translation of the micelle

Re: Thanks for your suggestion.

All the materials that made up the nano-micelles and the drugs delivered have been in clinical use or undergoing preclinical trials, and their characteristics, such as biocompatibility, have been tested [1-3]. Our experiments have demonstrated the biosafety and therapeutic efficacy of the nano-micelles, so the nanoformula has the potential for clinical application.

The materials used in our study, aPD-L1 and PTX have demonstrated manageable toxicity in early clinical trials [1,2]. PEG-PLL is commercially available, and some of the nano-micelles represented by Genexol® PM (PTX-encapsulated nano-micelles) are already clinically available [3].

In our experiments, A2-APM demonstrated good stability (Supplementary Figure 5), and a good safety profile in terms of histology and hematology. It did not cause vascular abnormalities in the brain or cerebral edema (Supplementary Figure 25-31).

Of course, acute toxicity tests, long-term dosing tests, genotoxicity tests, and reproductive toxicity tests should be performed and the pharmacokinetics of A2-APM should be thoroughly evaluated before clinical translation.

The description has been added to the discussion section (page 22, line 601-608).

Reference:

[1] Reardon, David A et al. "Treatment with pembrolizumab in programmed death ligand 1-positive recurrent glioblastoma: Results from the multicohort phase 1 KEYNOTE-028 trial." *Cancer* vol. 127,10 (2021): 1620-1629. doi:10.1002/encr.33378

[2] Sonabend, Adam M et al. "Repeated blood-brain barrier opening with an implantable ultrasound device for delivery of albumin-bound paclitaxel in patients with recurrent glioblastoma: a phase 1 trial." *The Lancet. Oncology* vol. 24,5 (2023): 509-522.

[3] Nam, So Hyun et al. "Safety and Tolerability of Weekly Genexol-PM, a Cremophor-Free Polymeric Micelle Formulation of Paclitaxel, with Carboplatin in Gynecologic Cancer: A Phase I Study." *Cancer research and treatment* vol. 55,4 (2023): 1346-1354. doi:10.4143/crt.2022.1436

5. Please carefully check the article for typos or formatting errors.

Re: Thanks for your careful checks. We have carefully proofread the manuscript and corrected the typos and grammatical errors.

6. In the Figure legends in Fig.1 b-c, it is indicated that the number of experimental samples (n) is 3, while no error bar was shown.

Re: Thanks for pointing this out. We have examined data from three independent samples, and now add the experimental statistics about figure 1b, c to the supplementary materials

(Supplementary Figure 3).

Supplementary Figure 3. Size distribution and morphology of APM by DLS and TEM a DLS analysis of the mean particle size distribution of APM. **b** TEM image of APM. Scale bar = 50 nm. n = 3 biologically independent samples.

Reviewer #2 (Remarks to the Author): with expertise in glioblastoma, neuro-immunology

This is a comprehensive manuscript that assess a novel NP delivery platform for PD-L1 and PTX agents. It extensively describes the NP mode of action and its *in vivo* characteristics in a glioma model. The manuscript is overall well written and the *in vivo* characteristics including the memory effect are impressive.

Major points

1. bio distribution analysis are crucial for these kind of NP platforms. It would be very informative to use flow cytometry in addition to better understand the cellular biodistribution of the NP within the tumor and peripheral organs including blood cells. Also there have been previous studies that performed biodistribution analysis of NP formulation that quantify NP biodistribution on the cellular level (10.1073/pnas.1609397113; doi: 10.1038/s41467-023-36321-6)

Re: Thanks to your suggestion, we have added an analysis for the biodistribution of nano-micelles in GL261 GBM bearing-mouse using flow cytometry.

Alexa647-labeled aPD-L1 was used to prepare A2-APM. A2-APM-Alexa647 was injected into GL261 tumor mice, and after 90 minutes, blood, brain tissue (divided into two hemispheres), heart, liver, spleen, lungs, kidneys, and lymph nodes were removed, and the tissues were snipped, digested, and then flow-stained. The results revealed a high proportion of monocyte-derived myeloid cells that conjunct with A2-APM in tumor hemisphere, and the tumor microglia, and the monocyte-derived myeloid cells and macrophage in blood, liver, lung, kidney, heart revealed less nano-micelle uptake (Fig.3i, j, k and Supplementary Figure 19).

Fig. 3 A2-APM effectively crossed *in vivo* and *in vitro* BTB models. i Representative histological image shown A2-APM accumulation in the TME of GL261 glioma. n = 3. Scale bar = 100 μ m. **j, k** Biodistribution analysis of A2-APM-Alexa647 90 min after a single, intravenous dose as assessed by flow cytometry of T cells ($CD45^+CD3^+$), NK cells

(CD45⁺NK1.1⁺), microglia (CD45⁺CD11b^{intermediate}), monocyte-derived myeloid cells (CD45⁺CD11b^{high}Ly6C⁺) and macrophages (CD45⁺CD11b^{high}F4/80⁺) in the tumor-bearing hemisphere, contralateral hemisphere and blood. n = 5 A2-APM-Alexa647 mice and 3 saline-treated animals. All statistics are expressed as mean ± SD. Statistical significance was calculated by independent sample *t-test* in (j).

Supplementary Figure 19. Distribution of aPD-L1 in GL261 GBM-bearing mice by flow cytometry. Biodistribution analysis of A2-APM-Alexa647 90min after a single, intravenous dose as assessed by flow cytometry of T cells (CD45⁺CD3⁺), NK cells (CD45⁺NK1.1⁺), microglia (CD45⁺CD11b^{intermediate}), monocyte-derived myeloid cells (CD45⁺CD11b^{high}Ly6C⁺) and macrophages (CD45⁺CD11b^{high}F4/80⁺) in the liver, spleen, lung, kidney, heart and lymph nodes. n = 5 A2-APM-Alexa647 mice and 3 saline-treated mice.

Description of the results have been added to the results section (page11-12, line 307-315) as:

In addition, the cellular biodistribution of A2-APM within tumors and peripheral organs, including blood cells, would be very informative^{42,43}. Histological analysis showed that the tumor site specifically took up Alexa647 dye labeled A2-APM (A2-APM-Alexa647), and nano-micelles incorporation was detected in G1261-GFP⁺ tumor cells (Fig. 3i). Flow cytometry results revealed a high proportion of monocyte-derived myeloid cells that conjunct with A2-APM in tumor hemisphere, whereas the tumor microglia, and the monocyte-derived myeloid cells and macrophage in liver, blood, lung, kidney, heart revealed less nano-micelle uptake (Fig. 3j, k and Supplementary Fig.19).

2.checkpoint inhibition also likely acts systemically on peripheral immune cell priming and exhaustion and not necessarily in the TME. What is the argument that checkpoint inhibitor delivery is actually necessary in the TME (and not just systemically). Please discuss.

Re: Thank you for your questions.

PD-L1 is predominantly expressed in tumor cells and immunosuppressive cells in the TME, delivery of checkpoint inhibitors (ICIs) to the TME can increase the concentration of ICIs in the TME to improve the bioavailability of ICIs [1-3], while reducing systemic side effects caused by drug leakage [4,5].

In this study, the nano-micelles were decorated with targeted angiopep-2 (A2) peptide, delivered PD-L1 antibody to TME and released it in a specific environment, which overcome the difficulty of getting free antibodies into the brain, and precisely target PD-L1 in TME and

improve the local immunosuppressive microenvironment. Meanwhile, the released PTX induced ICD effects, stimulated DC cell maturation and migration, activated the immune system, promoted CD8⁺ T cell activation and proliferation, and enhanced the killing effect of CD8⁺ T cells.

Reference:

[1] Vimalathas, Gayaththri, and Bjarne Winther Kristensen. "Expression, prognostic significance and therapeutic implications of PD-L1 in gliomas." *Neuropathology and applied neurobiology* vol. 48,1 (2022): e12767. doi:10.1111/nan.12767;

[2] Wang, Xin et al. "Challenges and potential of PD-1/PD-L1 checkpoint blockade immunotherapy for glioblastoma." *Journal of experimental & clinical cancer research: CR* vol. 38,1 87. 18 Feb. 2019, doi:10.1186/s13046-019-1085-3;

[3] Yang, Tao et al. "Conjugation of glucosylated polymer chains to checkpoint blockade antibodies augments their efficacy and specificity for glioblastoma." *Nature biomedical engineering* vol. 5,11 (2021): 1274-1287. doi:10.1038/s41551-021-00803-z).

[4] Vargason, Ava M et al. "The evolution of commercial drug delivery technologies." *Nature biomedical engineering* vol. 5,9 (2021): 951-967. doi:10.1038/s41551-021-00698-w

[5] Zhang, Xiaoqiong et al. "Cell microparticles loaded with tumor antigen and resiquimod reprogram tumor-associated macrophages and promote stem-like CD8⁺ T cells to boost anti-PD-1 therapy." *Nature communications* vol. 14,1 5653. 13 Sep. 2023, doi:10.1038/s41467-023-41438-9

3. Myeloid cells are major determinants of glioma TME immunotherapy (e.g. doi: 10.1038/s41467-023-36321-6). The effect of the described NP platform could be discussed more in the context of the myeloid compartment.

Re: During glioma development, blood-derived myeloid cells increasingly infiltrate the tumor, which is a key driver of immunosuppressive programs [1, 2]. Thanks for your suggestion.

We analyzed the flow cytometry for myeloid cells, including MDSCs and macrophages. To clarify the alterations of macrophages and MDSCs in GBM tumors, the flow cytometry of macrophage and MDSCs performed after three days of treatment with A2-APM. The increased percentage of CD86⁺F4/80⁺ cells in CD11b⁺CD45⁺ cells and decreased percentage of CD206⁺F4/80⁺ cells in CD11b⁺CD45⁺ cells were observed in A2-APM group (Fig.6c, Supplementary Fig. 43a and 44), and the number of either Gr-1⁺CD11b⁺ cells or Ly6C⁺Ly6G⁻ (monocytic-MDSCs, M-MDSCs) and Ly6C⁺Ly6G⁺ (polymorphonuclear-MDSCs, PMN-MDSCs) cells was minimized in the A2-APM treated group (Fig. 6d, Supplementary Fig. 43b and 45).

Reference:

[1] De Leo, Alessandra et al. "Myeloid Cells in Glioblastoma Microenvironment." *Cells* vol. 10,1 18. 24 Dec. 2020, doi:10.3390/cells10010018

[2] Ravi, Vidhya M et al. "T-cell dysfunction in the glioblastoma microenvironment is mediated by myeloid cells releasing interleukin-10." *Nature communications* vol. 13,1 925. 17 Feb. 2022, doi:10.1038/s41467-022-28523-1

Fig. 6 c Quantification of tumor-infiltrating IFN γ^+ , Granzyme B⁺ (GrzB⁺), PD-1⁺ and TIM-3⁺ T cells three days after two treatments with saline, free aPD-L1, free PTX, A2-AM, A2-PM, APM and A2-APM, n = 3.

Supplementary Figure 43. A2-APM treatment resulted in pro-inflammatory transformation of macrophages and decreased myeloid-derived suppressor cells in GL261 tumor mice. **a** Representative flow cytometric contour plots of tumor-infiltrating CD86⁺F4/80⁺ and CD206⁺F4/80⁺ cells three days after two treatments with saline, free aPD-L1, free PTX, A2-AM, A2-PM, APM and A2-APM. Insets: the subsets of CD86⁺F4/80⁺ cells in CD45⁺CD11b⁺ cells; the numbers indicate the percentage of CD86⁺F4/80⁺ and CD206⁺F4/80⁺ cells in CD11b⁺CD45⁺ cells after various treatments. **b** Representative flow cytometric contour plots of tumor-infiltrating Gr-1⁺CD11b⁺, Ly6C⁺ and Ly6G⁺ cells three days after two treatments with saline, free aPD-L1, free PTX, A2-AM, A2-PM, APM and A2-APM. Insets: the subsets of Gr-1⁺CD11b⁺ in CD45⁺ cells, Ly6C⁺ and Ly6G⁺ cells in CD11b⁺CD45⁺ cells; the numbers indicate the percentage of r-1⁺CD11b⁺ in CD45⁺ cells, Ly6C⁺ and Ly6G⁺ cells in CD11b⁺CD45⁺ cells after various treatments. n = 3 saline treated mice, 4 free aPD-L1, free PTX, A2-AM, A2-PM, APM and A2-APM treated mice.

Supplementary Figure 44. A2-APM treatment resulted in pro-inflammatory transformation of macrophages in G422 tumor mice. **a** Representative flow cytometric contour plots of tumor-infiltrating CD86⁺F4/80⁺ and CD206⁺F4/80⁺ cells three days after two treatments with saline, free aPD-L1, free PTX, APM and A2-APM. Insets: the subsets of CD86⁺F4/80⁺ and CD206⁺F4/80⁺ cells in CD11b⁺CD45⁺ cells; the numbers indicate the percentage of CD86⁺F4/80⁺ and CD206⁺F4/80⁺ cells in CD11b⁺CD45⁺ cells after various

treatments. **b** Quantification of tumor-infiltrating CD86⁺F4/80⁺ and CD206⁺F4/80⁺ cells three days after two treatments with saline, free aPD-L1, free PTX, APM and A2-APM. All statistics are expressed as mean ± SD, n = 4. Statistical significance was calculated by one-way ANOVA with Fisher's LSD test.

Supplementary Figure 45 A2-APM treatment decreased myeloid-derived suppressor cells in G422 tumor mice. **a** Representative flow cytometric contour plots of tumor-infiltrating Gr-1⁺CD11b⁺, Ly6C⁺ and Ly6G⁺ cells three days after two treatments with saline, free aPD-L1, free PTX, A2-AM, A2-PM, APM and A2-APM. Insets: the subsets of Gr-1⁺CD11b⁺ in CD45⁺ cells, Ly6C⁺ and Ly6G⁺ cells in CD11b⁺CD45⁺ cells; the numbers indicate the percentage of r-1⁺CD11b⁺ in CD45⁺ cells, Ly6C⁺ and Ly6G⁺ cells in CD11b⁺CD45⁺ cells after various treatments. **b** Quantification of tumor-infiltrating Gr-1⁺CD11b⁺, Ly6C⁺ and Ly6G⁺ cells three days after two treatments with saline, free aPD-L1, free PTX, A2-AM, A2-PM, APM and A2-APM. All statistics are expressed as mean ± SD, n = 3 saline treated mice, 4 free aPD-L1, free PTX, A2-AM, A2-PM, APM and A2-APM treated mice. Statistical significance was calculated by one-way ANOVA with Fisher's LSD test.

Description of the results have been added to the results section (page 16, line 448; page17, line 473-475; page 18, line484-494) as:

A2-APM treatment resulted in a pro-inflammatory shift of macrophages and decreased myeloid derived suppressor cells. Myeloid cells are one of determinants of glioma TME immunotherapy⁴³, and PTX-induced ICD increases the adjuvanticity, which can activate both

adaptive and innate antitumor immune responses⁴⁴.

Further, we confirm the results by flow cytometry, and the increased percentage of CD86⁺F4/80⁺ cells in CD11b⁺CD45⁺ cells and decreased percentage of CD206⁺F4/80⁺ cells in CD11b⁺CD45⁺ cells were observed in A2-APM group in GL261 and G422 tumor mice (Fig. 6c, Supplementary Fig. 43a and 44).

Next, the change of myeloid derived suppressor cells (MDSCs) was detected. The number of either Gr-1⁺CD11b⁺ cells or Ly6C⁺Ly6G⁻ (monocytic-MDSCs, M-MDSCs) and Ly6C⁻Ly6G⁺ (polymorphonuclear-MDSCs, PMN- MDSCs) cells was minimized in the A2-APM treated group as shown in Fig. 6d, Supplementary Fig. 43b and 45. These results imply that A2-APM is a nano-micelle that not only acts on T cells but also affects macrophages shift to pro-inflammatory phenotype, and decreased MDSCs. The protocols have been added to the method section (page 36-37, line 997-1007) as:

For the macrophage analysis, mice lymphocyte separation medium was used for dissociation. The tumor cells were stained with anti-CD11b, anti-F4/80 and anti-CD86 antibodies for 30 min and stained with CD206 after fixed. After that, flow cytometry (FACSCelesta, Becton,Dickinson and Company, New Jersey, USA) was performed on the stained cells to analyze the macrophage population (collecting 1×10^4 events for analysis).

For the MDSC analysis, mice lymphocyte separation medium was used for dissociation. The tumor cells were stained with anti-CD11b, anti-Gr-1, anti-Ly6C, and anti-Ly6G antibodies for 30 minutes. After that, flow cytometry (FACSCelesta, Becton,Dickinson and Company, New Jersey, USA) was performed on the stained cells to analyze the MDSC population (collecting 1×10^4 events for analysis).

4. PTX is a rather old cytotoxic drug. Is the platform also versatile to include other drug formulations? Please discuss.

Re: Thanks for your kind suggestion. The nano-micellar backbone referred to herein is a generalized platform that can include other drugs. PTX can be substituted with some other negatively charged drugs, such as mRNA, porphyrins, etc.

Minor suggestions:

5. Biodistribution analysis should be shifted to the main figures as this is an important aspect of the platform

Re: Thanks for your kind comments. Biodistribution analysis is crucial for nano-micelles, so we've moved the biodistribution analysis (Supplementary figure 14) to the main figure 3.

Reviewer #3 (Remarks to the Author): with expertise in glioblastoma, immunotherapy

The manuscript by Zhang et al reports on the development of a redox-sensitive micelle that co-encapsulates an immune checkpoint inhibitor (aPD-L1) and paclitaxel. This nanoparticle drug delivery system targets the LRP1 on endothelial cells, enhancing the delivery of both aPD-L1 and paclitaxel into the tumor microenvironment. In GBM tumor-bearing mouse models, this strategy resulted in a significantly higher accumulation of these agents compared to a free aPD-L1 group. The engineered micelles effectively targeted GBM, halted tumor growth, prevented postoperative GBM recurrence, and induced long-term immune memory, protecting against tumor rechallenge. The authors suggest that their system could improve the low response rate of IDCB therapy for GBM, as demonstrated by its potent effectiveness in preclinical models. Overall, this study investigates the potential of the approach, which is of high interest and timely. However, there are concerns that should be addressed before this work can be considered for publication. Please find specific comments below:

Major comments:

1. Although the co-encapsulation approach is performed, it would be beneficial to discuss why this is a novel or superior method for drug delivery compared to other methods. The authors could further elaborate on how their findings contribute to the existing body of knowledge and how they compare to previous studies. This could provide a clearer picture of the novelty and significance of their work.

Re: Thank you for your kind advice.

“Free drug + Nano” is the mostly used formula to evaluate the potentiating effect of drugs by nanomedicines. However, it is usually difficult to deliver free drugs into the target organ. Thus, overdose of the given drug may induce insufficient therapeutic efficacy or side effects.

“Nano + Nano” can be used for therapies that target different organs, such as lymph nodes and tumors, but it may also allow the drug to accumulate in excess in healthy organs due to multi-nano formulation delivery. Moreover, when “Nano + Nano” are used, the two nanoparticles may have mismatched half-lives and pharmacokinetics *in vivo*, and the onset of action in tumor tissue may be uncontrollable.

Whether it is “Free + Nano” or “Nano + Nano” delivery, clinicians must consider the sequence of drug delivery, balance the pharmacokinetics of the two drugs in tumor site, and the time window between the two drugs, which increases the clinical workload.

The “Co-encapsulation” approach is a good solution to solve the above problems, allowing the drugs to be released from the nano-micelle in a synchronized manner to achieve the optimal synergistic ratio [1-3]. In this study, aPD-L1 and PTX were co-encapsulated in nano-micelles, and after the nano-micelles reached the GBM, the two drugs were released synchronously under a high GSH environment, so that the chemotherapeutic agent and the antibody could be simultaneously accumulated in the TME, which is a better solution to the above problem compared with the formulas of “Free Antibody + Nano-chemotherapeutic Agent” [4] and “Nano Antibody + Nano Antibody” [5].

The description has been added to the discussion section (page20, line 541-553).

Reference:

- [1] Zhang, Bo et al. "Development and evaluation of oxaliplatin and irinotecan co-loaded liposomes for enhanced colorectal cancer therapy." *Journal of controlled release : official journal of the Controlled Release Society* vol. 238 (2016): 10-21.
- [2] Mu, Weiwei et al. "A Review on Nano-Based Drug Delivery System for Cancer Chemotherapy." *Nano-micro letters* vol. 12,1 142. 5 Jul. 2020.
- [3] Xie, Xiaoling et al. "Therapeutic vaccination against leukaemia via the sustained release of co-encapsulated anti-PD-1 and a leukaemia-associated antigen." *Nature biomedical engineering* vol. 5,5 (2021): 414-428. doi:10.1038/s41551-020-00624-6
- [4] Kinoh, Hiroaki et al. "Translational Nanomedicine Boosts Anti-PD1 Therapy to Eradicate Orthotopic PTEN-Negative Glioblastoma." *ACS nano* vol. 14,8 (2020)
- [5] Guo, Chunlei et al. "Liposomal Nanoparticles Carrying anti-IL6R Antibody to the Tumour Microenvironment Inhibit Metastasis in Two Molecular Subtypes of Breast Cancer Mouse Models." *Theranostics* vol. 7,3 775-788. 26 Jan. 2017, doi:10.7150/thno.17237

2. Several studies have demonstrated low PD-L1 expression in the tumor microenvironment of human GBM. Though the authors show that aPD-L1 binds to GL261, it would be important to demonstrate in PDX cell lines that the human antibody against PD-L1 is binding to GBM cells or tumor-associated macrophages from human tissues to provide a justification for using an anti-PD-L1 therapy.

Re: Thanks for the thought-provoking comments.

Studies have shown that PD-L1 was present in about 40% of the GBM patients recruited [1-5]. Here, we analyzed the TCGA database, collected data from GBM patients and analyzed PDX cells, human glioma cell lines and mice glioma cell lines. These results demonstrated that PD-L1 expression is high in GBM and correlates with patient survival.

First, we analyzed the proportion of PD-L1 in common solid tumors using the TCGA database, and the results showed that the expression of PD-L1 in GBM is high. We also analyzed the relationship between the expression of PD-L1 and the overall survival (OS) of GBM patients, and it was seen that the survival of GBM patients with high PD-L1 expression was significantly shorter than that of patients with low PD-L1 expression.

We also collected tissue sections from LGG patients and HGG patients for immunofluorescence staining of CD11b (to exclude bone marrow-derived cells) and PD-L1. It was found that the results of the immunofluorescence staining followed the same trend as the results in the TCGA database, i.e., the expression of PD-L1 was significantly higher in HGG patients than in LGG patients. These results suggested that PD-L1 expression is high in gliomas and correlates with survival and WHO grade in GBM patients.

In addition, we implanted G7, WL1, U87, U251, GL261, and G422 cells in mouse brain (G7 and WL1 cells were GBM patient-derived xenograft (PDX) cell lines, U87 and U251 cells were human glioma cell lines, GL261 and G422 cells were mouse glioma cell lines), respectively. Then the immunofluorescence staining of CD11b and PD-L1 was performed, which showed that the PD-L1 expression of GL261 with G422 GBM model, the model used in this paper, was similar to that of G7, WL1 GBM model (PDX model). Next, we also analyzed these cells by flow cytometry, and found that all glioma cells had significantly higher PD-L1

expression than the isotype group. These results provided strong evidence for our use of anti-PD-L1 therapy (Supplementary Figure 1).

Reference:

- [1] Nduom, Edjah K et al. "PD-L1 expression and prognostic impact in glioblastoma." *Neuro-oncology* vol. 18,2 (2016): 195-205. doi:10.1093/neuonc/nov172
- [2] Sobhani, Navid et al. "Analysis of PD-L1 and CD3 Expression in Glioblastoma Patients and Correlation with Outcome: A Single Center Report." *Biomedicines* vol. 11,2 311. 22 Jan. 2023, doi:10.3390/biomedicines11020311
- [3] Chen, Ruo Qiao et al. "The Prognostic and Therapeutic Value of PD-L1 in Glioma." *Frontiers in pharmacology* vol. 9 1503. 9 Jan. 2019, doi:10.3389/fphar.2018.01503
- [4] Lynes, John P et al. "Biomarkers for immunotherapy for treatment of glioblastoma." *Journal for immunotherapy of cancer* vol. 8,1 (2020): e000348. doi:10.1136/jitc-2019-000348
- [5] Hao, Chengcheng et al. "PD-L1 Expression in Glioblastoma, the Clinical and Prognostic Significance: A Systematic Literature Review and Meta-Analysis." *Frontiers in oncology* vol. 10 1015. 24 Jun. 2020, doi:10.3389/fonc.2020.01015

Supplementary Figure 1. PD-L1 expression in the GBM patients and tumor cell bearing-mouse models. **a** PD-L1 expression percentage in clinical patients. **b** Survival curves of glioma patients from TCGA database (low-grade glioblastoma, LGG, n = 636; high-grade glioblastoma, HGG, n = 138) (PD-L1 expression $\geq 50\%$ was defined as high expression, $< 50\%$ was defined as low expression). **c, d** Respective images and qualification of PD-L1 immunofluorescence of LGG (n = 10) and HGG (n = 10) histological sections obtained from patients. The left images scale bar = 500 μm , the right images scale bar = 50 μm . **e** PD-L1 analysis of 144 HGG and 500 LGG cases acquired from The Cancer Genome Atlas (TCGA) database. Each dot represented a single individual. **f** Qualification of PD-L1 immunofluorescence of G7, WL1, U87, U251, GL261 and G422 tumor mice histological sections obtained from GBM tumor mice. This data only counts the Cy3-PD-L1 fluorescence, n = 3. **g** Representative flow cytometry histogram and qualification of PD-L1 expression on G7, WL1, U87, U251, GL261 and G422 cells. This data only counts the Cy3-PD-L1 fluorescence, n = 3. **h, i** MFI and percentage of PD-L1 expression G7, WL1, U87, U251, GL261 and G422 cells, n = 3. HGG, high grade glioma; LGG, low grade glioma; LUSC, lung squamous cell carcinoma, n = 496; LUAD, lung adenocarcinoma, n = 510; SKCM, skin cutaneous melanoma, n = 469; BRCA, breast invasive carcinoma, n = 1079; PAAD, pancreatic adenocarcinoma, n = 178; ACC, Adrenocortical carcinoma, n = 79. TPM, Transcripts Per Million. G7 and WL1 cells were GBM patient-derived xenograft (PDX) cell lines, U87 and U251 cells were human glioma cell lines, GL261 and G422 cells were mice glioma cell lines. Statistical significance was calculated by one-way ANOVA with Fisher's LSD test in (f), and independent sample *t-test* in (d) and (g).

Description of the results have been added to the results section (page 4-5 line107-136) as:

High expression of PD-L1 from histological sections in glioma patients and glioma-bearing mice

Given reports that the overexpressed PD-L1 is a major prognostic biomarker and predominant therapeutic target for GBM^{7,8}, the PD-L1 expression was investigated from The Cancer Genome Atlas (TCGA), PD-L1 expression is high in high-grade glioma (HGG, GBM). A Kaplan-Meier survival curve investigation of PD-L1 expression with mortality indicated that glioma patients with higher PD-L1 expression had poorer survival outcomes (Supplementary Figure 1a). In addition, we collected tumor samples from human glioma patients, which were classified by WHO classification, including low-grade glioma (LGG) and HGG. We evaluated the integral optical density (IOD) of PD-L1 immunofluorescence (IF) in the samples to quantify the PD-L1 expression (Supplementary Figure 1c and d, Table S1). The PD-L1 was considerably higher in the HGG group than in LGG groups ($P < 0.05$), which is the same trend as the results of TCGA (Supplementary Figure 1e, including 144 cases of HGG and 500 cases of LGG, and was significantly higher ($P < 0.001$) in the HGG group than in the LGG group). These results confirmed that a high PD-L1 expression is apparently an adverse prognostic factor for glioma. Then, the PD-L1 expression of GBM patient-derived xenograft (PDX) cell lines (G7 and WL1), human glioma cell lines (U87 and U251) and mouse glioma cell lines (GL261 and G422) were evaluated. These cell lines were planted in the brains of mice (BALB/c

nu for G7, WL1, U87 and U251 cells, C57BL/6 for GL261 cells and Kunming for G422 cells). To Evaluate the IOD of PD-L1 IF in histological sections from G7, WL1, U87, U251, GL261 and G422 tumor mice, we found that GL261 and G422 tumor mice were similar to the PDX model ($P > 0.05$) (Supplementary Figure 1f). We subsequently assessed the expression of PD-L1 on the surface of these cell lines by flow cytometry and showed that all these cell lines had significant PD-L1 expression compared to isotype ($P < 0.05$) (Supplementary Figure 1g-i). These findings demonstrate PDX cell lines, human cell lines and mice cell lines all bind to aPD-L1 providing a justification for using an anti-PD-L1 therapy in GBM.

The cell culture and patient samples have been added to the method section (page24 line 650-659, page24-25 line 669-675) as:

Cell culture. The G7, WL1 patient-derived xenograft (PDX) cell lines were obtained from Professor Xiuxing Wang's laboratory cell bank. The human U87 cells and U251 cells and the murine GL261-GFP, G422-GFP cells were obtained from our laboratory cell bank. G7, WL1 cells were cultured in Neurobasal® medium (Life Technologies, Cat. No.21103049), supplemented with 2% B27-supplement without Vitamin A (Life Technologies, Cat. No. 12587010), 1% Sodium Pyruvate (Life Technologies, Cat. No.11360-070), 1% GlutaMAX™ Supplement (Life Technologies, Cat. No. 35050061), 100 U/mL penicillin/streptomycin (Invitrogen), 20ng/mL rhEGF (R&D, Cat. No. 4114-TC-01M) and 20ng/mL FGF (Proteintech, Cat. No. 100-8C-10).

Patient samples. Brain tumor specimens were obtained from patients with informed consent and were reviewed by the pathologist and surgeon. Pathologists classified the type and grade of the tumors in accordance with the WHO histological grading of central nervous system tumors. 14 cases of gliomas were selected from Huashan Hospital, Fudan University, and 6 cases of gliomas were selected from Zhongda Hospital, Southeast University. Characteristics of glioma patients were shown in Table S1.

The protocols have been added to the method section (page 26, line 715-725) as:

Detection of PD-L1 expression in GBM patients, mice and cells. For immunofluorescence, tissue sections were used, and PD-L1, CD11b, DAPI stainings were performed according to the manufacturer's instructions. CaseViewer 2.4 was used to observe the images. Five different parts of each slice were randomly selected, and Image Pro Plus 6.0 was used to count positive cells.

For cell flow cytometry, G7, WL1, U251, U87, GL261, G422 tumor cells were incubated with aPD-L1 for 30 min and washed with PBS 3 times. Next, the Alexa647-labeled or Cy3 labeled secondary antibody was incubated with different cells for another 30 min, before flow cytometry (NovoCyte Flow Cytometer, ACEA Biosciences, California, USA), resuspend the cells in 500 μ L PBS, and then analyzed using the FlowJo 10.4 software package to measure the binding of PD-L1 on the cell membrane.

3.The authors performed cytotoxicity assays with free paclitaxel and PTX-contained micelles

in GL261 and GSCs. The results derived from these assays are shown in the main figures for GL261 but not for the GSCs which are more clinically relevant for GBM patients considering the big genomic and phenotypic differences between GL261 and human gliomas. Thus, it is recommended to show the cytotoxicity assay and ICD results derived from GL261 and the GSCs in the main figures.

Re: Thank you for your kind suggestion. GSCs are indeed more relevant to clinical GBM patients, and we have moved the cytotoxicity and ICD results of GSCs to the main figure 2.

4. The rationale for the use of paclitaxel as opposed to other chemotherapeutic drugs to induce immunogenic cell death is missing. Where there any comparative experiments in this context? There is some clinical trial literature on the use of chemotherapy to enhance response to immunotherapy. This should be considered and properly cited.

Re: Thank you for your comment.

PTX is one of the most widely used anti-tumor drugs due to its significant efficacy in a wide range of cancers [1-4]. However, its application in the treatment of GBM was limited after the failure of early clinical trials because of its poor brain penetration [5,6]. The non-invasive and low-toxicity delivery of PTX to the brain remains a challenging area of research [7]. Meanwhile, PTX can induce a stronger ICD effect compared with other chemotherapeutic drugs (Ref. [8] compared with oxaliplatin and cisplatin, Ref. [9] temozolomide monotherapy fails to elicit significant ICD effects).

With your kind reminder, we have summarized the clinical studies of chemotherapy drugs plus immunotherapy, and the results shown that chemotherapy drugs could effectively enhance immunotherapy and improved the survival of patients. Chemotherapy drug-enhanced immunotherapy is a promising treatment modality. [Ref. 16-19].

Reference:

- [1] Scribano, Christina M et al. Chromosomal instability sensitizes patient breast tumors to multipolar divisions induced by paclitaxel. *Science translational medicine* vol. 13,610 (2021): eabd4811.
- [2] Abu Samaan, T. M., Samec, M., Liskova, A., Kubatka, P. & Büsselberg, D. Paclitaxel's mechanistic and clinical effects on breast cancer. *Biomolecules* 9, 789 (2019).
- [3] Kampan, N. C., Madondo, M. T., McNally, O. M., Quinn, M., & Plebanski, M. Paclitaxel and its evolving role in the management of ovarian cancer. *Biomed. Res. Int.* 2015, 413076 (2015).
- [4] Blair, H. A. & Deeks, E. D. Albumin-bound paclitaxel: a review in non- small cell lung cancer. *Drugs* 75,2017–2024 (2015).
- [5] Fountzilias, G et al. "Radiation and concomitant weekly administration of paclitaxel in patients with glioblastoma multiforme. A phase II study." *Journal of neuro-oncology* vol. 45,2 (1999): 159-65.
- [6] Elinzano, Heinrich et al. "PPX and Concurrent Radiation for Newly Diagnosed Glioblastoma Without MGMT Methylation: A Randomized Phase II Study: BrUOG 244." *American journal of clinical oncology* vol. 41,2 (2018): 159-162
- [7] Sonabend, Adam M et al. "Repeated blood-brain barrier opening with an implantable ultrasound device for delivery of albumin-bound paclitaxel in patients with recurrent glioblastoma: a phase 1 trial." *The Lancet. Oncology* vol. 24,5 (2023): 509-522.
- [8] Yang, Qianmei et al. "Nanomicelle protects the immune activation effects of Paclitaxel and sensitizes tumors to anti-PD-1 Immunotherapy." *Theranostics* vol. 10,18 8382-8399. 9 Jul. 2020, doi:10.7150/thno.45391
- [9] Zhou, Yan et al. "Temozolomide-based sonodynamic therapy induces immunogenic cell death in glioma." *Clinical immunology (Orlando, Fla.)*, vol. 256 109772. 15 Sep. 2023, doi:10.1016/j.clim.2023.109772

The description has been added to the introduction section (page 3 line 67, 68) as:

Current clinical trials using chemotherapy to enhance immunotherapy response advancing in many solid tumors¹⁶⁻¹⁹.

Reference:

16. Heymach, J. V. *et al.* Design and Rationale for a Phase III, Double-Blind, Placebo-Controlled Study of Neoadjuvant Durvalumab + Chemotherapy Followed by Adjuvant Durvalumab for the Treatment of Patients With Resectable Stages II and III non-small-cell Lung Cancer: The AEGEAN Tr. *Clin. Lung Cancer* **23**, e247–e251 (2022).
17. Oh, D.-Y. *et al.* Gemcitabine and cisplatin plus durvalumab with or without tremelimumab in chemotherapy-naïve patients with advanced biliary tract cancer: an open-label, single-centre, phase 2 study. *Lancet Gastroenterol. Hepatol.* **7**, 522–532 (2022).
18. Vogelzang, N. J. *et al.* Efficacy and Safety of Autologous Dendritic Cell–Based Immunotherapy, Docetaxel, and Prednisone vs Placebo in Patients With Metastatic Castration-Resistant Prostate Cancer. *JAMA Oncol.* **8**, 546 (2022).
19. Rob, L. *et al.* Safety and efficacy of dendritic cell-based immunotherapy DCVAC/OvCa added to first-line chemotherapy (carboplatin plus paclitaxel) for epithelial ovarian cancer: a phase 2, open-label, multicenter, randomized trial. *J. Immunother. Cancer* **10**, e003190 (2022).

5. It is important to provide a paclitaxel concentration context. Are the concentrations achieved leading to immunogenic cell death in human glioma cell lines and PDX?

Re: Thank you for your professional suggestion. We detected PTX concentrations in brain tumors (GL261 model) using HPLC (Mean \pm SD = 6.05 ± 2.19 $\mu\text{g/mL}$). Then, we assayed CRT expression in different cell lines (G7, WL1, U87, U251, GL261, G422 cells) stimulated with different PTX concentrations. The results showed that in any type of cells, 6 $\mu\text{g/mL}$ PTX could induce sufficient CRT expression compared with the 0 $\mu\text{g/mL}$ group ($P < 0.05$) (Supplementary Figure 16).

Supplementary Figure 16. Different concentrations of PTX induced different levels of CRT expression. Quantification of CRT exposure on the surface of G7, WL1, U87, U251, GL261 and G422 cells after receiving different PTX concentration treatments. n = 4 biologically independent samples. Statistical significance was calculated by one-way ANOVA with Fisher's LSD test.

The description has been added to the results section (page 10 line262-267) as:

In addition, in order to demonstrate that PTX can induce ICD effects in both mouse GBM

cells and human GBM cells, we also examined the ability to induce CRT expression in G7, WL1, U87, U251, GL261, G422 cells under different concentrations of PTX conditions, and the results showed 2 $\mu\text{g}/\text{mL}$ PTX induced significant CRT expression in mouse cell lines, and in any type of cells, 6 $\mu\text{g}/\text{mL}$ PTX induced noteworthy CRT expression ($P < 0.05$) (Supplementary Fig. 16).

The protocols have been added to the method section (page 31, line 844-855; page 33, line 916-921) as:

To evaluate the ICD effect of G7, WL1, U87, U251, GL261, G422 cells before and after PTX treatment in different concentration, G7, WL1, U87, U251, GL261, G422 cells (1×10^5 cells) were seeded in a 12-well plate. After 24 hours of incubation, the cell culture medium was removed, and the cells were not treated or treated with for different concentration of PTX for 24 h to induce ICD. To measure the CRT expression, after treatment, the cells were washed with cold PBS, and further incubated with anti-CRT polyclonal antibody (diluted 1:100, anti-human antibody for G7, WL1, U87, U251 cells and anti-mouse antibody for GL261, G422 cells) at 4 °C for 30 min, and stained with Alexa Fluor 647 labeled goat antirabbit IgG (H&L) highly cross-adsorbed secondary antibody (1:200 dilution) 30 min at room temperature. Then the cells were resuspended in 500 μL PBS and examined using a NovoCyte Flow Cytometer (ACEA Biosciences, California, USA), analyzed with Flowjo.

To determine the amount of PTX reaching the brain in A2APM, at 90 min post-injection of A2-APM, the GL261 tumor-bearing mice were euthanized, and the brains were harvested. The homogenate (0.2 mL) was mixed with acetonitrile (0.2 mL), vortexed for 5 min and centrifuged at 10,000g for 10 min. The amount of PTX extracted in the supernatant was quantified by HPLC. The brain-targeting concentration was calculated.

6. Appropriate statistical analyses need to be performed for several figures. For instance, it is recommended to use one-way ANOVA with multiple comparison post hoc tests in Fig. 1d, 1e, 2b, 2d, 2e, 2f, 2g, 3f, 5c, and 5e. A different statistical test is recommended to be used in outcomes involving 2 variables such as two-way ANOVA. The inclusion of these tests would strengthen the argument that observed differences are significant and not due to random chance.

Re: Thanks for your careful checks. We went through the whole manuscript, analyzed all statistics with SPSS 26.0, and corrected these mistakes as follows:

Fig. 1d, e was calculated by one-way ANOVA with Fisher's LSD test.

Fig. 2a, b, d-f, h, i, k was calculated by one-way ANOVA with Fisher's LSD test.

Fig. 3b, d, f was calculated by one-way ANOVA with Fisher's LSD test, and j was calculated by independent sample *t*-test.

Fig. 4d, e was calculated by log-rank test.

new Fig. 5b, c, d was calculated by one-way ANOVA with Fisher's LSD test.

new Fig. 6b, c, d was calculated by one-way ANOVA with Fisher's LSD test.

new Fig. 7c, e was calculated by independent sample *t*-test, and g was calculated by log-rank test.

Supplementary Figure 1f was calculated by one-way ANOVA with Fisher's LSD test, and

d and g were calculated by independent sample *t-test*.

Supplementary Figure 12b, 13 14, 16, 18, 31, 33, 35-42, 44, 45 was calculated by one-way ANOVA with Fisher's LSD test.

Supplementary Figure 15b, d was calculated by independent sample *t-test*.

Supplementary Figure 23, 24c was calculated by log-rank test.

7. The BBB in vitro system in Figures 3a and 3b has several limitations. The BBB is normally constituted by not just endothelial cells, but astrocytic end-feet, microglia, pericytes, and basement membrane. The findings cannot be extrapolated to an in vivo model or a phenomenon occurring in GBM patients.

Re: Thanks for this insightful comment. We tried to build a BBB model consisting of endothelial cells, astrocyte end-feet, microglia, pericytes, and basement membranes, but the results were not satisfactory. It is technically difficult for us to conduct this experiment. The *in vitro* BBB model in our study is a well-recognized model used in many GBM-studying literatures (e.g. Xue, Jingwei et al. Nature Nanotechnology vol. 12,7 (2017): 692-700. doi:10.1038/nnano.2017.54; Xie, Jinbing et al. Biomaterials vol. 224 (2019): 119491. doi:10.1016/j.biomaterials.2019.119491).

8. Figure 4e needs a group of mice treated with a combination of free aPD-L1 and paclitaxel to compare with the group that receive A2-APM. Without this group, it is not possible to assess the additive effect of encapsulating both drugs for delivery into the brain. Same comment applies to Supplementary Figure 17.

Re: Thank you for your comment. To verify the additive effect of co-delivery of both drugs, after planting GL261 or G422 cells into the mice brain, we injected the combination of free aPD-L1 and PTX, and A2-APM, respectively, into GBM mice on days 7 and 14. It was found that the survival of the A2-APM group was obviously longer than that of the free aPD-L1 + PTX group (Supplementary Figure 23).

Supplementary Figure 23. Assessment of the additive effect of co-encapsulated PTX and aPD-L1 delivery into the brain. a Survival curves of treated and control GL261 tumor mice. n = 6. **b** Survival curves of treated and control G422 tumor mice. n = 5. Statistical significance was calculated by log-rank test.

The description has been added to the results section (page 13 line 344-348) as:

To assess the additive effect of delivering co-encapsulated PTX and aPD-L1 into the brain, the

antitumor efficacy of free aPD-L1 and free PTX and A2-APM were examined in GL261 and G422 tumor mice (Supplementary Fig. 23). The rate of mouse survival in the A2-APM treatment group was significantly improved as compared to the free aPD-L1 and free PTX groups ($P < 0.05$).

9. Would recommend moving Supplementary Figure 17 to the main Figure to show efficacy in two separate intracranial model.

Re: Thank you for your kind suggestion that survival of G422 mice model should be moved to the main figure to show A2-APM anti-tumor efficacy in two intracranial models. We have moved the survival of G422 mice model (old Supplementary Figure 17) to main Figure 4.

10. Did the A2-APM lead to survival benefit in the post- resection model? This is not clearly shown.

Re: In the post-resection model, the A2-APM group showed prolonged survival (new Supplementary Figure 24c), and we added the P-value to the figure. The size of the tumor change was counted to better reflect the therapeutic effect of A2-APM (Supplementary Figure 24e).

Supplementary Figure 24 c Survival curves of treated and control mice in postresection GBM mice. n = 4. Statistical significance was calculated by log-rank test.

Supplementary Figure 24 e Quantified MR signal intensity of GL261 tumor-bearing mice treated with saline, free aPD-L1, free PTX, APM, and A2-APM.

11. For Figures 2g and 2h, it is unclear, and it is important to show how the gating was done to arrive to the conclusion that there are more CD11c⁺ MHC II⁺ CD80⁺ CD86⁺ cells. For instance, the authors mention that “the expression of CD11c, MHC II, CD80, and CD86 was similar to that of the free PTX treated group, but more than a 10-fold increase as compared to that of the blank micelles and AM treated groups.” But there is no comparison of CD11c and MHC II expression between groups. It is recommended to perform the comparison of the expression of each marker to determine that there are mature DCs

Re: Thank you for this comment. We're sorry for our lack of clarity. The DC gating strategy was shown (new figure 2g and Supplementary Figure 46c) and each marker's expression was individually analyzed (new figure 2h). The results showed that APM-treated group had a higher level in MHC II, CD80 and CD86 expression, which indicated mature DCs.

Fig. 2g Gating strategy to assess the levels of MHCII, CD80, and CD86 in DC2.4 cells gated on Live⁺CD11c⁺ cells. **h** Promotion of DC2.4 maturation after co-incubation with pretreated GL261 cells (n = 3 biologically independent samples).

Supplementary Figure 46 c Gating strategy to assess the levels of MHCII, CD80, and CD86 in tumor tissues gated on CD45⁺Live⁺CD11c⁺ cells presented in Figure 2h, Supplementary Figure 36 and Supplementary Figure 40.

The description has been changed to the results section (page 9 line 228-234) as:

Analysis of biomarkers of mature DCs (CD11c, MHC II, CD80, and CD86) by flow cytometry showed that when GL261 cells were pretreated with PM or APM, the maturation of DCs was greatly promoted (Fig. 2g, h). The expressions of CD11c, MHC II, CD80, and CD86 were similar to those of the free PTX treatment group, but significantly increased as compared to those of the blank micelles and AM treatment groups ($P < 0.05$).

12. It is recommended to refrain from using the terms M1/M2 to refer to tumor-associated macrophages as this can mislead the real phenotype of both macrophages and microglia described by single-cell RNA-sequencing analyses where the M1/M2 has not been demonstrated. Instead, the function of the markers identified by flow cytometry can be described for the experimental group. More compelling demonstration of the macrophage phenotype and how this is modulated by their treatment, would require co-culture with T-cells to determine the effect of these macrophages on T-cell activity.

Re: Thank you for your suggestion. We removed M1/M2 terms and added the macrophage and T cell co-culture experiment. The primary macrophages (BMDMs) were extracted from the bone marrow, and T cells were extracted from the spleen of mice. Then, BMDMs were co-cultured with untreated tumor cells or APM-treated tumor cells. Next, T cells were added to the culture dish (Supplementary Figure 13a and 14a). Afterwards we analyzed and compared

the MFI of CD86 and CD206 in F4/80⁺CD11b⁺ cells, and the percentage of IFN γ ⁺ cells in CD3⁺CD8⁺ T cells in these groups. The results showed that APM increased the expression of CD86, decreased the expression of CD206 in F4/80⁺CD11b⁺ cells, and increased the proportion of IFN γ ⁺CD3⁺CD8⁺ T cells.

Supplementary Figure 13. APM shifted macrophages toward a pro-inflammatory phenotype and enhanced T cell function. **a** Schematic illustration of the *in vitro* model to evaluate the effect of untreated tumor cells and APM treated tumor cells on macrophages and T cells. **b** MFI of CD86⁺ and CD206⁺ cells in F4/80⁺ CD11b⁺ cells by flow cytometry. **c** Percentage of IFN γ ⁺ cells in CD3⁺CD8⁺ T cells by flow cytometry. n = 3. Statistical significance was calculated by one-way ANOVA with Fisher's LSD test.

The description has been added to the results section (page 9, line236-245) as:

To investigate the effect of APM treated tumor cells on macrophages or DCs and their effect on T cells, bone marrow-derived macrophages (BMDMs), bone marrow-derived dendritic cells (BMDCs) and T cells were extracted from C57 mice. Then BMDMs and BMDCs were treated with tumor cells or A2-APM treated tumor cells (Supplementary Fig. 13a and Supplementary Fig. 14a) before adding T cells into the culture dish. In the A2-APM treated tumor cells group, CD206, the marker of inhibition, significantly decreased, and CD86, the marker of anti-tumor, increased (Supplementary Fig. 13b), meanwhile, the BMDCs were matured with the rise of CD80 and CD86 expression (Supplementary Fig. 14b). Additionally, T cells were observed with the IFN- γ expression level elevated in CD3⁺ CD8⁺ T cells.

The protocols have been added to the materials section (page 31, line 884-887) as:

To evaluate the ICD effect on macrophages or DCs and their effect on T cell activation, we extracted primary BMDMs and BMDCs. In brief, the femurs, tibias, and spleen were dissected from C57BL/6 mice. Then, the bone marrow was rinsed and cultured in a medium (DMEM containing 20 ng mL⁻¹ M-CSF for macrophage, RPMI1640 containing 20 ng mL⁻¹ M-

CSF for DC) for 6 days, the spleen was sheared and digested, then sorted by magnetic beads and cultured in a RPMI1640 medium. On day 7, naive BMDMs were collected and cultured with a medium containing untreated tumor cells and APM treated tumor cells for 24 h. After that, T cells were collected and cultured in the culture dish for another 24 h. Then the cells were washed with cold PBS, and further incubated with anti-F4/80, anti-CD86, anti-CD206 antibodies for macrophages, anti-CD11c, anti-CD80, anti-CD86 antibodies for DCs, and anti-CD3, anti-CD8, anti-IFN- γ for T cells for 30 min. Then the cells were resuspended in 500 μ L PBS, and examined using a flow cytometry (FACSCelesta, Becton, Dickinson and Company, New Jersey, USA), analyzed with Flowjo.

Minor comments

13. It is not clear the description of what is A2-APM in the introduction and the first section of the results.

Re: Thank you for your comment. We described the angiopep-2-aPD-L1@PTX nano-micelles (A2-APM) in the introduction section and the first part of the results (page 4 line 91, page 5 line 135).

14. There are no P values in Fig 4e.

Re: Thanks for your detailed check, we have added the P-values in new fig. 4d and e.

Fig. 4 d Survival curves of treated and control GL261 tumor-bearing mice. **e** Survival curves of treated and control G422 tumor mice. $n = 5$. Statistical significance was calculated by log-rank test.

Reviewer #4 (Remarks to the Author): with expertise in nanotechnology for cancer therapy, glioblastoma

In this manuscript, Zhang et al have developed a redox-sensitive polymer micelle called A2-APM, which demonstrates the co-encapsulation of anti-PD-L1 antibody and PTX. By employing angiopep-2 peptide decoration, A2-APM exhibits the ability to traverse the blood-tumor barrier (BTB). Within the reductive microenvironment of glioblastoma (GBM), the anti-PD-L1 antibody and PTX are selectively released. Notably, the released PTX elicits immunogenic cell death in glioma cells, thereby enhancing immunogenicity and facilitating a localized immune response. Furthermore, the released anti-PD-L1 antibody serves to revitalize infiltrated T cells. Despite the authors' comprehensive investigations supporting the strong anti-GBM efficacy, the manuscript suffers from substandard writing quality and lacks the requisite novelty to warrant publication in this journal. Moreover, several concerns remain unaddressed and require further attention.

1. The paper exhibits limited novelty in regards to the designed micelles, as the authors have not adequately demonstrated the essentiality of combining the anti-PD-L1 antibody and PTX. Moreover, the manuscript lacks a coherent structure that effectively highlights the pivotal scientific problem and the unique contributions of the current study.

Re: Thank you for your insightful comments.

The GBM tumor microenvironment expresses high level of PD-L1 (Supplementary Fig. 1) [Ref. 1-5], therefore it is rational to GBM with aPD-L1. However, due to the immunosuppressive tumor microenvironment (TME) of GBM, the effect of mono-immunotherapy is limited. Chemotherapy can improve the efficacy of immunotherapy by ICD effect, but mono-chemotherapy leads to upregulation of PD-L1 expression [6-8]. Therefore, combining aPD-L1 with paclitaxel can both improve the inhibitory microenvironment of GBM and block the PD-1/PD-L1 axis. PTX-induced ICD promotes the maturation of DCs, which delivers antigens to T cells and activates them. aPD-L1 further deregulates immunosuppression and enhances the function of cytotoxic T cells, thus enhancing the efficacy of anti-GBM therapy.

In this study, A2-APM increased the number and activity of CD8 T cells in TME. Through depletion experiments, the efficacy of A2-APM which mainly depends on CD8 T cells was determined. Meanwhile, A2-APM induced a prolonged immune memory that protected the GBM mice against rechallenge with tumor cells. A2-APM also promoted DC maturation and resulted in pro-inflammatory transformation of macrophages and myeloid-derived suppressor cells.

This was mainly due to the design of the nano-micelles, where angiopep-2-aPD-L1@PTX nano-micelle (A2-APM) co-encapsulated aPD-L1 and PTX. The angiopep-2 (A2) peptide was decorated on the surface of nano-micelles, which allowed the BTB penetration. PTX and aPD-L1 released in tumor microenvironment of high GSH concentration. The co-encapsulation strategy improved the targeting of aPD-L1 and PTX to gliomas and allowed both drugs to be released simultaneously in the TME and work together to enhance the therapeutic efficacy.

To better highlight the significance of this study, we changed the structure of the article

and added subheadings in results section, mainly in the section on A2-APM induced immune effects. The subheadings are now as follows:

1. High expression of PD-L1 from histological sections in glioma patients and glioma-bearing mice.
2. Characterization of redox-responsive A2-APM.
3. Cytotoxicity and ICD induction ability of APM in GL261 cells.
4. ICD induction ability of APM in glioma stem-like cells and human cell lines.
5. Accumulation and biodistribution of A2-APM in tumor models.
6. *In vivo* antitumor efficacy of A2-APM.
7. Biosafety of A2-APM.
8. A2-APM treatment activates effector T cells.
9. A2-APM treatment efficacy is dependent on CD8 T cells.
10. A2-APM treatment promotes DC maturation.
11. A2-APM treatment results in a pro-inflammatory shift of macrophages and myeloid derived suppressor cells.
12. A2-APM treatment elicits a prolonged immune memory.

The main figure of the article was increased to seven, with major changes to figs. 5, 6, and 7:

Fig. 5 A2-APM enhances specific immunity and its treatment effect is dependent on CD8 T cells

Fig. 6 A2-APM treatment promotes DC maturation and results in pro-inflammatory transformation of macrophages and myeloid-derived suppressor cells.

Fig. 7 A2-APM treatment induces prolonged immune memory and protects the GBM mice against rechallenge with tumor cells.

We have also changed parts of the introduction and discussion sections, and changed inaccurate description in the original text.

Reference:

- [1] Nduom, Edjah K et al. "PD-L1 expression and prognostic impact in glioblastoma." *Neuro-oncology* vol. 18,2 (2016): 195-205. doi:10.1093/neuonc/nov172
- [2] Sobhani, Navid et al. "Analysis of PD-L1 and CD3 Expression in Glioblastoma Patients and Correlation with Outcome: A Single Center Report." *Biomedicines* vol. 11,2 311. 22 Jan. 2023, doi:10.3390/biomedicines11020311
- [3] Chen, Ruo Qiao et al. "The Prognostic and Therapeutic Value of PD-L1 in Glioma." *Frontiers in pharmacology* vol. 9 1503. 9 Jan. 2019, doi:10.3389/fphar.2018.01503
- [4] Lynes, John P et al. "Biomarkers for immunotherapy for treatment of glioblastoma." *Journal for immunotherapy of cancer* vol. 8,1 (2020): e000348. doi:10.1136/jitc-2019-000348
- [5] Hao, Chengcheng et al. "PD-L1 Expression in Glioblastoma, the Clinical and Prognostic Significance: A Systematic Literature Review and Meta-Analysis." *Frontiers in oncology* vol. 10 1015. 24 Jun. 2020, doi:10.3389/fonc.2020.01015
- [6] Lau, Tat San et al. "Paclitaxel Induces Immunogenic Cell Death in Ovarian Cancer via TLR4/IKK2/SNARE-Dependent Exocytosis." *Cancer immunology research* vol. 8,8 (2020): 1099-1111. doi:10.1158/2326-6066.CIR-19-0616
- [7] Yamaguchi, Hirohito et al. "Mechanisms regulating PD-L1 expression in cancers and associated opportunities for novel small-molecule therapeutics." *Nature reviews. Clinical oncology* vol. 19,5 (2022): 287-305. doi:10.1038/s41571-022-00601-9
- [8] Huang, Kevin Chih-Yang et al. "Decitabine Augments Chemotherapy-Induced PD-L1 Upregulation for PD-L1 Blockade

2. To elucidate the underlying mechanisms driving the antitumor activity, a depletion study is imperative to investigate the specific contribution of immune cells in this context.

Re: Thanks for this valuable suggestion. We performed lymphatic depletion experiment to identify the subtype of T cells in antitumor activity.

Our results demonstrated that T cells play an important role in A2-APM anti-GBM efficacy. The subtype of T cells (CD4⁺ or CD8⁺) plays a dominant role in A2-APM antitumor therapy was investigated. Depletion of CD8⁺ and CD4⁺ T cells was performed by *i.p.* injection of respective *in vivo* antibodies in 200 μ L PBS per mouse (all from BioXCell): anti-CD4 (Clone, YTS191; 1000 μ g/mouse); anti-CD8 (Clone, 2.4; 500 μ g /mouse) on days 7, 12, 18, and 23. At the experiment endpoint, depletion was verified at the brain tumor site using flow cytometry.

Based on the increase in tumor volume over four weeks, depletion of CD8⁺ T cells significantly abrogated the therapeutic effect of A2-APM ($P = 0.042$), but not for that of CD4⁺ T cell depletion ($P = 0.140$) (Fig.6d, Supplementary Fig. 35). Therefore, we identified a major role for CD8⁺ T cells in mediating tumor regression of A2-APM therapy.

Fig. 6 d Schematic of the treatment regimen for depletion of T cells. **e** GL261 tumor volume (measured by MRI) after treatment with saline (n = 3), A2-APM (n = 5), A2-APM + α -CD4 (n = 5), A2-APM + α -CD8 (n = 4). All statistics are expressed as mean \pm SD. Statistical significance was calculated by one-way ANOVA with Fisher’s LSD test.

Supplementary Figure 35. Depletion of CD8⁺ T cells abrogates A2-APM treatment

efficacy. a Gating strategy for CD4⁺ and CD8⁺ T cells. **b, c** Depletion was confirmed on day 29 in the TME by flow cytometry. All data are presented as individual values and the mean \pm SEM. Statistical significance was calculated by one-way ANOVA with Fisher's LSD test.

The description has been added to the results section (page 16 line 442-449) as:

A2-APM treatment efficacy is dependent on CD8⁺ T cells. Since our results above demonstrated that T cells play an important role in A2-APM anti-GBM efficacy, we next investigated which subtype of T cells (CD4 or CD8) plays a dominant role in A2-APM antitumor therapy. Notably, based on the increase in tumor volume over four weeks, depletion of CD8⁺ T cells significantly abrogated the therapeutic effect of A2-APM ($P = 0.042$), whereas depletion of CD4⁺ T cells did not ($P = 0.140$) (Fig. 5d, Supplementary Fig. 35). Therefore, we identified a major role for CD8⁺ T cells in mediating tumor regression of A2-APM therapy.

The protocols have been added to the results section (page 37 line 1013-1018) as:

In vivo depletion experiments

Depletion of CD8⁺ and CD4⁺ T cells was performed by i.p. injection of respective antibodies in 200 μ L PBS per mouse (all from BioXCell): anti-CD4 (Clone, YTS191; 1000 μ g/mouse); anti-CD8 (Clone, 2.4; 500 μ g /mouse) injected on days 7, 12, 18, and 23. At the experiment endpoint, depletion was verified at the brain tumor site using flow cytometry.

3. In the discussion section, the authors highlighted the potential of chemotherapeutic agents inducing immunogenic cell death (ICD) to augment the efficacy of immune checkpoint blockade (ICB) through the activation of T cells. To validate this hypothesis, it is crucial to confirm alterations in immune markers (such as PD-1, TIM-3, IFN- γ , GrzB, et al) within T cells via flow cytometry analysis.

Re: Thanks for your insightful comment. After administration of different drugs, changes in the PD-1, TIM-3, IFN- γ , Granzyme B on the CD8⁺ T cells in GL261 and G422 tumor mice were detected by flow cytometry (Supplementary Fig.34 and 40). The results showed that in the A2-APM treated group, markers for antitumor (IFN γ ; Granzyme B, GrzB) were upregulated, whereas the markers for inhibitory (PD-1, TIM-3) were decreased.

Fig. 6 c Quantification of tumor-infiltrating IFN γ ⁺, Granzyme B⁺ (GrzB⁺), PD-1⁺ and TIM-3⁺ T cells three days after two doses of saline, free aPD-L1, free PTX, A2-AM, A2-PM, APM and A2-APM, n = 3.

Supplementary Figure 34. A2-APM treatment increases the proportion of cytotoxic CD8 T cells and decreases the proportion of exhausted CD8 T cells in GL261 tumor mice. Representative flow cytometric contour plots of tumor-infiltrating IFN γ ⁺, Granzyme B⁺ (GrzB⁺), PD-1⁺ and TIM-3⁺ T cells three days after two doses of saline, free aPD-L1, free PTX, A2-AM, A2-PM, APM and A2-APM. Insets: the subsets of IFN γ ⁺, Granzyme B⁺ (GrzB⁺), PD-1⁺ and TIM-3⁺ T cells in CD3⁺ CD8⁺ T cells; the numbers indicate the percentage of IFN γ ⁺, Granzyme B⁺ (GrzB⁺), PD-1⁺ and TIM-3⁺ T cells in CD3⁺ CD8⁺ T cells after various treatments.

a**Gated on CD8⁺ CD3⁺ CD45⁺ cells****b**
Supplementary Figure 39. A2-APM treatment increases the proportion of cytotoxic CD8 T cells and decreases the proportion of exhausted CD8 T cells in G422 tumor mice. a Representative flow cytometric contour plots of tumor-infiltrating IFNγ⁺, Granzyme B⁺ (GrzB⁺), PD-1⁺ and TIM-3⁺ T cells three days after two doses of saline, free aPD-L1, free PTX, A2-AM, A2-PM, APM and A2-APM. Insets: the subsets of IFNγ⁺, Granzyme B⁺ (GrzB⁺), PD-1⁺ and TIM-3⁺ T cells in CD3⁺ CD8⁺ T cells; the numbers indicate the percentage of IFNγ⁺, Granzyme B⁺ (GrzB⁺), PD-1⁺ and TIM-3⁺ T cells in CD3⁺ CD8⁺ T cells after various treatments. **b** Quantification of tumor-infiltrating IFNγ⁺, Granzyme B⁺ (GrzB⁺), PD-1⁺ and TIM-3⁺ T cells three days after two doses of saline, free aPD-L1, free PTX, A2-AM, A2-PM, APM and A2-APM. All statistics are expressed as mean ± SD, n = 3. Statistical significance was calculated by one-way ANOVA with Fisher's LSD test.

The description has been added to the method section (page 16 line 437-440) as:

Further, we examined the key markers on CD8⁺ T cells and found that in the A2-APM treatment group, the antitumor markers (IFN γ ; Granzyme B, GrzB) were upregulated, whereas the inhibitory markers (PD-1, TIM-3) were decreased (Fig. 5c, Supplementary Fig. 34).

4. The authors employed t-tests instead of employing one-way analysis of variance (ANOVA) comparisons to assess the significance among multiple groups. As the data in this paper are analyzed, it is essential to meticulously examine the statistical significance and draw appropriate conclusions by utilizing the appropriate comparison method.

Re: Thanks for your careful checks. We are sorry for our mistakes. We went through the whole manuscript, analyzed all statistics with SPSS 26.0, and revised these mistakes as follows:

Fig.1d, e was calculated by one-way ANOVA with Fisher's LSD test.

Fig.2a, b, d-f, h, i, k was calculated by one-way ANOVA with Fisher's LSD test.

Fig.3b, d, f was calculated by one-way ANOVA with Fisher's LSD test, and j was calculated by independent sample *t-test*.

Fig.4d, e was calculated by log-rank test.

new Fig.5b, c, d was calculated by one-way ANOVA with Fisher's LSD test.

new Fig.6b, c, d was calculated by one-way ANOVA with Fisher's LSD test.

new Fig. 7c, e was calculated by independent sample *t-test*, and g was calculated by log-rank test.

Supplementary Figure 1f was calculated by one-way ANOVA with Fisher's LSD test, and d and g were calculated by independent sample *t-test*.

Supplementary Figure 12b, 13, 14, 16, 18, 31, 33, 35-42, 44, 45 was calculated by one-way ANOVA with Fisher's LSD test.

Supplementary Figure 15b, d was calculated by independent sample *t-test*.

Supplementary Figure 23, 24c was calculated by log-rank test.

5. In the introduction section, the authors made a claim regarding the potential lymphodepletion resulting from the systemic administration of chemotherapeutic agents. However, the current studies lack data supporting this assertion. It is imperative for the authors to provide evidence demonstrating whether the constructed micelles, through the systemic administration of PTX, can effectively mitigate lymphodepletion.

Re: Thank you for your comment, when writing the introduction, we referred to the literature (Mathios, Dimitrios et al. "Anti-PD-1 antitumor immunity is enhanced by local and abrogated by systemic chemotherapy in GBM." *Science translational medicine* vol. 8,370 (2016): 370ra180. doi:10.1126/scitranslmed.aag2942). However, this statement was inappropriate for our study, so we added an experiment to explore whether A2-APM causes lymphodepletion.

Saline, free PTX, A2-APM (5 mg/kg PTX) were administered on days 14, 16 and 18, respectively. For the flow cytometry analysis, blood, lymph nodes and spleen were collected on day 21. There was no significant difference in the proportion of CD45⁺ cells between the saline and A2-APM groups in blood, lymph nodes and spleen. Furthermore, we find that the

proportion of inhibitory T cells (PD-1⁺ or TIM-3⁺CD3⁺CD8⁺ T cells) in the A2-APM group was significantly lower than that in the free PTX group. These results suggested that A2-APM did not lead to lymphatic depletion and did not increase the proportion of exhausted T cells.

Supplementary Figure 31. A2-APM can effectively mitigate lymphodepletion. **a** Schematic of the treatment regimen. **b** Gating strategy to sort CD45⁺ cells, CD3⁺CD8⁺ T cells, PD-1⁺ cells, TIM-3⁺ cells in tumor tissues. **c-e** Quantification of blood, lymph nodes and spleen-infiltrating

CD45⁺ cells, CD3⁺CD8⁺ T cells in CD45⁺ cells and PD-1⁺, TIM-3⁺ cells in CD3⁺CD8⁺ T cells three days after three treatments with saline, free PTX and A2-APM (PTX conc. = 5 mg/kg). n = 4 saline treated mice and 5 free PTX and A2-APM treated mice. Statistical significance was calculated by one-way ANOVA with Fisher's LSD test.

The description has been added to the results section (page 15 line 403-414) as:

To evaluate whether A2-APM can effectively mitigate lymphodepletion, the drugs were injected intravenously to GL261 tumor-bearing mice. Saline, free PTX, A2-APM (5 mg/kg PTX) were administered on days 14, 16 and 18. Blood, lymph nodes and spleen were collected on day 21 for flow cytometry analysis (Supplementary Fig. 31). There was no significant difference in the proportion of CD45⁺ cells between the saline and A2-APM groups in blood, lymph nodes and spleen. In contrast, the free PTX group showed a significant decrease in the proportion of CD45⁺ in the blood and in the proportion of CD3⁺ CD8⁺ T cells in the spleen. Furthermore, we can find that the proportion of inhibitory T cells (PD-1⁺ or TIM-3⁺CD3⁺CD8⁺ T cells) in the A2-APM group was significantly lower than that in the free PTX group. These results suggested that A2-APM did not lead to lymphatic depletion and did not increase the proportion of exhausted T cells.

The protocols have been added to the method section (page 38 line 1052-1060) as:

To evaluate whether A2-APM can effectively mitigate lymphodepletion, GL261 tumor-bearing mice were randomized, and the drugs were injected intravenously *via* the tail vein. Saline, free PTX, A2-APM (5 mg/kg PTX) were administered on days 14, 16 and 18. For the flow cytometry analysis, blood, lymph nodes and spleen were collected on day 21. Mouse lymphocyte separation medium was used for dissociation. The cells were stained with anti-CD45, anti-CD3, anti-CD8, anti-PD-1, and anti-TIM-3 antibodies for 30 min. Thereafter, the stained cells were analyzed by flow cytometry (FACSCelesta, Becton, Dickinson and Company, New Jersey, USA) for CD8 and Treg cell populations (collecting 1×10^4 events for analysis).

6. HMGB1, a cytokine closely associated with inflammatory responses, predominantly localizes within the nucleus. Upon chemotherapeutic treatment, HMGB1 can undergo translocation from the nucleus to the extracellular matrix. Surprisingly, the expression of HMGB1 was observed to be low in the PBS group but high in the APM group within the nucleus, contrary to the expectations based on Supplementary Figure 10. This discrepancy contradicts the data presented in Figure 2e.

Re: Thanks to your kind reminder, we repeated this experiment, and the results showed that the HMGB1 immunofluorescence of A2-APM had a lower rate of co-staining with the nucleus than that of the saline group, which suggests that A2-APM prompted the migration of HMGB1 from the nucleus of the cells. In addition, we separated the cytoplasm from the nucleus and examined the HMGB1 level in the nucleus by ELISA, which showed a decrease in HMGB1 secretion in the nucleus of PTX-associated nano-micelles.

Supplementary Figure 12. Intracellular HMGB1 level after different treatments. a, b After 24 h of co-cultivation with different treatments, the HMGB1 level was observed with a fluorescence microscope and the level of intracellular HMGB1 after receiving different treatments. Scale bar = 50 μ m. The experiment was conducted independently three times with similar results. Statistical significance was calculated by one-way ANOVA with Fisher's LSD test.

The description has been added to the results section (page 8 line 222-223) as:

the intracellular HMGB1 in PTX-contained micelles groups was lower than other groups (Supplementary Fig. 12).

The protocols have been added to the method section (page 31-32 line 864-867) as:

To evaluate the secretion of intracellular HMGB1, the cell nucleus was separated by Nuclear and Cytoplasmic Extraction Reagents (Hangzhou FUDE Biological. Technology CO., LTD. Cat. N0. FD0199) first and then detected with the mouse HMGB1 ELISA kit.

7. The release of PTX within the reductive microenvironment of GBM facilitates the induction of ICD in glioma cells. However, it is crucial to consider the potential impact of PTX release on the activity of infiltrated immune cells, including T cells and dendritic cells (DCs). It is

necessary to provide an explanation for how PTX-induced ICD might affect the functionality of these immune cells.

Re: Thank you for your insightful comment.

In *in vitro* experiments, co-culture of PTX with immune cells resulted in immune cell death. However, in *in vivo* experiments, an increased number in immune cells in the TME after A2-APM treatment were observed. This may be because the killing of immune cells by PTX is lower than the PTX-induced ICD effect, or it may be due to the fact that the majority of immune cells detected in the TME migrated from the periphery to the tumor locally, and thus PTX does not affect the activity of these immune cells. In the future, we would like to explore this area in more depth.

Since the immune microenvironment is composed of a variety of cells and they interact with each other, here we focus on the ICD effect induced by APM treated tumor cells on macrophages and DCs, and their effect on T cell activation *in vitro*. The results showed that APM promoted the expression of CD86 in macrophages and the expression of CD80 and CD86 in DCs, and increased the proportion of IFN γ ⁺CD3⁺CD8⁺ T cells.

Supplementary Figure 13. APM shifts macrophages toward a pro-inflammatory phenotype and enhanced T cell function. **a** Schematic illustration of the *in vitro* model to evaluate the effect of untreated tumor cells and APM treated tumor cells on macrophages and T cells. **b** MFI of CD86⁺ and CD206⁺ cells in F4/80⁺CD11b⁺ cells by flow cytometry. **c** Percentage of IFN γ ⁺ cells in CD3⁺CD8⁺ T cells by flow cytometry. n = 3. Statistical significance was calculated by one-way ANOVA with Fisher's LSD test.

Supplementary Figure 14. APM promoted DC maturation and activated T cell function. **a** Schematic illustration of the *in vitro* model to evaluate the effect of untreated tumor cells and APM treated tumor cells on DC maturation and mDC-induced T cell activation. **b** MFI of CD80⁺ and CD86⁺ cells in CD11c⁺ cells by flow cytometry. **c** Percentage of IFNγ⁺ cells in CD3⁺CD8⁺ T cells by flow cytometry. n = 3. Statistical significance was calculated by one-way ANOVA with Fisher's LSD test.

The description has been added to the results section (page 9-10 line 250-260) as:

To investigate the effect of APM treated tumor cells on macrophages or DCs and their effect on T cell activation, bone marrow-derived macrophages (BMDMs), bone marrow-derived dendritic cells (BMDCs) and T cells were extracted. Then BMDMs and BMDCs were treated with tumor cells or A2-APM treated tumor cells (Supplementary Fig. 15a and Supplementary Fig. 16a) before adding T cells into the culture dish. In the A2-APM treated tumor cell group, CD206, the marker of inhibition, was significantly decreased, and CD86, the marker of anti-tumor, was increased (Supplementary Fig. 15b). Meanwhile, the BMDCs were matured with the rise of CD80 and CD86 expression (Supplementary Fig. 16b). Additionally, T cell activation was observed with elevated IFN-γ expression level in CD3⁺CD8⁺ T cells.

The protocols have been added to the method section (page 32 line 874-887) as:

To evaluate the ICD effect on macrophages or DCs and their effect on T cell activation, we extracted primary BMDMs and BMDCs. In brief, the femurs, tibias and spleen were dissected from C57BL/6 mice. Then, the bone marrow was rinsed and cultured in a medium (DMEM containing 20 ng mL⁻¹ M-CSF for macrophage, RPMI1640 containing 20 ng mL⁻¹ M-CSF for DC) for 6 days, the spleen was sheared and digested, then sorted by magnetic beads and cultured in a RPMI1640 medium. On day 7, naive BMDMs were collected and cultured with a medium containing untreated tumor cells and APM treated tumor cells for 24 h. After

that, T cells were collected and cultured in the culture dish for another 24 h. Then the cells were washed with cold PBS, and further incubated with anti-F4/80, anti-CD86, anti-CD206 antibodies for macrophages, anti-CD11c, anti-CD80, and anti-CD86 antibodies for DCs, and anti-CD3, anti-CD8, and anti-IFN- γ antibodies for T cells for 30 min. Then the cells were resuspended in 500 μ L PBS, and examined using a flow cytometry (FACSCelesta, Becton, Dickinson and Company, New Jersey, USA), analyzed with Flowjo.

8. In line 237-238 on page 9, the author asserted that the PEG-shell effectively hindered the off-target distribution of free drugs, preventing their accumulation in healthy organs. However, it is essential to verify if any cited references support this claim. Additionally, the description of this mechanism appears inaccurate. In reality, the PEG-shell serves to prolong the circulation time of coated nanoparticles and impedes their recognition and phagocytosis by the mononuclear phagocyte system (MPS). As a result, the distribution of nanoparticles in the liver and spleen is reduced. However, it is important to note that the prolonged blood circulation time might increase their distribution in other organs such as the kidney and heart.

Re: Thank you for your kind advice. It was indeed poorly expressed here (the PEG-shell effectively hindered the off-target distribution of free drugs, preventing their accumulation in healthy organs.) and has been rewritten.

Further we observed fluorescence accumulation in major organs at 1.5, 24 and 96 h after injection of A2-APM in GL261 and G422 tumor mice. The prolonged circulation in the blood caused by the PEG-shell did increase the accumulation of nano-micelles in other organs, while the nano-micelles accumulated mainly in tumors and were gradually excreted over time. The time interval at which we inject the drug also ensures that most of the drug is excreted which may reduce the risk of side effects of drugs in other organs.

GL261-bearing mice

Supplementary Figure 26. A2-APM excretion from GBM mice over time. Semiquantitative biodistribution of Cy7.5-aPD-L1 in brain, heart, liver, spleen, lung, kidneys collected from mice injected with saline and A2-APM at 1.5, 24 and 96 h after *i.v.* injection (n = 4).

The description has been added to the results section (page 14 line 371-378) as:

To measure whether the prolonged blood circulation time might increase their distribution in other organs, saline and A2-APM (Cy7.5-labeled aPD-L1, aPD-L1 equivalent dose 1.5 mg/kg) were intravenously given to C57BL/6 mice bearing GL261 and G422 tumors. The mice were sacrificed at 90 min, 24 h and 96 h post-injection to extract the heart, liver, spleen, lung, kidneys, and brain (Supplementary Fig. 26). The semiquantitative results revealed that A2-APM was excreted from the body over time and did not accumulate in healthy organs.

The protocols have been added to the method section (page 38 line 1036-1041) as:

To measure whether the prolonged blood circulation time might increase their distribution in other organs, Cy7.5-labeled aPD-L1 was used to prepare fluorescent A2-APM, then saline and A2-APM (aPD-L1 equivalent dose 1.5 mg/kg) were intravenously given to the tumor mice. The mice were sacrificed at 90 min, 24 h and 96 h post-injection to extract the heart, liver,

spleen, lung, kidneys, and brain for *ex vivo* fluorescence imaging.

9. The G422 cell lines were derived by intracranial injection of methylcholanthrene in Kunming mice, which are frequently utilized as hosts. However, it is necessary to clarify the rationale behind employing C57 mice for constructing the G422 model. Furthermore, it is crucial to address whether the authors took into account the potential immunogenicity associated with the use of C57 mice in this context.

Re: When we first inoculated G422 cells in Kunming mice, we found that the tumorigenicity was low. Therefore, we reviewed the literature and found that G422 can also be grown in ICR and BALB/c mice (Sun, Rong et al. Nature communications vol. 13,1 5127. 1 Sep. 2022, doi:10.1038/s41467-022-32837-5; Xue, Nina et al. Scientific reports vol. 7 39011. 3 Jan. 2017, doi:10.1038/srep39011), so we tried to plant G422 cells in BALB/C, ICR and C57 mice and found that G422 cells had better tumorigenicity in the brains of C57 mice. However, we cannot guarantee whether the use of C57 mice in this situation will produce immunogenicity. Thus, we repeated the experiments on the G422 GBM mouse model with the Kunming mouse, including biodistribution, some of flow cytometry, such as:

Supplementary Figure 18. Distribution of aPD-L1 in G422 tumor mice. **a, b** Fluorescence imaging and signal intensities at tumor sites at 1, 2, 4, 8, 12, 24, 48 and 72 h after i.v. injection into G422 tumor mice with saline, aPD-L1, APM and A2-APM. **c, d** Representative *ex vivo* fluorescence images and semiquantitative biodistribution of Cy7.5-aPD-L1 in brain collected from mice injected with saline, free aPD-L1, APM, A2-APM at 12 h post i.v. injection. **e, f** Representative *ex vivo* fluorescence images and semiquantitative biodistribution of Cy7.5-aPD-L1 in major organs collected from mice injected with saline, free aPD-L1, APM, A2-APM at 12 h after i.v. injection. $n = 3$ saline treated mice, $n = 5$ free aPD-L1, APM and A2-APM groups. All statistics are expressed as mean \pm SD. Statistical significance was calculated by one-way ANOVA with Fisher's LSD test in **b** and **d**.

a**Gated on CD8⁺ CD3⁺ CD45⁺ cells****b**
Supplementary Figure 39. Analysis of the G422 tumor-infiltrated T cells after various treatments using flow cytometry. a Representative flow cytometric contour plots of tumor-infiltrating IFN γ ⁺, Granzyme B⁺ (GrzB⁺), PD-1⁺ and TIM-3⁺ T cells three days after two doses of saline, free aPD-L1, free PTX, A2-AM, A2-PM, APM and A2-APM. Insets: the subsets of IFN γ ⁺, Granzyme B⁺ (GrzB⁺), PD-1⁺ and TIM-3⁺ T cells in CD3⁺CD8⁺ T cells; the numbers indicate the percentage of IFN γ ⁺, Granzyme B⁺ (GrzB⁺), PD-1⁺ and TIM-3⁺ T cells in CD3⁺CD8⁺ T cells after various treatments. **b** Quantification of tumor-infiltrating IFN γ ⁺, Granzyme B⁺ (GrzB⁺), PD-1⁺ and TIM-3⁺ T cells three days after two doses of saline, free aPD-L1, free PTX, A2-AM, A2-PM, APM and A2-APM. All statistics are expressed as mean \pm SD, n = 3. Statistical significance was calculated by one-way ANOVA with Fisher's LSD test.

a**b**
Supplementary Figure 41. Analysis of the exposure of CRT on G422 tumor cells after various treatments. Representative flow cytometry histogram **a** and quantification **b** of the exposure of CRT on tumor cells three days after two doses of saline, free aPD-L1, free PTX, APM and A2-APM. $n = 3$ biologically independent samples. Statistical significance was calculated by one-way ANOVA with Fisher's LSD test.

10. In Supplementary Figure 19, a black signal was observed in day 21 in A2-APM group. Please explain this phenomenon.

Re: We measured this black signal and found that it showed low signal on both T1WI and T2WI and was higher than the ambient signal, which, combined with its shape, led us to believe that it was a hematoma left by surgical trauma. To avoid unnecessary misunderstandings, we replaced this image with another mouse from the same group.

11. In Supplementary Figure 25, the signals of CD3 seem to be less than those of CD4 and CD8. Please explain this. In addition, CD4 signals seem to be not overlapped by CD3 signals.

Re: Thank you for your careful check.

This may be due to the small size of the image given resulting in an insufficiently clear image, alternatively, the superimposition of signals of various colors may result in the weaker fluorescence being partially covered.

Note that the original images were used for the data analysis, so this did not affect the data analysis.

The images in the article have been replaced with clearer ones.

Here we give the enlarged picture as follows (A2-APM group, Alexa594-CD3; Cy3-CD4; FITC-CD8). To show the overlap of CD3 with CD4 and CD8, here we merge CD3 and CD4, CD3 and CD8, respectively. Due to the strong fluorescence of FITC-CD8, it is hard to observe the overlap of CD3 and CD8, we adjusted the transparency of CD8, and then merged with CD3.

12. In Supplementary Figure 29, the authors made reference to the exposure of calreticulin (CRT) on tumor cells. However, a noteworthy concern arises as the authors did not utilize antibodies or fluorescence labeling to specifically identify the tumor cells. It is crucial to thoroughly evaluate this description and ensure the appropriateness of the experimental approach. To address this concern, the authors are encouraged to employ tumor cells expressing green fluorescent protein (GFP) or red fluorescent protein (RFP) to conduct the corresponding experiments, thereby enabling precise visualization and characterization of the tumor cell

population of interest.

Re: Thank you for your kind suggestion. To better illustrate CRT expression on tumor cells, we established glioma models using GL261-GFP and G422-GFP and reevaluated the CRT expression in GFP tumor cells (Supplementary Figure 37b, c and 41). The gating strategy for CRT on GFP tumor cells was shown on Supplementary Figure 47b. The results showed that CRT on the surface of tumor cells was increased in the A2-APM group.

Supplementary Figure 47 b Gating strategy to sort CRT⁺ cells in tumor tissues gating on GFP⁺ Live⁺ cells presented on Supplementary Figure 37 and Supplementary Figure 41a.

Supplementary Figure 37. Analysis of the exposure of CRT on G422 tumor cells after various treatments. **b** Representative histograms display CRT expression levels compared to isotype controls. **c** MFI of each marker's expression were calculated based on (b). All statistics are expressed as mean \pm SD, $n = 3$, Statistical significance was calculated by one-way ANOVA with Fisher's LSD test.

a**b**
Supplementary Figure 41. Analysis of the exposure of CRT on G422 tumor cells after various treatments. Representative flow cytometry histogram **a** and quantification **b** of the exposure of CRT on tumor cells three days after two treatments with saline, free aPD-L1, free PTX, APM and A2-APM. $n = 3$ biologically independent samples. Statistical significance was calculated by one-way ANOVA with Fisher's LSD test.

REVIEWERS' COMMENTS

Reviewer #1 (Remarks to the Author):

The authors successfully addressed all of the remaining questions.

Reviewer #2 (Remarks to the Author):

The authors have addressed all my concerns during the revision

Reviewer #3 (Remarks to the Author):

The authors have addressed my critiques, and from what I see, the other reviewer's comments in a compelling way. I don't have further comments.

Reviewer #4 (Remarks to the Author):

Please see Confidential comments to the EDITORS ONLY